# Selective Laser Melting of Aluminum and Its Alloys

**DOI:** 10.3390/ma13204564

**Published:** 2020-10-14

**Authors:** Zhi Wang, Raghunandan Ummethala, Neera Singh, Shengyang Tang, Challapalli Suryanarayana, Jürgen Eckert, Konda Gokuldoss Prashanth

**Affiliations:** 1National Engineering Research Center of Near-net-shape Forming for Metallic Materials, School of Mechanical and Automotive Engineering, South China University of Technology, Guangzhou 510640, China; shengyangtang@yeah.net; 2Department of Mechanical and Industrial Engineering, Tallinn University of Technology, Ehitajate tee 5, 19086 Tallinn, Estonia; raghu.ummethala@gmail.com (R.U.); neerasingh1191@gmail.com (N.S.); 3Department of Mechanical and Aerospace Engineering, University of Central Florida, Orlando, FL 32816-2450, USA; surya@ucf.edu; 4Erich Schmid Institute of Materials Science, Austrian Academy of Sciences, Jahnstraße 12, A-8700 Leoben, Austria; juergen.eckert@unileoben.ac.at; 5Department of Materials Science, Montanuniversität Leoben, Jahnstraße 12, A-8700 Leoben, Austria; 6CBCMT, School of Engineering, Vellore Institute of Technology, Vellore 632 014, India

**Keywords:** additive manufacturing, selective laser melting, light metals, characterization, properties, applications

## Abstract

The laser-based powder bed fusion (LBPF) process or commonly known as selective laser melting (SLM) has made significant progress since its inception. Initially, conventional materials like 316L, Ti6Al4V, and IN-718 were fabricated using the SLM process. However, it was inevitable to explore the possible fabrication of the second most popular structural material after Fe-based alloys/steel, the Al-based alloys by SLM. Al-based alloys exhibit some inherent difficulties due to the following factors: the presence of surface oxide layer, solidification cracking during melt cooling, high reflectivity from the surface, high thermal conductivity of the metal, poor flowability of the powder, low melting temperature, etc. Researchers have overcome these difficulties to successfully fabricate the different Al-based alloys by SLM. However, there exists no review dealing with the fabrication of different Al-based alloys by SLM, their fabrication issues, microstructure, and their correlation with properties in detail. Hence, the present review attempts to introduce the SLM process followed by a detailed discussion about the processing parameters that form the core of the alloy development process. This is followed by the current research status on the processing of Al-based alloys and microstructure evaluation (including defects, internal stresses, etc.), which are dealt with on the basis of individual Al-based series. The mechanical properties of these alloys are discussed in detail followed by the other important properties like tribological properties, fatigue properties, etc. Lastly, an outlook is given at the end of this review.

## 1. Introduction

Additive manufacturing (AM) is one of the modern manufacturing processes where a 3D component is fabricated by progressively stacking and solidifying several 2D layers based on a digital model (metal incremental manufacturing) [1,2]. This is in contrast to the conventional manufacturing methods of removing an unwanted volume of the material from the bulk, namely an ingot/billet/bloom, to form a useful industrial component. This is now day commonly referred to as subtractive manufacturing (Figure 1). Ever since the first techniques in the area of rapid prototyping (RP)/additive manufacturing were developed (in the late 1980s), new advances have continuously emerged. AM, as a rapidly developing technology in the field of advanced manufacturing technologies, has positioned itself as a front runner and will perhaps be the main manufacturing method in the manufacturing sector in the decades to come, if not in the very near future [1,2,3,4,5,6]. Because of its popularity and the rapid growth of progress in this area, a number of reviews have appeared in the literature [2,7,8,9,10,11].

In this review, we would like to focus on aluminum and its alloys. This is the first review of its kind, where every Al-based alloy system is reviewed in detail with respect to the formation of microstructure and/or defect generation during laser fabrication, which makes this review unique. This will be followed by a review of the different properties of AM-Al and its alloys, including mechanical properties (tensile and compressive), tribological properties, fatigue properties, thermal conductivity, and weldability. A detailed outlook is presented at the end of the review, also suggesting future possibilities of fabricating Al and its alloys by additive manufacturing. The AM technology is also referred to as a freeform fabrication process (FFF-P) or digital manufacturing or e-manufacturing, which integrates the following branches of science and technology: physics, mechanical engineering, electrical and electronics engineering, materials science, design, and chemistry. AM technology offers the advantage of fabricating parts with extreme complexity at no additional cost. AM has attracted the attention of researchers mainly because of the following reasons (Figure 2):(a)Design freedom: The AM process does not involve tooling and machining and, hence, in the design process, an extra degree of freedom exists and design changes can be included at any time with ease.(b)Speed: The productivity of fabricating components of various shapes and intricate structures goes up because a plethora of sophisticated designs can be incorporated in a single build job by simply modifying the digital model of the component.(c)Cost: The elimination of conventional mold/die making and subsequent machining of parts results in extensive savings of both direct and indirect costs, thereby favourably influencing the affordability of the end-use part.(d)Environmentally friendly: Since AM leads to minimal waste of material, the process is considered to be environmentally friendly and(e)Convenience: The design and production processes can be carried out with convenience and ease and with minimal labour [13,14,15].

Several additive manufacturing processes were developed in the late 1980s and early 1990s. However, not every process is intended for the production of metallic components. Fusion deposition modeling (FDM) is one of the processes that uses a polymer filament/metal wire that is unwound to an extrusion nozzle. The materials are extruded via the heated nozzle, where the melting of the material and extrusion take place simultaneously [16]. The next-generation AM processes for metals were unveiled by Carl Deckard at the University of Texas Austin with the help of his PhD advisor Prof. Joseph Beaman. They developed the selective laser sintering (SLS) process in 1984, which was then duly patented [5,17]. The SLS process reached its demand as the industries started looking into AM as an alternative technology for manufacturing. The SLS process, however, may result in the presence of internal defects such as porosity that leads to poor mechanical properties of the parts. The demand to produce fully dense parts with minimal defects, improved functionality and superior properties have led to the invention of the selective laser melting (SLM) process, which shares the common working principle with the SLS process, except that in the former, the powder particles are fused completely by melting rather than just sintering [18].

Additive manufacturing (AM) is defined by ASTM as the “process of joining materials to make objects, usually layer by layer, from a 3D CAD data” [19]. The first successful attempt in the field of AM came from the technology developed in the 1970s, through additives. Earliest roots can be traced to topography and photo sculpture both first developed in 1890s to replicate objects [20,21]. Additive technology developed rapidly throughout the 1980s and 1990s and it has seen increasing industrial application in the last 20 years. Initial commercialization of an AM process took place in 1987 in the form of stereolithography from 3D systems [22]. Stereolithography is a process that uses a combination of both photochemistry and laser technology to build parts from photopolymer resins. SLA-1 is the first commercially made AM system and was popular with the name of SLA 250 machine (SLA—Stereolithography apparatus). The commercial SLA 250 was replaced by the next-generation Viper SLA product from the 3D system. First-generation acrylate resins were commercialized in 1988 and this invention is a collaborative work from the 3D systems and Ciba-energy.

The invention of new AM technologies slowly gained momentum in 1991, when three new AM technologies were commercialized. These include fused deposition modeling (FDM) from Stratasys [23], solid ground curing (SGC) from Cubital [24], and laminated object manufacturing (LOM) from Helisys [25]. FDM is a process, which uses a plastic filament or a metal wire that is unwound to an extrusion nozzle. The nozzle is heated to melt the material and at the same time, the material is extruded from the nozzle. The material hardens immediately upon extrusion from the nozzle. SGC uses a UV-sensitive liquid polymer, solidifying the entire layers in one pass by UV light through masks created using an electrostatic toner on a glass plate. LOM is an AM process that bonds and cuts sheet material using a digitally guided laser. The next-generation AM processes were commercialized in 1992, when both selective laser sintering (SLS) from 3D systems and solid form stereolithography system from Teijin Seiki were unveiled [26]. SLS fuses powder particles (plastics/metal/ceramic/glass) using the heat from a laser to produce a 3D component. The next innovation shaped in the form of a multi-color 3D printer from Z Corporation [27]. 3D printers produce parts and assemblies in a similar fashion to a standard inkjet printer, however it spreads layers of plaster or resin powder and binds them together. 3D printing is faster, more affordable, and easier to use than other additive technologies. Selective laser melting (SLM) got its birth and definition at the end of 1994 from the Fraunhofer Institute ILT in Aachen, Germany, and is patented immediately (ILT SLM patent DE 19649865). SLM uses a 3D computer-aided design (CAD) data as a digital source of information and energy in the form of a high-powered laser beam to create 3D metal parts by fusion of metallic powder particles [28]. SLM is one of the few processes in the arena of AM that plays a key role even until the date and so has received considerable attention and importance.

The inventions in the field of AM are never-ending. However, the AM processes can be classified into four different categories depending on the process used, the materials processes, the deposition mechanisms involved, and the source of energy used during the process (Figure 3). For instance, the various AM technologies use different processing techniques for the production of the parts/objects. The FDM technology uses an extrusion process whereas digital light processing (DLP) uses a polymerization process for building 3D parts. SLS uses the sintering process and ultrasonic consolidation (UC) utilizes ultrasonic waves for the production of parts. In addition, there are different types of energy sources that are used in the AM technologies, such as DLP which uses ultraviolet radiation, ultrasonic waves are used in UC, the electron beam is utilized in the electron beam melting (EBM) process, and a laser beam is used in both SLM as well as SLS processes.

### 1.1. Selective Laser Melting

Selective laser melting (SLM) is one of the additive manufacturing processes that fall under the category of powder bed fusion processes. This technique was developed by Fockele and Schwarze of Stereolithographietechnik GmbH along with Meiners, Wissenbach, and Andres of ILT Fraunhofer, Aachen in 1994 to produce three-dimensional metallic components from metallic powders. A patent for this technology was first applied in the year 1997 to the German Patent and Trade Mark Office, which was then officially accepted and published in 1998 [29]. This powder-bed fusion process uses a high-intensity laser beam as the source of energy to melt the metallic powder selectively, which is dictated by three-dimensional computer aided design (CAD) data. The production of components by this process can be divided into two important steps, namely (1) the computation part and (2) the actual production sequence. The parts are produced generally on a substrate plate or a base plate made of the same/similar materials. After the production of the part, it has to be detached from the base plate and for easy removal, hollow structures called ‘support structures’ are designed in between the base plate and the actual component. The different steps involved in the production of components by the SLM process are shown in Figure 4. The first step involves creating a 3D CAD model exactly resembling the part to be fabricated. The 3D CAD model of the actual component is fed to the printer, along with the data for the support structures, and this is then known as absolute data (data of the part + data of the support structures). Since the laser needs information about the part on a layer-by-layer basis, the absolute CAD data is sliced into layers. This sliced data (layer-wise 2D information) can then be used for fabricating the part using SLM.

The second step in the SLM process involves the fabrication of the actual part. A thin layer of powder is spread over the base plate or over the previously spread powder layer using a specially designed loader setup. The loader is then taken away from the path of the laser and the laser beam is activated. The laser beam melts the powder bed selectively as dictated by the 3D CAD data. The platform is then lowered by an amount equal to the layer thickness, which is predefined based on other process parameters and the powder to be processed. The next layer of powder is then coated and the process continues until the entire part is fabricated (Figure 4). Upon completion of the process, the base plate, along with the fabricated part, can be taken out of the build chamber and the two can be separated easily by inducing cracks at the support structures. Process parameters like the laser power, laser scan speed, hatch distance, hatch style, layer thickness, powder particle size, and physical properties of the powders, determine the density of the fabricated part. The process parameters may be tuned to improve the density of the laser fabricated parts (Figure 5).

#### 1.1.1. Advantages

The SLM process has the ability to fabricate parts without the use of any dies or tooling, which may help to shorten the design and production cycle, may result in saving production time, and in turn costs [6]. Moreover, the SLM process allows the re-use of the metal/alloy powder in an efficient way, since it offers the possibility to sieve and recycles/re-use the powder. The recycling ability is an important advantage of the SLM process since this leads to minimal waste of material and SLM is, thus, regarded as an environmentally friendly process [13,14]. It has been established that the powders can be used between 12 and 14 times after recycling, with no significant changes in the powder properties as well as that of the parts [14]. The SLM process needs fewer raw materials to produce components that can have added functionality, lightweight structures for weight reduction and exotic designs for optimal performance. Hence, SLM is considered to allow for a considerable reduction in fuel emissions, which confers the process of an outstanding ecological performance making it a green technology for the future [30]. The SLM process has already made a considerable impact in the manufacturing, automobile, aerospace, pharmaceutical, electronics, and sports sectors [31,32] because of its ability to produce a wide variety of materials (metals, alloys, cermets, composites) without any theoretical restrictions [33,34,35,36,37,38,39,40,41,42,43,44,45,46,47,48,49,50].

#### 1.1.2. Drawbacks

Even though the SLM process has several merits to be considered as the technology for the future, there are certain drawbacks such as poor surface quality, occasional high production time, dimensional accuracy and material properties (some alloys show undesirable brittleness for industrial applications). To overcome these problems, post-processing treatments such as polishing including electro-polishing, heat treatments, surface grinding, furnace infiltration, etc. can be performed [51]. Extensive research has been carried out to find a balance between the process and material parameters that can be employed for obtaining parts with better surface finish (by introducing contours), optimized production time (by manufacturing several components in the same build job), appropriate mechanical properties, increased dimensional accuracy (smaller melt pool, smaller layer thickness), and at the same time shorter production cycle. The installation and machine costs play a crucial role since the machine costs are quite high. In addition, the raw material costs are also high, when it comes to powder-based additive manufacturing. Mostly, gas atomized powders were required, which accelerates the cost of the part. However, this powder cost may be marginally overcome by using elemental powder instead of pre-alloyed powder. On the other hand, using elemental powders will introduce other issues like segregation [33]. However, the properties of the powder, process parameters, and the properties of the alloy itself play a significant role in the fabrication of end-use metallic components using SLM.

#### 1.1.3. Powder Characteristics

The properties of the powder play a significant role during the fabrication of a component by the SLM process. Various physical properties of the powder like the melting point, density, latent heat of fusion, thermal conductivity, and thermal heat capacity determine the melt enthalpy and in turn the density and mechanical properties of SLM-fabricated parts. The amount of energy that is required to melt a unit mass of powder completely is defined as the melting enthalpy. The melting enthalpy further determines the heat balance of the system, which in turn determines the characteristics of the melt pool [52,53,54,55,56]. One of the important factors that determine the heat balance of the system is the thermal conductivity. Elements/alloys with high thermal conductivity need higher energy in order to have a stable melt pool. The amount of internal stresses that are introduced into an SLM-fabricated part during the solidification process depends on the coefficient of thermal expansion of the system, and the aforementioned properties are intrinsic properties of the metal/alloy system, which cannot be altered significantly [52,53].

However, factors like the particle size distribution, which is a mathematical function, can be altered depending on the requirements. The particle size distribution defines the relative number of particles by mass, present in different size ranges [55,56,57,58]. This is one of the important powder properties, which determines the flowability of the powder. The ratio of larger to smaller particles in powder has a strong influence on its flowability [57]. Particles tend to stick or agglomerate with each other if the particle size distribution of the powder contains a large fraction of very fine particles, which influences the flowability of the powder. This may be explained in terms of the pronounced van der Waals forces experienced by small particles that negatively influence the uniform deposition of the powder layers [58,59,60]. On the other hand, a high volume fraction of finer particles helps to reduce the energy required for melting and decreases the layer thickness. This aids in further reducing the energy input required for the fabrication process (either the power quantity can be reduced or the laser scan speed can be increased, which increases the speed of the fabrication process) [61,62,63,64]. The consistency of the melt can be improved to a certain extent by introducing powder particles with a narrow size distribution, however it may have detrimental effects on the packing density of the powder [61].

Mazumder et al. [62] reported that the laser beam has to diverge through a larger distance through the powder bed to form a melt pool if the layer thickness of the powder is increased. Under such circumstances, there may be an asymmetry in the pool width between the top and the bottom layers, with the melt pool at the bottom being significantly larger than at the top. This asymmetry in the melt pool width can be minimized by reducing the layer thickness of the powder bed by using finer particles. The particle size and the particle size distribution not only determine the parameters to be used for the process, but also influence the quality of the SLM-fabricated components [55]. For example, a finer particle size leads to an improved surface finish of the components [60]. The other important property of the powder is the shape of the particles, which strongly depends on the powder preparation method. For instance, powder particles produced by conventional mechanical milling/alloying tend to have an irregular shape [63]. On the other hand, gas atomization leads to the formation of spherical powder particles. Spherical powder particles improve the flowability of the powder, which in turn improves the quality of the final part [64]. Non-spherical powder particles lead to lower compaction density, which in turn leads to an increased level of defects (porosity) in the SLM-fabricated parts, which is detrimental for high-performance applications [62]. The powder density itself is another important property to be considered. The density can be classified as the density of the individual particles or powder packing density. The density of the powder (or the material) is an intrinsic property of the material and the packing density depends strongly on the morphology of the powder, namely size, shape, and size distribution. The packing density determines the thermal conductivity of the powder bed, which depends on the number of contact points existing between the powder particles. A higher packing density leads to higher number of contact points between the powder particles, thereby aiding uniform heat transfer across the powder layer. The above dependencies suggest that the particle morphology (size and shape), along with the particle size distribution, plays a major role in determining the quality of SLM components. Other properties, like the heat absorption coefficient (which can be influenced by varying the bed temperature by employing suitable heating systems) [31], viscosity of the melt pool (that can be determined by the laser power) [31], and surface free energy, also play a role during the SLM processing. However, they are of lesser significance compared to the above-discussed powder properties.

#### 1.1.4. Process Parameters

The SLM process is very complex in nature. Several process parameters can influence the fabrication process and, consequently, the quality of the final component. Among others, parameters such as laser power, laser scan speed, spot size of the laser, hatch spacing and style, and the layer thickness are the most critical factors that have a strong influence (Figure 6).

#### 1.1.5. Energy Density

The quality and properties of SLM parts strongly depend on the above parameters that in turn define the energy density or the energy input. The energy density *E_d_*, is defined as the amount of energy that is supplied to a unit volume of powder during the melting step and is expressed as:*E_d_* = *P*/(*v_s_ h t*) (J/mm^3^)(1)
where *P* is the laser power (W), *v_s_* is the laser scanning speed (mm/s), *h* indicates the hatch distance (mm), and *t* represents the layer thickness (mm) [65,66]. To elucidate the influence of the SLM process parameters on the mechanical properties of the end product, some experiments performed on the near-eutectic Al-12(wt.%)Si alloy will be discussed below. The optimized process parameters for printing dense Al-12Si parts, reported elsewhere, were taken as the reference point [66,67,68,69]. During the process conducted in an SLM 250 HL device from SLM solutions, the laser power was gradually reduced from an optimum value of 320 W to 80 W in 6 steps of 40 W each. The laser scan speed was held constant at 1455 mm/sec. Tensile tests were carried out on these 7 samples and it was observed that the strength (both yield strength and ultimate tensile strength), as well as the ductility of the Al-12Si samples decreased gradually as the laser power was decreased (Figure 7). This is because, as the laser power decreases, the effective energy supplied to the powder bed decreases, and this results in improper melting of the metal powder creating defects in the solidified parts causing poor mechanical properties [68].

Equation (1) describes the amount of energy supplied to the powder bed in the form of energy density. However, this parameter is empirical and is derived more as a rule of thumb. For example, the two parameters, laser power, and laser scan speed were varied to maintain a constant energy density at a fixed layer thickness and hatch distance. The laser power was varied between 320 W and 40 W and the laser scan speed was varied between 1455 mm/s and 182 mm/s, respectively, to maintain a constant energy density of 55 J/mm^3^ for all the 8 test specimens. The tensile test results in Figure 8 show that the sample fabricated with the highest laser power and laser scan speed exhibits the highest strength (for the parameters: 320 W and 1455 mm/s, which is the optimized parameter). The strength and ductility of the samples decrease gradually when both laser power and laser scan speed are decreased. These findings illustrate that both laser power and laser scan speed are two individual parameters that are crucial for achieving optimum mechanical properties of SLM samples. The energy density parameter, as widely used by many researchers dealing with parameter optimization studies [69,70] is only empirical and the influence of the processing parameters cannot be strictly judged based solely on the energy density [68].

#### 1.1.6. Hatches and Hatch Styles

In order to understand the influence of hatches and hatch styles, one needs to understand the basics behind the fabrication process. Hence, we now focus on the building sequence carried out during the SLM process. A focused laser beam melts the powder particles selectively over a powder bed. Generally, the layer thickness of the powder bed is kept constant, depending on the average powder particle size. Hence, this is a fixed term in the calculation of energy density.

The area to be melted on the powder bed is usually several times larger than the diameter of the laser beam. Considering the size of the laser spot as ~80 µm, the laser beam cannot melt the entire scan area in a single exposure. Hence, the laser beam is made to traverse along a single track in the powder bed at a time, which corresponds to a width of ~80–120 µm, depending on the material and the laser parameters (Figure 9). This single track, which the laser melts in every operation (exposure), is known as hatch (with a distance ‘a’—Figure 10), and is regarded as the building block for the SLM process. The laser has to make several hatches to melt the entire scan area in a single layer, as shown in Figure 9. In order to ensure the soundness of the sample and to eliminate porosity at the interface of two consecutive hatches, every hatch has an overlap, known as hatch overlap (∆x), with its former one. The extent of hatch overlap also depends on the material to be processed and the laser parameters. Hence, the effective track distance is the difference between the hatch distance and the hatch overlap (a − ∆x). In most of the SLM processes, an overlap of ~20% or more is maintained to ensure a better quality of the SLM parts [53]. In order to complete a 3D part, the SLM process involves several hatches within each layer and several layers within each part (Figure 10). Hatches play an important role in the SLM process [12,67]. The arrangement of hatches within and between the layers is defined as the hatch style. The hatch style plays a major role in dictating the microstructure of the SLM part and hence its properties [12].

The hatch style can be varied in a number of different ways and its design depends on the creativity of the user and on the specific requirements of the SLM part. Some of the possible hatch styles (also known as laser scan strategy) are shown in Figure 11. For example, the first hatch style, designated ‘A’ (in Figure 11), represents a pattern in which the hatches are aligned in a unidirectional vector and rotate 90° for every consecutive layer, which is called ‘single melt hatch style’. On the other hand, the hatch style ‘B’ shows bidirectional vectors, where two neighboring hatches do not have the same direction. The direction is reversed between each hatch and this type of hatch style is referred to as single melt continuous. Similarly, there is the possibility that each layer can be melted twice before the next layer of powder is applied, which is termed as ‘double melt hatch style’. One of the most popular hatch styles that is prominent in the SLM community is the checkerboard hatch style (designated as ‘D’ in Figure 11) [71,72,73,74]. These hatch styles may be repeated every layer with or without the presence of hatch style rotations between the layers. The rotation of the hatch style between two consecutive layers is expected to improve the bonding between the layers [75]. All the aforementioned variables, namely hatch distance, hatch style, and hatch rotations, are expected to have a strong influence on the microstructure and properties of the final parts produced by SLM [12,71,75,76].

#### 1.1.7. The Contour

Similar to the hatch style, contours have a significant effect on the quality and surface finish of SLM parts. Figure 12 shows a contour, which is the outermost layer/surface of the SLM part usually with a thickness of 0.2–0.5 mm. The surface quality such as roughness is important for aesthetic or functional purposes. However, generally, the contours are subjected to different process parameters than those used for processing the bulk of the part. Generally, the laser power may be held the same but the contour layer is melted at a faster speed to have a smoother surface finish. An intermediate layer between the bulk of the part and the contour may also be present, identified as contour offset. The building sequence of an SLM part is melting of the bulk volume first, then the contour offset and finally the contour, depending on the shrinkage levels observed in the specimens. This sequence may, however, be altered based on the needs and material shrinkage properties.

## 2. Microstructure

### 2.1. Surfaces

Al and its alloys pose a plethora of challenges for fabrication through additive manufacturing techniques. While the characteristics of the powder, composition, and process parameter selection themselves require meticulous optimization, the rather poor laser-aluminum interaction is an additional challenge requiring serious attention [77,78,79]. Al and its alloy powders are highly reactive to oxygen and nitrogen and readily form thermodynamically very stable oxide, nitride or oxynitride films on the surface of the powder particles that are very hard to reduce [80]. Firstly, aluminum oxides have a higher melting point compared with the pure metal and, hence it will impede the melting of the powder particle [31]. Secondly, the oxides are not compatible with the molten metal due to their poor wetting characteristics, thus leading to pore generation within the solidifying material, which deteriorates the material properties. Oxide films can be easily formed on the melt surface, even at low concentrations of oxygen [66]. Hence, to break the stable oxide films and maintain a stable melt pool, sufficient thermal energy has to be imparted to the metal powders [80]. High laser power, combined with low laser scan speeds and close overlap distances (high-energy input), may be employed to impart high thermal energies during the SLM process. Nevertheless, oxide particles may still be encapsulated in the melt pool. Using low scan speeds will further lead to increased production time and higher overall manufacturing costs. On the other hand, imparting high thermal energy to the powder bed leads to the formation of a large melt pool, which may be uncontrollable and may result in a detrimental balling effect [80]. Other adverse effects of the oxides include:Hindering of diffusion, resulting in the presence of un-melted particles.Formation of weak zones in the end parts due to the brittle nature of oxides (ceramics).Formation of pores due to poor wettability of the oxide particles with the liquid metal [81,82] and also due to the entrapment of gases in the melt pool.

### 2.2. Porosity

The porosity present in an SLM part can be categorized into two types: (a) metallurgical pores [83] and (b) keyhole pores [84,85]. Pores resulting from hydrogen gas entrapment are generally spherical in shape and less than 100 µm in size and are termed as metallurgical pores. On the other hand, irregularly shaped pores that are much larger than 50 µm in size are defined as keyhole pores [86,87]. Such pores may arise from keyhole instability. The keyhole instability is attributed to the rapid cooling associated with the SLM process. Fast cooling impedes the molten metal to flow into the gaps and fill them completely leading to irregularly shaped pores. The metallurgical pores are created at very slow scanning speed due to the entrapment of gases within the melt pool due to pick up either from the atmosphere or from the powder [88]. Proper setting of the parameters is required to eliminate these porosities and to manufacture theoretically dense samples [89,90,91].

### 2.3. Flowability

Poor flowability also leads to defects and anisotropic properties in the SLM parts. The SLM process needs a stable, flat and uniformly distributed powder layer over the substrate or a previously deposited layer. Aluminum, being a light metal and in the presence of moisture, exhibits poor flowability, often leading to powder bridging inside the chamber and may result in clogging of the flow channels or cause hinders uniform powder flow onto the powder bed or itself. The poor flowability of the powder is also ascribed to strong inter-particle cohesion, which is associated with van der Waals forces [56]. In addition, strong inter-particle cohesion leads to the agglomeration of powder, which severely affects its flowability. None of the present-day deposition mechanisms can yield a perfect and uniform powder bed with Al and its alloys, irrespective of the particle size and shape used for the manufacturing process. The presence of non-uniform powder spread over the bed will often lead to defects in the fabricated parts and non-uniform properties of the parts.

### 2.4. Laser Sources

There are several commercially available laser sources, which can be incorporated in an SLM device. Some of the common laser sources are Nd:YAG and CO_2_ lasers, of which the former is more widely employed due to its ability to generate high output power and a finer focus [92]. Nd:YAG lasers have the advantage of exhibiting high absorptivity, which may provide larger penetration depths at low power [92,93]. However, Al and its alloys are highly reflective to the laser wavelengths of around 1.06 µm (laser wavelength for Nd:YAG laser), which is typically used in the SLM process. Under such conditions, the amount of energy absorbed will be less than 10% of the incident energy from the source [77,78]. Hence, to overcome the reflectivity issues in Al and its alloys, laser energies much higher than the theoretical values are applied. A significantly high-temperature gradient exists within the powder bed, the solidifying layers, and the overlaps that can lead to a detrimental balling effect [80].

### 2.5. Processability Issues

Al and its alloys are known for their high thermal conductivity and/or thermal diffusivity (thermal conductivity represents the ability of the materials to conduct the heat. The rate of heat transfer in materials is characterized by the thermal diffusivity), suggesting that heat is dissipated through such materials at very high rates. Thus, the powder bed not only reflects the laser beam but also conducts the heat rapidly away from the melt pool into the previously solidified layer and to the powder bed surrounding the melt pool or the substrate plate. Hence, it requires an additional supply of energy to compensate for the rapid conduction of heat from the melt pool [94]. Such high energy supplied to the powder bed will result in a significantly wider melt pool that creates limitations in the minimum size of the features that can be produced by the SLM process. In addition, high heating and cooling rates that lead to significant thermal gradients around the melt pool may cause undesirable levels of cracking. Relatively higher viscosity is observed in the melt pool owing to the relatively large difference between the liquidus and solidus temperatures resulting in poor fluidity and weldability, which could lead to insufficient filling of the melt during solidification. It has been demonstrated by researchers recently that the cracking problem in the 7075 alloys may be solved by adding elements that form a secondary eutectic, thereby increasing the fluidity (decreasing the viscosity of the melt) at a given temperature, where a high viscosity of the melt and shrinkage can cause solidification cracking [95].

In summary, the problems associated with Al and its alloys for fabrication by SLM are: (a) Oxidation of the surface of the metal powder, (b) obstructed flowability of the powder, (c) low absorptivity of the laser beam and high reflectivity to energies corresponding to a wavelength of 1.06 µm, (d) high thermal conductivity and, hence, wider melt pools placing restrictions on the size of the smallest features in the part that can be fabricated, (e) high solidification shrinkage, and (f) high viscosity of the melt [92,95,96,97,98,99,100]. Such difficulties may result in undesirable microstructures resulting in poor properties of SLM parts. In addition, Al and its alloys may face problems such as (a) porosity—improper processing parameters; (b) balling due to too high-energy input; (c) formation of a distorted layer due to too high-energy input and/or with the presence of brittle parts; (d) increased cracking tendency due to the brittle nature of the processing material; (e) high surface roughness of parts because of coarse powder and/or too high energy input; (f) loss of alloying elements, especially when the alloy contains elements with low boiling points and vapor pressures that can be lost during the SLM process; and (g) poor dimensional accuracy due to the presence of oxide layers, where the energy input has to be increased unprecedentedly [6,92,101].

Until now, the most studied SLM processed Al-based alloys are pure aluminum, Al-Si alloys, AlSi10Mg, Al-Cu alloys, and Al-Zn-Mg-Cu alloys. A few studies were devoted to some other aluminum alloys such as Al-Mg based alloys [102,103] and nanocrystalline alloys [104]. For instance, the SLM-fabricated Al_85_Nd_8_Ni_5_Co_2_ alloy reveals a hybrid nanostructure (nanocrystalline alloy), which results in very high strength at both room and elevated temperatures [104,105,106,107,108,109]. Therefore, one of the most important challenges is to find ways to resolve the above-discussed problems. Moreover, finding Al alloy compositions that are particularly suitable for the SLM process is vital for producing high strength Al alloys [110]. Al alloys have been studied and applied extensively for more than a century. Along these lines, different alloying elements have been used, but the amount of these alloying elements is typically very small. However, compared with conventional production processes such as casting or powder metallurgy, SLM has major differences in terms of a very high heating/cooling rate and a much smaller melt pool. Therefore, the well-established Al alloy compositions may not necessarily be suitable for production by SLM. In the last few years, an enormous amount of research has been carried out to develop new Al alloy compositions suitable for SLM that would solve the problems listed above and these will be discussed in a systematic fashion. The following sections will discuss different Al-based alloys processed by SLM according to their designated series.

### 2.6. Pure Aluminum (1XXX Series)

Aluminum is the most heavily consumed non-ferrous metal in the world for engineering applications [111]. It has a unique combination of attractive properties such as low density (lightweight), high specific strength and resistance to corrosion, making it an ideal material for use in innumerable structural applications. The burgeoning applications of aluminum are distinctively evident in the production of automobile parts, packaging of food and beverages, production of defense and aerospace equipment, transmission of electricity, and so on. Pure aluminum products can be easily fabricated and are processed by conventional processes such as casting and powder metallurgy; however, fabrication by SLM is particularly difficult. Besides the challenges mentioned above, e.g., oxidation, flowability, high reflectivity to the laser beam, etc., commercial purity aluminum exhibits a much higher thermal conductivity compared to its alloys owing to the absence of other solute elements that lower the conductivity, thus rendering fabrication by SLM more challenging. Therefore, SLM of pure aluminum has been rarely reported until the systematic study performed by Kimura and Nakamoto [112].

Defects such as irregularly shaped/spherical gas pores and fine oxide particles are prevalent in SLM-fabricated pure aluminum parts. The relative density of the SLM parts is significantly affected by the energy density of the incident laser beam. Too low input energy density leads to the presence of un-melted powder particles and irregularly shaped pores, while too high input energy density results in abundant spherical gas pores due to the sputters of molten and solidified metals. The gas trapped in the pores comes from the Ar/N_2_, which is used during the process or from the residual gases like hydrogen present in the powder. The SLM parts usually also contain large amounts of fine oxide particles (less than 0.3 µm) in the aluminum matrix, which might be formed from the pre-existing oxides on the surface of the atomized powder. The oxides on the powder particles may be broken into many smaller particles due to the rapid thrust of the high-energy laser beam, which can introduce large thermal stresses in the oxides. There is a peak relative density of pure aluminum SLM parts as a function of the input energy density, which is more than 99% for an energy density, *E_d_* = 60–80 J/mm^3^ in the work done by Kimura and Nakamoto [112]. However, the *E_d_* value also depends on other process parameters such as laser scan speed and laser power. The SLM pure aluminum parts show a cellular microstructure resulting from the laser scan traces. A clear texture of the (101) plane developed toward the stacking direction (ND) in the horizontal plane, and an elongated columnar crystallographic microstructure along the stacking direction in the vertical plane [112]. However, the production of pure-Al by SLM [112] is not of much interest due to the: (1) the poor mechanical properties of pure Al and (2) SLM processing difficulties in producing bulk defect-free components. Hence, pure Al SLM parts are not of considerable interest.

### 2.7. Al-Cu-Based Alloys (2XXX Series)

Al-Cu—2XXX is one of the well-established heat treatable Al-based alloys [113]. It has been used for decades in the automobile and aerospace sectors and SLM processing of Al-Cu—2XXX series alloys are discussed below.

#### 2.7.1. Al-Cu Alloys

SLM of Al-Cu alloys was first attempted by Bartkowiak et al. [114]. Custom-developed Al-Cu powders were used, where the composition varied from Al-5Cu to Al-11.8Cu. Al and Cu powders with average particle diameters of 35 µm and 3 µm, respectively, were blended together. The blended powders are shown in Figure 13. The finer Cu particles are attached to the surface of Al particles and are distributed uniformly. A single laser exposure via a single scan/track on the Al-Cu alloy powder created a crack-free, high-density structure with no brittle/hard surface oxidation layer. Due to the high cooling rates during SLM, a very fine microstructure was established. Wang et al. reported on the microstructure and mechanical properties of bulk Al-Cu alloys produced by SLM [115]. Al-xCu powders (x = 4.5, 6, 20, 33 and 40 wt.%) (particle size: 20–60 µm) were obtained by mixing Al-4.5Cu pre-alloyed powder with Cu powder (particle size ≤ 63 µm). After SLM, the Al-xCu SLM samples showed no visible XRD peaks corresponding to elemental Cu, suggesting the formation of an Al(Cu) solid solution and/or intermetallics. The Al_2_Cu intermetallic was found in all the samples with a relative volume fraction increasing from 10 wt.% to 77 wt.% corresponding to the Cu contents of 4.5 and 40 wt.%, respectively. The microstructure revealed an inhomogeneous distribution of the solute (Cu) phase owing to the different cooling rates along the width of the melt pool and the limited Cu diffusion into the Al matrix. Different microstructural zones were observed: a high-cooling-rate zone (HCRZ), a low-cooling-rate zone (LCRZ), and a heat-affected zone (HAZ). The Al_2_Cu phase in the Al-4.5Cu alloy is distributed uniformly and shows a fine granular microstructure in the HCRZ, whereas a lamellar (plate-like) Al_2_Cu is formed in the LCRZ. With increasing Cu content, the size of the Al_2_Cu phase increases significantly, and the morphology changes from eutectic to a hypereutectic microstructure. The Rapid solidification during SLM with shallow and narrow melt pool prevents complete dissolution of Cu in Al, especially at high Cu contents leading to the presence of Cu particles and Cu-rich clusters, as shown in Figure 14.

However, recently Pauly et al. have demonstrated formation of a lamellar eutectic in an Al-33Cu alloy fabricated by SLM from gas atomized Al-33Cu powders [116]. The Al-33Cu composition shows a perfect lamellar eutectic microstructure. The width of the lamellae may be adjusted by varying the cooling rate (i.e., by modifying the process parameters; especially laser power and laser scan speed). In addition, Pauly et al. were able to demonstrate that the volumetric energy density is an inaccurate measure for inferring cooling rates, at least for this alloy [116].

#### 2.7.2. Al-Cu-Mg Alloys

Al-Cu-Mg wrought alloys exhibit high strengths achieved through precipitation hardening. Zhang et al. first reported in detail on the production of Al-Cu-Mg alloys by the SLM technique [117]. For this, gas atomized Al-Cu-Mg alloy powders were used. The chemical composition (wt.%) was 4.24 Cu, 1.97 Mg, 0.56 Mn, with the balance aluminum. With optimized SLM process parameters (like laser power, laser scan speed, hatch distance, and hatch style), a high relative density of 99.8% was obtained, as shown in Figure 15. The energy density of the incident laser beam influenced the densification behavior significantly. A threshold energy of 340 J/mm^3^ was determined, above which flaws and microcracks were almost either non-existent or insignificant, due to sufficient liquid phase flowability and appropriate filling. However, considerable pores and cracks were found in the SLM samples produced below the threshold energy, as can be seen in Figure 16 [117,118].

A higher solid solubility limit of Cu and Mg in α-Al compared to the conventionally cast samples was found in the SLM samples. This is due to the higher cooling rates upon SLM processing. Due to the high solid solubility of Cu and Mg, the formation of secondary phases was suppressed. A strong Al {200} texture was observed in the SLM samples, similar, to what was found in, SLM Al-12Si and AlSi10Mg alloys. The as-fabricated SLM samples have an in-homogenous microstructure in which the typical laser tracks and fine cellular microstructure can be observed (Figure 17). The diffusion of Cu and Mg is hindered and a supersaturated cellular-dendritic structure was formed. The in-homogenous microstructure disappears after solid solution heat-treatment. After such treatment, the laser tracks and hatch overlaps are no longer visible and the microstructure coarsened. This reveals that heat-treatments can bring the microstructure of the SLM alloys closer to that of the conventionally processed alloy. Figure 18 shows the secondary phase (Al_2_CuMg) distributed in the Al matrix in the SLM sample after H4 treatment.

### 2.8. Al-Si-Based Alloys (4XXX Series)

The 4XXX series of aluminum alloys is predominantly alloyed with silicon. The addition of silicon increases the corrosion and wear resistance of aluminum, lowers the melting point (at near-eutectic composition), increases the fluidity of the aluminum melt, reduces the thermal expansion coefficient and moderately improves the strength [119]. Al-Si cast alloys are extensively used in automotive and aerospace industries because of their lightweight characteristics (i.e., rather low density), high specific strength, low recycling costs, and good corrosion and wear resistance [104,105,106,120,121]. For example, engine parts and heat exchangers used in cars are often produced from Al-Si alloys due to their low coefficient of thermal expansion, high wear resistance and the high strength-to-weight ratio [122,123,124,125,126].

The Al-Si phase diagram features a eutectic reaction at a composition of 12.6 wt.% Si [67]. The eutectic Al-Si phases start nucleating from the liquid under eutectic conditions (12.6 wt.% Si and 848 K) and growth proceeds until the end of solidification. Under equilibrium conditions, a pro-eutectic aluminum phase nucleates and grows in a dendritic fashion in hypo-eutectic Al-Si alloys immediately below the liquidus temperature, while primary angular silicon particles form in hyper-eutectic alloys. Based on such a solidification sequence, hypo-eutectic Al-Si alloys contain a soft and ductile proeutectic aluminum matrix in which a hard and brittle secondary silicon phase is embedded. On the other hand, hypereutectic Al-Si alloys are rather brittle due to a high concentration of brittle phases: (a) coarse, angular pro-eutectic silicon particles as well as (b) a eutectic silicon phase. Under non-equilibrium conditions, i.e., via SLM processing, metastable phases are formed due to the high -heating and cooling rates (10^3^–10^8^ K/s) [127,128,129]. In addition, excessive superheating and undercooling enhance the nucleation rate and suppress grain growth in Al-Si alloy [130]. Therefore, the microstructure, mechanical properties and other properties of SLM Al-Si alloys may be significantly different from those of their cast counterparts. Recent advances in the SLM of Al-Si alloys led to the generation of ultrafine-grained silicon phases in the SLM parts, which is very tedious or almost impossible to achieve by casting [131].

#### 2.8.1. Densification and Defects

##### Pores and Balling

Pores are the most common defects that arise in SLM-processed Al-based alloys and can significantly deteriorate the mechanical properties of the parts. It is vital to acquire high quality (dense) Al-based alloys by avoiding or minimizing defects such as porosity and cracking during additive manufacturing. The origin of porosity in aluminum alloys during SLM can be due to lack of fusion, the formation of keyholes or balling as described before [132]. The pores in selective laser sintered (SLS) parts that formed because of insufficient fusion were well studied by Olakanmi et al. [133]. They observed that the powder layer thickness has a significant effect on the degree of interlayer bonding: An sufficiently thin layer enhances the degree of bonding and, hence, increases the average sintered density of the Al-12Si alloy, as shown in Figure 19. Insufficient laser penetration into thick powder layers, caused by low laser energy inputs, leads to incomplete melting of particles, resulting in pores and a non-conformal end part, formed due to a lack of fusion [133]. Insufficient energy densities often result in large and irregular pore morphologies (Figure 20). On the contrary, a keyhole with a fine and intricate shape/size develops due to the supply of excess energy in a local area. Narrow and deep melt pools form in such areas, where hot vapors are trapped in the form of bubbles before they can escape into the atmosphere through the solidifying material, which results in the formation of keyholes in the SLM parts [132]. The keyhole pores are significantly influenced by the fluid flowability inside the melt pool where the surfaces are affected by the temperature gradient, the liquid/solid and liquid/vapor surface tensions, and recoil pressures. Compared to the pores formed by a lack of fusion, keyhole pores have less detrimental effects on the mechanical properties. Teng et al. reported that the mechanical properties of SLM Ti-6Al-4V parts were mostly unaffected for keyhole fractions below 1% [132].

A keyhole pore formation is schematically shown in Figure 21. Another phenomenon, namely balling, is also often observed in SLM of aluminum alloys. This phenomenon is caused by non-stabilized melt pools induced by certain laser energies [134]. Balling can occur either at too low laser energies (incomplete wetting) or too high laser energies (molten material splashes onto cohesive powder particles), which is also called humping or swelling. Balling pores are typically larger than keyhole pores and thus, they have a stronger impact on the mechanical properties of the parts even for an overall concentration of 1%. Furthermore, balling may significantly affect the melt pool overlaps between scan lines and layers, causing discontinuous fusion. Balling, pores, and laser tracks can lead to high surface roughness of SLM samples. Siddique et al. reported that the average surface roughness of SLM Al-12Si samples can reach 7.98 ± 1.50 µm with a maximum roughness of 53.98 ± 10.07 µm [135].

##### Cracks

Cracks are another type of defects in SLM Al-Si alloys, which may be unavoidable for certain cases in SLM processing. During rapid melting and solidification, residual stresses are introduced into the material, which causes cracks if these stresses are higher than the yield strength of the material.

Two different forms of cracks, namely solidification cracks and liquation cracks are often observed during laser processing. Solidification cracks occur when residual stresses develop because of the large thermal gradient between the melt region and the solidified region, while liquation cracks occur in the partially or incompletely melted zone. One of the main factors responsible for the formation of solidification cracks in Al-1Si alloys is tensile stresses induced by the thermal gradient within the material, i.e., when the thermally-induced tensile stresses exceed the tensile strength or the elongation of the molten metal (in the solid-liquid co-existing state) of the SLM alloy [136]. Al-Si alloys with lower flowability (Al-0~4Si) require a higher energy density to form a homogenous melt pool than that required for Al-Si alloys with higher flowability (Al-7~20Si) in order to minimize cracks and to obtain maximum densification [136]. Since Si in Al increases the fluidity and minimizes the solidification range (near the eutectic point), it is expected to decrease the cracking tendency of Al-Si-based alloys [110], unlike other non-ferrous alloy systems (both with and without Si). Undesirable residual stresses in engineering components lead to premature failure. Residual stresses in SLM parts mainly arise due to thermal gradients caused by rapid melting and solidification. For instance, the presence of residual stresses in Al-12Si SLM parts can be determined by Raman intensity mapping, as shown in Figure 22. The figure shows the Raman spectra for Si in an Al-12Si alloy in the as-fabricated condition and after solution treatment (stress-relieving), in comparison with that of standard Si. The Si line in the SLM Al-12Si alloy has a lower wavenumber (~517.6 cm^−1^ compared to 520.7 cm^−1^ for stress-free Si [80]), indicating that significant residual tensile stresses remained in the Si phase in the SLM Al-12Si alloy. Upon solution treatment, the Raman shift becomes smaller, implying that stress relaxation occurs through solution treatment. A large number of dislocations and boundaries are introduced in the Al-12Si SLM parts due to significant residual stresses and ultrafine grain sizes of the Si particles and the Al matrix. During solution treatment, these dislocations and boundaries act as fast diffusion paths and promote precipitation and growth of the Si particles [130]. Residual stresses can also be reduced by employing anchorless selective laser melting or by preheating the substrate plate in order to facilitate a smaller thermal gradient and thus decrease the residual stresses.

#### 2.8.2. Microstructure

The hatch direction observed in the SLM Al-xSi alloys corresponds to the samples produced with a stripe scanning strategy. The width of these hatches strongly depends on the spot diameter of the laser beam, the processing parameters and the hatch overlaps (between 80 µm and 150 µm) [136], and essentially depict the typical laser tracks created during SLM processing [71,137]. The boundaries or the sites with relatively continuous coarse microstructures correspond to the overlapped regions between two adjacent laser tracks (i.e., hatch overlaps), which are melted twice. For most Al-Si alloys, a cellular microstructure develops in the track cores, which experience a single melting step. The core is rich in Al and the cellular boundaries contain Si precipitates. The cellular structure in the SLM Al-12Si alloy has a size of about 500–1000 nm, with a cellular boundary thickness of about 200 nm, as shown in Figure 23 [67].

The cellular structure is a typical structure in many SLM-fabricated alloys such as Al-Si alloys, CoCr based alloys, 316L steels and even Cu based alloys [138,139,140,141,142,143]. Yang et al. reported their results on Cu-12.5Zn-2.9Si silicon brass alloy, in which Si is helpful to form the cellular structure by precipitation of the κ-Cu7Si phase [144]. Such a fine microstructure is ascribed to the high cooling rate during SLM and will be explained in detail below [129,145,146]. The microstructure of SLM Al-Si alloys changes significantly with the Si content. Si can exist in solid solution, as Si precipitates, or within the cellular structure as discussed above. The solubility of Si in Al is 1.6 wt.% under equilibrium conditions at room temperature according to the Al-Si binary phase diagram, while it increases up to 10 wt.% in the eutectic composition for melt-spun (rapidly solidified) Al matrix [147,148,149]. Kimura et al. observed solubility of ~7 wt.% Si in Al in SLM Al-Si alloys [136]. As the solidification progresses rapidly, the excess solute concentration in the solidifying front is rejected into the surrounding melt pool. Hence, the solute concentration in the liquid rises as the solidification front moves [67]. With increasing Si content, fine dendritic structures with cell sizes of ~0.5 µm form in the Al-Si SLM parts. The cell boundaries are lined with Si precipitates, which become much thicker when the Si content increases beyond 12 wt.%, due to nucleation of primary Si precipitates along the boundaries [136]. The Al-12Si SLM alloy is the most rigorously studied Al-Si alloy because of its excellent castability at near-eutectic composition. A cellular structure is preferably formed in Al-12Si SLM samples, which results from the fast solidification during the SLM process. Due to the high cooling rate, α-Al solidifies first in a cellular morphology with an extended solubility of Si. The residual Si then segregates along the cellular boundaries, as observed in Figure 23d–f [67].

Li et al. proposed that the fine cellular structure comes from the inhomogeneous microstructure of the molten pool of Al-12Si, which may be mostly retained after rapid solidification. According to their analysis, the center of the melt pool can reach a maximum temperature of about 1712.15 K and a large portion of the melt pool probably undergoes superheating. This facilitates the formation of an inhomogeneous microstructure within the melt pool. Additionally, liquid oscillations and the short interaction time between the laser and the material enhance the formation of the inhomogeneous microstructure. In Al-Si SLM alloys, such inhomogeneous microstructure corresponds to nano-sized Si-rich and Al-rich regions in the melt pool, with a size below 100 nm, which constitutes heterogeneous nucleation and enhancement of the nucleation rate (Figure 24). The super-high cooling rate helps to retain such an inhomogeneous microstructure and restrain cellular growth [130]. Prashanth et al. reported that the cells in the cellular morphology are rich in Al and have a size of about 500–1000 nm, and the cellular boundaries where Si is preferentially located have a thickness of about 200 nm [67].

The XRD pattern of the Al-12Si SLM alloy exhibits diffraction peaks corresponding to Al and Si, as shown in Figure 25. The XRD pattern reveals that a strong texture is observed in the alloy based on the XRD pattern, where the intensities of the Al (111) and (200) peaks are reversed with respect to the same material produced by casting. Prashanth et al. corroborated the extended solid solubility of Si in α-Al in the SLM alloy based on the weak intensities of the Si peaks, which is a result of the reduced amount of ‘free’ Si in the alloy. The Si peaks are broader for the SLM alloy compared to its cast counterpart, indicating a reduced size of the Si phase in the former. Rietveld peak fitting analysis [118] indicates that the lattice parameter of Al is 4.0508 Å and the amount of free residual Si is ~1 wt.% in the SLM alloy, while it is ~10 wt.% in the conventional cast Al-12Si alloy. This confirms the formation of a supersaturated solid solution of Si in Al in the SLM alloy, which is in accordance with other processes involving high cooling rates [67,130,150,151]. The crystallite sizes of Al and Si were determined as 118 and 8 nm, respectively [67].

Hyper-eutectic Al-Si alloys, where the Si content is higher than 12.6 wt.% have also been produced by SLM. Compared to Al-12Si or hypo-eutectic Al-Si alloys, these alloys show a significantly different microstructure owing to the presence of pro-eutectic Si phase, as opposed to pro-eutectic Al in the hypoeutectic alloys. Kang et al. [152] studied the Al-18Si alloy by mixing Al-12Si and pure Si powders with a weight proportion of Al-12Si:Si = 92.6:7.4. Nano- and micron-sized Si particles were observed, and the amount of the latter decreased with increasing laser power. At a high laser power of 225 W, the irregular-shaped Si particles turned completely spherical, indicating the complete melting of the element [153]. The Al-20Si SLM alloy was studied by Kimura et al. [136] and Ma et al. [154] and another Al-Si alloy with Si content as high as 50 wt.% was studied by Jia et al. and Kang et al. [90,152]. The difference between the microstructures of the SLM- and as-cast Al-50Si alloys is depicted in Figure 26 [90]. In the hypereutectic compositions, the cast sample exhibits a plate-like primary Si phase (average length of 220 ± 5.2 µm) surrounded by the eutectic. The eutectic Si phase exhibits a needle-like shape with an average length of 2.3 ± 0.5 µm with the largest size of 10.0 µm.

On the other hand, the Si phase in hypereutectic SLM Al-Si alloys shows a significantly different morphology compared with their cast counterparts (with respect to both the primary phase and the eutectic). Owing to the overlapping of adjacent laser tracks, the microstructure of the SLM samples is comprised of two different areas; one correspondings to the hatch overlap area, and the other one to the hatch cores. Both hatch overlap areas and hatch cores show a microstructure with a primary Si phase and a phase containing the eutectic mixture of Si and Al matrix. However, primary Si in the hatch overlap areas has a mean grain size of 5.5 ± 0.3 µm, which is larger than that in the hatch cores (3.6 ± 0.2 µm). The microstructure of the Al-50Si SLM alloy shown in Figure 27 suggests that primary Si undergoes macro-segregation [152]. A fine primary silicon phase is found in the contour region (Figure 27a), with a mean particle size of 2.6 µm (Figure 27d). In the center region, i.e., the core or volume of the component (Figure 27c,f)), the average size of the primary silicon phase is about 6 µm. In the middle region, i.e., the contour offset, which is in between the contour and center regions, large irregular shaped primary silicon particles (8.5 µm) are visible (Figure 27b,e), which are surrounded by the eutectic structure. It is proposed that an internal flow of the fluid occurs within the melt pool [155,156] owing to the temperature gradient existing between the internal and external melt pool during SLM. Figure 28 shows a schematic illustration of the flow within the molten material and macro-segregation of the primary silicon phase, as proposed by Kang et al. [152].

Figure 28a depicts the scanning mode of the laser beam on the alloy powder. For hypereutectic Al-Si binary alloys, primary Si first nucleates from the melt pool in the low-temperature region during solidification. Consequently, the internal region is relatively Si-deficient (Figure 28d). The size of the primary silicon grains formed at the external region is smaller in the internal region of the solidified pool, due to faster cooling in the external region, as shown in Figure 28e. Therefore, a high degree of segregation of the primary silicon phase may occur when the laser power is too high, as it may lead to a large melt pool (Figure 28f). Accordingly, the alloy obtained with low laser power contains a mostly homogeneous distribution of primary silicon, whereas the alloy obtained at a high laser power shows the separation of the primary silicon phase along the entire cross-section.

The effect of heat treatment on the microstructure and mechanical properties of Al-12Si SLM alloys was studied in detail by Prashanth et al. [67,157] and Li et al. [130]. They have shown that irrespective of the changes in annealing time or temperature, the microstructure of the material transforms from a cellular kind of microstructure to a composite type of microstructure, where the supersaturated Si particles are ejected from the Al lattice as shown in Figure 29 and Figure 30. The intensity of the Si peaks continues to increase and the broadening of the Al and Si peaks continues to decrease due to the relaxation of internal strain and growth of the grains simultaneously. In addition, the texture observed in the as-prepared SLM material is partially/completely reversed depending on the heat treatment time and/or temperature. It is interesting to note from the work of Prashanth et al. [67], that the Si rejected from the Al matrix is diffused and preferentially deposited at the hatch overlaps. It has been observed that as much as twice the amount of Si particles are observed at the hatch overlaps compared to the core of the hatches. In addition, the size of the Si particles at the hatch overlaps is larger than the Si particles observed at other places in the same sample. A schematic transformation of the as-produced SLM microstructure (cellular microstructure) to composite like microstructure is shown in Figure 30.

The effect of different hatch styles on the texture and phase distribution of Al-12Si SLM alloys was studied by Prashanth et al. [125]. Texture coefficient is a measure of texture; it is defined as:
(2)$$T_{c}\left(hkl\right)=\frac{I\left(\mathrm{hkl}\right)/I_{0}\left(\mathrm{hkl}\right)}{\left(1/N\right)\sum_{\mathrm{n1}}^{\mathrm{n2}}I\left(\mathrm{hkl}\right)/I_{0}\left(\mathrm{hkl}\right)]}$$ where *T_c_(hkl)* is the texture coefficient of the (hkl) plane, *I* is the measured intensity, *I_0_* is the standard intensity and *N* is the number of diffraction peaks. *T_c_* is close to unity for a randomly distributed powder sample, while it changes from unity for preferentially oriented (hkl) planes. The XRD patterns of the Al-12Si samples produced with different hatch styles are shown in Figure 31a and the variation in the texture coefficients of the first two intense peaks of Al ({111} and {200} peaks) are summarized in Figure 31b. The crystallite size and the lattice parameter of Al are 110 ± 3 nm and 0.40509 ± 0.00002 nm, respectively, and the amount of free residual Si is 1.35 ± 0.03 wt.% for all the SLM Al-12Si samples regardless of the hatch style used. Apart from the variation in texture, no distinct structural changes have been observed for the SLM Al-12Si samples fabricated with different hatch styles [157].

### 2.9. Al-Mg-Si-Based Alloys (6XXX Series)

#### 2.9.1. Densification and Defects

The 6XXX series of aluminum is generally alloyed with magnesium and silicon [158]. The combination of elements present in the alloy allows thermal treatment/aging, which improves the strength of the alloy considerably [158,159,160]. The 6XXX alloys are the most widely used castable and weldable aluminum alloys exhibiting good corrosion resistance. One of the prominent alloys in the 6XXX family is AlSi10Mg, which is widely used for automobile components. AlSi10Mg can bear considerable mechanical loads and is regarded as an alternative to titanium in case of lightweight parts (when they are not exposed to excessive fatigue). AlSi10Mg can be subjected to various post-processing operations like machining, spark erosion, welding, coating, etc. Moreover, AlSi10Mg alloys have good melt fluidity and low shrinkage, which is favourable for casting. Since AlSi10Mg has good castability and weldability, it is favored for selective laser melting. Hence, AlSi10Mg is presumably one of the most widely used Al-alloys for additive manufacturing, especially for SLM processing. Besides the AlSi10Mg alloy, a few compositions belonging to the 6XXX family have also been studied by using the SLM process, e.g., the AlSi7Mg alloy [161]. Since AlSi10Mg is one of the prominent alloys for SLM, its optimum process parameters are readily available today and are qualified for producing industrial components. However, several groups have tried to vary the process parameters to study their influence on defect formation and microstructure development (Figure 32). The influence of laser scan speed [76], hatch spacing [76], scan strategy (scan orientation) [76], layer thickness [76], single-laser or multi-laser melting [162,163], melt pool boundary condition [164], amount of defocusing [165], and energy density (the combination of laser power and laser scan speed) [166,167] were studied along these lines. For example, Aboulkhair et al. varied the laser scan speed between 250 and 1000 mm/s and observed significant changes in the microstructure [76].

A slow laser scan speed leads to the formation of numerous spherical-shaped metallurgical pores. When the laser scan speed is increased gradually, a transition from metallurgical to keyhole pores occurs. This transition is observed at around 500 mm/s. For faster scanning keyhole porosity dominates (Figure 33). Even though two sets of samples may have a similar amount of porosity, the type of porosity has to be considered in order to make suitable changes of the process parameters [76] and where the keyhole pores entirely depend on the process parameters. Similarly, at high laser scan speeds, the melt pool becomes capillary unstable, which promotes the splashing of the liquid leading to balling phenomena (due to non-linear solidification of the metal) [168].

The balling of SLM parts, along with keyhole pores, becomes more pronounced with increasing laser scan speed. Unlike keyhole pores, the balling phenomenon can be easily observed from the top surface of samples [76]. In the presence of excessive balling, irregular surfaces are observed on SLM parts. It has been suggested that the double scan strategy (even though it takes considerable time for fabricating the parts) can help to eliminate the keyhole pores. However, the excessive energy supplied to the powder bed leads to hydrogen pickup and, hence, the formation of metallurgical pores in the AlSi10Mg alloys [78,88]. Weingarten et al. observed that hydrogen pickup is not the only reason that may lead to hydrogen/metallurgical pores [88]. The powder particles can have moisture on their surface that acts as the dominating hydrogen source, but can be reduced either by drying the powder particles isothermally in a furnace (external process) or by drying of the powders internally before melting them using a low power laser source [88]. The growth of the hydrogen pores occurs when the amount of hydrogen in the melt exceeds the maximum solubility limit of hydrogen in molten aluminium [88] and it is a diffusive process. The hydrogen pore density as a function of laser scan speed is shown in Figure 34.

Figure 34 reveals that both increase of scanning speed and the use of undried powder increases the hydrogen pore density (or, in other words, the porosity level) in AlSi10Mg samples. Read et al. performed an extensive process optimization using the statistical response surface methodology [166]. The response surface model predicting the porosity with respect to the laser parameters (laser power and laser scan speed) is shown in Figure 35. This plot suggests that both decreasing the laser power and increasing the laser scan speed are detrimental as they tend to increase the porosity level, similar to the reports of Aboulkhair et al. [76,169]. The laser power has a more significant influence on the formation of porosity than the laser scan speed. Low energy densities (low laser power and high scan speeds) result in reduced melt pool widths and, hence, porosity formation due to incomplete consolidation of the powder particles [166]. Considering all these aspects, a suitable parameter set was selected for AlSi10Mg alloys with a laser power of 200 W and a laser scan speed of 1350 mm/s [166].

The above discussed laser parameters are not the only parameters that determine the quality of the SLM parts. Depending on the type of laser, the laser focus diameter and the maximum power of the laser, both the laser power and the laser speed have to be adjusted to maintain minimum porosity levels in the material. Read et al. also showed that there is a critical energy density that results in a minimum pore fraction possible in AlSi10Mg alloys, which is ~60 J/m^3^ [166]. However, this is not an essential criterion, but only serves as a guideline. Before analyzing the effect of the hatch distance, it is very important to have the best possible parameters for laser single tracks and layers without defects like porosity. Aboulkhair et al. investigated the laser single tracks and layers using the AlSi10Mg powders [167]. The region with a stable melt pool was chosen for further parameter optimization.

It is obvious that the dimensions of the laser single track decrease with increasing laser scan speed and vice versa. Another reason for investigating laser single tracks is that formation of defects can be easily observed and can give hints for suitable hatch spacing, depending on the behavior of the material [170]. For instance, some of the melt pools have a conical instead of a cylindrical shape (Figure 36), which suggests the possible formation of keyhole pores in the part (suggesting relatively fast laser scan speed) [167]. Aboulkhair et al. also proposed that the results or the trend observed from laser single track cannot be taken as the only criterion to calculate the process parameters [167]; it may, however, be used as a guide for working with the process parameters. No traces of porosity were observed in the AlSi10Mg laser single tracks shown in Figure 36. Similarly, the possible selection of the laser scan speed is illustrated in Figure 37. It may be observed that the layers are too thick (400 µm) for the parameter selected. All three laser scan speeds show the presence of satellites and excessive balling. Moreover, the laser single tracks produced with a laser scan speed of 250 mm/s do not show many discontinuities. When the laser scan speed increases to 500 mm/s and 750 mm/s the volume of discontinuities increases and a complete disconnection within the tracks is observed. Some areas just show a series of droplets, suggesting that only minimal energy is conducted by the melt pool to the substrate, thereby reducing the remelted depth resulting in a lack of bonding between the substrate and the single laser track [167]. Aboulkhair et al. were able to show experimentally that pores do not form in single tracks or in layers using a predefined parameter set, but they may form in multi-layered samples [167].

Aboulkhair et al. also investigated the importance of the hatch spacing for AlSi10Mg SLM samples (Figure 38) [76]. They showed that sufficient overlaps between the hatches are observed when the hatch spacing is less than 150 µm. When hatch spacing increases to 200 µm or more, gaps/lack of overlaps between adjacent hatches is observed. It may be postulated that larger hatch spacings may be used to accelerate the fabrication process when a small layer thickness is employed. An effective hatch spacing distance improves not only the overlap between the adjacent hatches but also the intra-layer overlap depending on the shape of the laser beam. On the other hand, a smaller layer thickness increased the fabrication time [137]. However, smaller hatch spacings are preferred, because sufficient heat is accumulated in the melt pool, thereby reducing the cooling rate [171]. A reduced cooling rate allows the formation of a continuous and homogeneous layer.

#### 2.9.2. Microstructure

AlSi10Mg SLM samples show a unique microstructure consisting of a cellular-dendritic morphology with α-Al forming the core of the cells and Si decorating the cell boundaries (Figure 39) [160,169,172,173,174]. The microstructure of AlSi10Mg SLM samples is very similar to that of Al-12Si SLM specimens [67,130]. Li et al. demonstrated that a microstructure with cellular morphologies (but three different sizes) was observed across the melt track can form due to differences in thermal history [130,160]. The different cellular morphologies are a coarse cellular zone, a transition zone, and a fine cellular zone. The coarse cellular zone and the transition zone correspond to the boundary of the laser melt track or the track overlap, where the area is melted twice. Li et al. also showed that α-Al, Si, and Mg_2_Si phases form in the AlSi10Mg SLM samples, as determined from XRD patterns [160]. However, Wu et al. [172] conducted a detailed microstructural analysis of AlSi10Mg SLM samples and interpreted the microstructure in a different way. They showed the presence of columnar grains from EBSD patterns (Figure 40). These columnar grains may extend up to several hundreds of microns in length, along with some equiaxed grains along the YZ plane. The columnar grains are considered to encompass typical cell-like substructures that are about 500 nm in size and the boundaries of the cells are rich in Si [71,169,175]. Unfortunately, it is difficult to highlight the grain boundaries in these SLM samples from the secondary and columnar images. The Al-Si eutectic is present at sub-cell boundaries, cell boundaries, and in the grain boundaries [172]. Its volume fraction is ~35 vol.% in all three boundaries. The observations made by Wu et al. [172] on the XY plane are identical to other reports [71,160] where a typical cellular-dendritic microstructure with the presence of Al along the cells was observed, which are partitioned by boundaries that are rich in Si. This also leads to the conclusion that Al is the first phase to solidify by pushing Si towards the boundaries because of the solubility factors, similar to what was found for yhe solidification of Al-12Si SLM samples [67,172].

Detailed TEM studies were carried out on the AlSi10Mg SLM samples by Wu et al. [172]. The TEM images along the YZ plane show characteristic long Al cells with identical orientation between the adjacent cells (Figure 41). Along the length of the cells are sub-cell boundaries that are rich in Si. The presence of sub-cells and the decoration of Al in the sub-cell boundaries are shown in the dark-field image taken with the 200 reflections of Al (Figure 41b). Micro-diffraction patterns help to understand the orientations of the sub-cells and the sub-cell boundaries. The sub-cells and the sub-cell boundaries (rich in Al) have identical orientations that are consistent along the entire length of the cells despite the presence of the substructure within each long cell. The orientation of the cells changes at the grain boundaries. However, these data are hard to interpret from the SEM images and from the bright-/dark-field TEM images [172]. The diffraction patterns suggest epitaxial growth in the Al phase, which possesses the same orientation as Al along with the cell below and Al that solidifies above the boundary. The width of the individual cells is ~500 nm (which is similar to the finding for Al-12Si SLM samples [67]) and the Si particles in the boundaries have a random orientation. Thijs et al. showed the presence of both morphological and crystallographic texture in AlSi10Mg SLM samples [71]. They have investigated options to modify the texture in the SLM material by varying the scan strategy. The scan strategy is assumed to affect the texture in SLM samples because of the presence of an elongated melt pool (from a moving laser source) and, thus, directional solidification can be achieved [71]. Thijs et al. [71] have employed five different scan strategies as shown in Figure 11 and the resulting electron back-scattered diffraction images of the samples are shown in Figure 42.

The presence of a fiber texture was ascertained from the low intensities in the remaining pole figures; however, it was not complete. The overall texture index was calculated to be 1.974 as opposed to unity in isotropic materials. The normalized texture difference between unidirectional scanned samples and the samples with the bidirectional scan is only ~0.64%. Hence, the different scan strategies do not affect the crystallographic texture in the AlSi10Mg samples (Figure 42). However, the overall texture is significantly reduced (~35%) by rotating the scanning direction by 90° for every laser exposure (the texture is lowered from 1.974 for samples without scan rotation to 1.266 for samples with 90° scan rotation). The texture was also reduced by varying the scan directions within a layer by employing the island scanning strategy. A 90° scan rotation between the neighboring islands results in a texture index of ~1.127, leading to more isotropic AlSi10Mg samples. It was also established that the texture in the SLM material does not depend on the crystallography of the substrate plate but on changes in the local heat fluxes. Since AlSi10Mg is an age-hardenable alloy, both annealing and age hardening treatments have attracted considerable interest. Li et al. [160] explained the effect of the annealing treatment in detail and showed that AlSi10Mg samples behave in a similar way as Al-12Si specimens [67,130]. The fine cellular-dendritic structure transforms into a composite-like microstructure with Si particles dispersed in the Al matrix (Figure 43) [160]. The microstructure becomes coarse (i.e., size of the Si particles increases) with increasing annealing time and/or temperature, and the observed features are very similar to the annealing treatment of Al-12Si samples (see Figure 44). Interestingly, the samples were not kept for longer time (6 h) at higher temperatures like in the case of the Al-12Si annealing treatment; instead, the samples subjected to the solutionizing treatment were held only for 2 h. Nevertheless, artificial aging was carried out for longer times (12 h) [67,130,160]. Similar investigations of the Si particle size and density were made by Aboulkhair et al. [176]. However, no similarities were observed between the results of Aboulkhair et al. [176] and Li et al. [160].

The disparity in both size and spatial density of Si particles in the SLM-fabricated AlSi10Mg samples after solutionizing and artificial aging treatments are due to the following differences: (a) Aboulkhair et al. [176] used a Renishaw AM 250 SLM device, which can reach a maximum laser power of 200 W. In contrast, Li et al. [160] used an SLM solutions SLM 250 HL device to prepare the AlSi10Mg samples with a laser power of 350 W and a laser scan speed of 1140 mm/s. This laser scan speed is significantly higher than the laser scan speed used by Aboulkhair et al., which was 318 mm/s [176]. (b) Apart from the differences in the applied laser power and scan speed, both groups used different hatch distances, layer thicknesses, and so on. Hence, an apparent difference in the microstructure is observed, which leads to differences in the sizes and spatial density of Si particles after solutionizing and artificial aging treatments. This is especially true with respect to the length scales, while the compositions of the phases present in the microstructure remain the same. Overall, general comments can be made on the heat treatment of AlSi10Mg SLM specimens, compared to cast samples [177]. As-fabricated AlSi10Mg SLM samples show a very fine microstructure with a continuous Si phase along the cell boundaries of α-Al similar to the Al-12Si samples. On the other hand, cast AlSi10Mg samples exhibit a coarse dendritic microstructure with non-uniform distribution of Si in the form of flakes [178,179,180,181]. Precipitation hardening of the cast sample leads to the formation of fine Si particles and clustering at the grain boundaries of α-Al along with the formation of Mg_2_Si precipitates, but the SLM samples behave differently. The microstructure coarsens and becomes a solid solution of Si particles dispersed in the α-Al matrix, appearing like a composite.

### 2.10. Al-Zn-Mg-Cu-Based Alloys (7XXX Series)

7XXX alloys (Al-Zn-Mg-Cu) have lower densities than Al-Cu based alloys, combined with high strength, fracture toughness and resistance to stress corrosion cracking (SCC) and exfoliation corrosion, making them attractive candidates for lightweight aerospace applications [111]. However, additive manufacturing of the 7XXX alloys is a major challenge, firstly because of the evaporation of the low boiling point alloying elements such as Mg and Zn, resulting in undesired fluctuations in the alloy composition. The Zn:Mg ratio has a significant effect on the microstructure, for example, a T phase ((Al,Zn)_49_Mg_32_) may form for low Zn:Mg ratios in 7075 alloy [111]. The evaporation also causes porosity and voids in the deposit. Secondly, hot tearing cracks or solidification cracks can form easily. Thirdly, the poor laser absorption, as well as the high thermal conductivity of the alloy, leads to rapid heat dissipation into the solidified part or powder bed. Therefore, finding the right parameters and solutions for the additive manufacturing of 7XXX alloys has become increasingly important and in the following sections, we summarize the most recent efforts made in this direction.

#### 2.10.1. Densification and Defects

SLM Al 7XXX alloys often suffer from inferior strength compared to their cast counterparts. Due to the huge difference in the evaporation temperatures of zinc (1180.15 K) and aluminum (2792.15 K), zinc easily vaporizes during the SLM process and at least 1.6 wt.% loss is incurred on average [182]. The loss of Mg is usually lower than that of Zn. Wang et al., reported a loss of Zn and Mg in the order of ~2.81 wt.% and ~0.39 wt.%, respectively, in one Al-Zn-Mg-Cu alloy [183]. Similarly, a loss of ~3.46 wt.% Zn and ~0.8 wt.% Mg was reported in another study [95]. The loss of Zn and Mg was shown as one of the reasons for the deterioration of the mechanical properties of SLM 7XXX alloys [95]. Qi et al. reported the effects of scanning speed and defocusing distance on the melting mode transition for the SLM Al7050 alloy [184]. They showed that the melting mode undergoes a transition from the keyhole to conduction mode as the scanning speed increases from 100 to 1200 mm/s, with the other process parameters, kept constant. Compared to the conduction mode, the keyhole mode shows a deeper and narrower melt pool as the laser intensity is sufficiently high to cause metal vaporization. Crack propagation, as seen in Figure 45, occurs along the grain boundaries for both modes. However, for the conduction mode, the grains are predominantly columnar along the building direction and the cracks appear parallel to the building direction. In contrast, the keyhole mode comprises much smaller and randomly oriented grains, resulting in cracks propagating in different directions [185]. Kaufmann et al. studied in detail the influence of process parameters on the quality of an SLM EN AW7075 alloy. A relative density of over 99% could be achieved by carefully controlling the process parameters. They showed that exceptionally high relative densities might be obtained using high laser power and low scanning speeds. The variation of porosity and relative density with respect to the laser process parameters (power and scan speed) are shown in Figure 46 [182].

In order to improve the weldability or reduce hot cracking, several investigations were devoted to modifying the composition of 7XXX alloys by mixing nanoparticles of other elements to the micron sized 7XXX alloy powder [95,185]. For example, Sistiaga et al. reported a Si-added Al7075 alloy produced by SLM [95]. The Al7075 powder was mixed with 1 to 4 wt.% Si nanoparticles. Si is expected to improve the fluidity of the melt and reduce thermal expansion by decreasing the melting temperature and solidification range through the formation of a low-temperature eutectic [110]. It was also found that cracks were present only in samples with a Si content lower than 2 wt.%, but could not be observed in samples with higher Si contents. In the alloys with low Si content, long (100–300 µm) and wide (50–100 µm) grains were observed, which were parallel to the building direction, whilst for alloys with higher Si content, the Si nanoparticles induced grain refinement, as shown in Figure 47 [95]. The formation and propagation of cracks were also suppressed by adding Si. Grain refinement, as well as the fact that Si can reduce the melting temperature and form a second eutectic that can backfill the cracks during solidification, were suggested as the reasons for the absence of cracks in such alloys.

#### 2.10.2. Microstructure

Recently, high strength Al7075 alloys were successfully processed by SLM, and crack-free, equiaxed, fine-grained microstructures were achieved [185]. For instance, a general approach promoting the nucleation of new grains was reported by adding 1 vol.% hydrogen-stabilized zirconium nanoscale nucleants onto the surface of the pre-alloyed gas-atomized Al7075 spherical powder (Figure 48). Through this, solidification preferentially occurs through nucleation on existing grains, leading to grain growth vertical to the building direction with grains extending across multiple build layers, as depicted in the inverse pole figure map in Figure 48e. The addition of nanoparticles, which form Al_3_Zr during melting, provides a low-energy heterogeneous nucleation site. A fine equiaxed structure is formed instead of a columnar structure (Figure 48f), which allows easier grain rotation and deformation, providing a means to accommodate strain in the semisolid state, thus preventing crack initiation and growth during SLM.

Singh et al. reported a nickel-coated Al7050 alloy produced by laser metal deposition, a different additive manufacturing technique [186]. The Al7050 powder was coated with Ni to increase the absorption of photons from the laser in order to achieve a good quality of the deposit. In addition, the coating of Ni can reduce the evaporation of low melting point elements such as Zn and Mg. It was shown that the columnar dendrites above the bead/layer boundaries gradually transformed into equiaxed α-Al dendrites in the middle to the upper part of each bead/layer. However, Ni largely segregated in the inter-dendritic boundaries and produced brittle intermetallic phases, leading to poor tensile ductility of the as-deposited samples. A new phase or structure may form in the SLM-fabricated Al7075 alloy compared with conventional cast material. For example, an icosahedral quasicrystalline phase was observed in the SLM-fabricated AA7075 aluminum alloy [187].

Heat treatments such as solid solutionizing, quenching and aging are vital for Al-Zn-Mg-Cu alloys for modifying the microstructure and improving the mechanical properties. Considering the Al-Zn-Mg system, the following phases form during aging: GP zones (MgZn); η’; η (MgZn_2_); T’; and T (Al_2_Mg_3_Zn_3_ or (AlZn)_49_Mg_32_) [111]. It was shown that GP zones and the semicoherent η′ phase lead to maximum strengthening. Compared to the traditional 7XXX series of aluminum, samples produced by additive manufacturing techniques show a vast difference in both microstructure and phases, which can result in a different behavior during heat treatment. For example, the finer grain size and hence larger volume fraction of grain boundaries may influence the solid solution and aging behavior. Moreover, the evaporation of low melting point elements may change the precipitation of phases, since the Mg:Zn ratio has a significant effect on the phases formed during aging. Wang et al. studied the effect of heat treatment on the microstructure and mechanical properties of an Al-Zn-Mg-Cu alloy produced by casting and SLM [183]. Figure 49a1,a2 shows the microstructure of the cast sample, which consists of equiaxed Al grains (dark phase) along with a lamellar eutectic that precipitated around the grain boundaries. After T6 heat treatment of the cast sample, the amount of the eutectic decreases and appears with a discontinuous particle-like morphology (Figure 49b1,b2). On the other hand, in the as-fabricated SLM alloys, the distinct lamellar morphology of the h phase is absent (Figure 49c1,c2). Fine particles (η phase) are found in the interdendritic areas, maintaining a high degree of supersaturation, which can subsequently result in the precipitation of the η′ phase during T6 heat treatment [183,188,189,190]. The microstructure further changes in the T6 heat-treated SLM samples, where the η particles are hardly visible within the Al matrix and are mostly situated at the Al grain boundaries (Figure 49d1,d2). However, the precipitation behavior in the matrix was not shown in this work. Heat treatment was also applied to improve the mechanical properties of Al-Zn-Mg-Cu alloys produced by SLM in a few other studies [189]. Figure 50 shows the hardness of SLM Al7075 as a function of aging time, in which the alloy aged for 18 h displays the highest hardness [185]. However, much work still needs to be carried out fully understanding the heat treatment behavior and its influence on the mechanical properties and other physical propertiest.

## 3. Mechanical Properties

### 3.1. Pure Aluminum

SLM-processed pure aluminum shows excellent mechanical properties compared with commercial-purity aluminum wrought materials [112] (Table 1). SLM processed pure aluminum exhibits a tensile strength of ~110 MPa and a proof stress of ~90 MPa, which are significantly higher than that of the wrought counterpart, combined with a fracture elongation of ~30%, which is lower than the deformability of A1060-O. However, compared to the A1060-H14 wrought material, SLM pure aluminum shows almost equal strength with about three time’s higher elongation. It was reported that annealing treatment, for instance, annealing at 723 K for 10 min, has a negligible effect on the mechanical properties such as strength and plasticity. The high strength obtained in the as-fabricated SLM pure aluminum stems mainly from the characteristic granular microstructure with uniformly distributed fine oxides acting as a reinforcing phase in the aluminum matrix, giving a dispersion strengthening effect. In addition, the high density of dislocations existing in the as-fabricated SLM samples, owing to the high residual strains, help to further hinder dislocation movement, leading to dislocation–dislocation interaction strengthening [112].

### 3.2. Al-Cu-Based Alloys

SLM-processed Al-xCu binary alloys demonstrate superior mechanical properties owing to a fine microstructure and the presence of a nano-scale eutectic including a high volume fraction of hard Al_2_Cu intermetallic phase. Al-33Cu and Al-40Cu binary alloys show compressive strengths higher than 1 GPa, but with very limited or no plasticity, as seen in Figure 51 [115]. The tensile properties of SLM Al-Cu-Mg alloys are summarized in Figure 52.

The as-fabricated Al-Cu-Mg alloy shows tensile and yield strengths of 402 and 276 MPa, respectively, with an elongation of 6%. Suitable heat treatments can improve both the strength and ductility: the tensile strength, yield strength, and elongation increase approximately by factors of 1.3, 1.2 and 2.3, respectively. The cooling rate (water quenching or air-cooling), subsequent to heat treatment, also influences the mechanical properties [117]. Zhang et al. [190] studied the effect of Zr addition on crack, microstructure and mechanical behavior of SLM Al-Cu-Mg alloy. The Crack formation was prevented with the addition of Zr due to the formation of Al_3_Zr precipitates that act as nuclei for α-Al grains during solidification, leading to grain refinement. Figure 53a shows large columnar grains in the SLM Al-Cu-Mg alloy, while the grains become ultrafine or nanostructured and equiaxed with the addition of Zr (Figure 53b,c). The boundary misorientation distribution shifts to a higher angle with the addition of Zr (Figure 53d,e). Moreover, the ultimate tensile strength and yield strength improve significantly from 393 ± 20 MPa and 253 ± 9.8 MPa to 451 ± 3.6 MPa and 446 ± 4.3 MPa, respectively, with the addition of Zr. However, the elongation decreases from 6 ± 1.6% to 2.67 ± 1.1%. Figure 54 illustrates the corresponding fracture surfaces of SLM processed Al-Cu-Mg and Zr/Al-Cu-Mg specimens. Refined shallow dimples with a size of 0.4–0.9 µm are found in the Zr/Al-Cu-Mg part, indicating a relatively brittle fracture mode [190,191,192]. The porosity in the SLM parts can also result in lower ductility by accelerating crack propagation.

More recently, heat-treatable Al-Cu-Mg based alloys were studied by Wang et al., [191]wherein the effect of T6 heat treatment on the SLM Al-3.5Cu-1.5Mg-1Si alloy was investigated. They found that heat treatment results in a slight increase in the grain size and the Q phase (a phase with unknown space group formed in the as-produced SLM specimen) transforming into Al_2_Cu(Mg), Mg_2_Si, and Al_x_Mn_y_ phases. Both strength and elongation improved after heat treatment: both, the Yield Strength (YS) and the ultimate tensile strength (UTS) increased from 223 ± 4 MPa and 366 ± 7 MPa in the as-prepared alloy to 368 ± 6 MPa and 455 ± 10 MPa after heat treatment, respectively. The elongation increased from 5.3 ± 0.3% to 6.2 ± 1.8% post-treatment as indicated in Figure 55 [191]. Ahuja et al. [192] reported the production of an EN AW-2618 alloy by laser beam melting (LBM), in which Ni and Fe were added to the Al-Mg-Cu-based alloy. By carefully choosing the process parameters, dense samples in the shape of a single line, thin wall, and cube were obtained by means of LBM. It was proved that the volumetric energy density is only a rough indicator for high relative density parts with low predictive accuracy. An average density of 99.97% was observed for the EN AW-2618 alloy. The aging behavior of the SLM 2618 alloy was studied by Casati et al. [193]. Besides Al-Cu-Mg alloys, an AlCu_6_Mn alloy was also studied by Ahuja et al. [192].

### 3.3. Al-Si Based Alloys

A detailed analysis of the mechanical behavior of Al-xSi SLM alloys, conducted by Kimura et al. [136] are presented in Figure 56 and Figure 57. The tensile strength and the proof stress of the alloys increase with increasing Si content, with an exception for the Al-1Si alloy. For example, the tensile strength of the Al-20Si SLM alloy is about 575 MPa, which is significantly higher (~5 times) than that of SLM pure Al. However, the fracture elongation drops strongly for higher Si contents, i.e., from ~30% for an Al-0Si SLM sample to less than 5% for an Al-20Si SLM sample [136]. Table 2 provides a summary of the tensile properties of Al-xSi alloys produced by SLM processing with/without ensuing heat treatments.

Beyond 12 wt.% Si, the SLM Al-xSi samples become more brittle and so it is not possible to test under tensile loading. Among the various Al-xSi alloys, Al-12Si is the most studied alloy owing to its excellent castability. Table 3 provides a summary of the tensile properties of Al-12Si alloys produced by different processing methods such as casting/SLM under varying atmospheres, with/without ensuing heat treatments. Figure 58 summarizes the mechanical properties of various Al-12Si alloys produced by die-casting, sand casting, and SLM. The as-produced SLM Al-12Si alloy shows much higher strength compared to the alloy produced by conventional casting, such as die-casting and sand casting. The yield strength and tensile strength of the as-produced SLM alloys are in the range of 200–250 MPa and 325–425 MPa, respectively, whereas the fracture strain does not exceed 5%, which is comparable to the alloy produced by die-casting. Compared to the cast Al-12Si alloy, the as-produced SLM alloy exhibits superior mechanical properties mainly due to the strengthening mechanisms resulting from a fine grain size in accordance with the Hall–Petch relationship and a homogeneous distribution of fine Si precipitates along the cellular grain boundaries [193,194,195,196].

The strength and ductility of the as-produced SLM Al-12Si alloys can be significantly modified by performing suitable heat treatments. The Variation in the mechanical properties of the Al-12Si alloy upon solution heat treatment for different durations is shown in Figure 59 [130]. Both yield strength and tensile strength decreased by more than 50% after solution heat treatment, while the ductility increases significantly from about 5% to 25%. Figure 60 shows the mechanical properties of Al-12Si after annealing heat treatments carried out at different temperatures for 6 h. With increasing annealing temperature, the strength of the alloy decreases and the ductility improves [67].

The changes occurring in the microstructure of the SLM Al-Si alloys during heat treatment lead to changes in their mechanical properties. The as-fabricated SLM Al-Si alloys possess a fine grain structure of a α-Al matrix supersaturated with Si owing to the rapid cooling. The highly metastable microstructure can be modified easily by heat treatment, where the grain size increases and the Si concentration in the Al-Si solid solution matrix decreases. For the Al-12Si alloy, the Si content in the α-Al matrix drops rapidly to ~2 wt.% after just 15 min and to 1.6 wt.% after 30 min upon heat treatment. Since the Si content of 1.6 wt.% corresponds to the equilibrium concentration, longer heat treatment has no further influence on the rejection of Si from α-Al. After very short solution treatment durations (≤15 min), tiny Si particles precipitate from the α-Al matrix. These Si particles continuously grow, even if there is no further Si rejection from the α-Al matrix, due to the combination of Ostwald ripening and coalescence of adjacent small Si particles [130]. Similarly, Prashanth et al. [67] reported that the amount of free Si increases from about 1 wt.% for as-prepared SLM samples to ~10 wt.% for the Al-12Si alloy annealed at 723 K, which is in accordance with the observations made by Li et al. [130]. The lattice parameter of Al increases after annealing at high temperatures due to the rejection of Si from the supersaturated α-Al matrix. The lattice parameter of α-Al increased from 0.405079 nm for the as-prepared SLM samples to 0.405225 nm for samples heat-treated at 723 K. As the annealing temperature increases, the microstructure became coarser. The crystallite sizes of Si and Al increase from about 8 and 118 nm, respectively, for the as-prepared SLM alloy to about 142 and 218 nm for the SLM samples annealed at 723 K.

Hence, for SLM Al-Si alloys, rejection of excess Si from the supersaturated α-Al matrix and subsequent precipitation is the fundamental phenomenon occurring during the heat treatments. Prashanth et al. [67] reported that precipitation of Si in Al-Si alloys showed two typical aspects. One was that the Si particles tend to agglomerate along the cellular boundaries and hatch overlaps, and their size grew continuously with increasing annealing temperature, all the while, the size of the particles in the hatch overlaps being constantly larger than in the track cores. Another aspect characterizing the heat-treated samples was the heterogeneous distribution of the Si particles. The amount of Si particles along the hatch overlaps varied from 7 to 0.3 particles/µm^2^ for the samples annealed at 473 K and 723 K, respectively, whereas in the track cores, the density of Si particle decreased in the same temperature range from 3 to 0.1 particles/µm^2^. The decrease of the particle density with increasing annealing temperature was apparently due to the growth of the particles because of the coalescence of smaller particles, as well as due to Ostwald ripening. The Si particles grew further and the cellular boundaries were no longer visible at higher annealing temperatures. The excess Si along the hatch overlaps acts as preferential sites for failure [67]. Most of the Si particles after solution treatment remained spherical, with only a small fraction becoming elongated particles [130].

There are certain ways to vary the room temperature tensile properties of Al-12Si samples during SLM fabrication. The combination of three variables, namely hatch style, contour and base plate heating can be effectively employed to tune the mechanical properties of the fabricated materials. For instance, samples with isotropic properties can be produced by employing different hatch styles without contour. Samples with anisotropic properties can be fabricated with a combination of (1) hatch style and base plate heating (2) contour and base plate heating and (3) hatch style, contour and base plate heating. The tensile data for Al-12Si SLM samples produced under thirty different conditions with and without heat treatment are shown in Figure 61, where the highest and lowest values are highlighted. Besides these parameters, *ex situ* heat treatment, combined with changing the laser parameters (laser power, laser scan speed, laser spot size) and layer thickness are other ways by which the mechanical properties of SLM components can be modified. The above parameters, both *in situ* and/or *ex situ*, and their various combinations offer a wide spectrum of properties to the SLM Al-12Si parts to match the service requirements in any desired application.

### 3.4. Al-Si-Mg-Based Alloys

The mechanical properties of AlSi10Mg SLM samples have been studied in detail at different length scales (nano, micro, and macro levels). Everitt et al. [196] evaluated the mechanical properties of SLM AlSi10Mg samples using nanoindentation. They showed that the hardness (around 2 GPa) across the melt pool is uniform (hardness variations are within the experimental error limits) (Figure 62) unlike for the cast sample, where the hardness varies between 1 GPa and 8 GPa. Such spatial uniformity in the nanohardness data is due to the fine microstructure with uniform distribution of Si. They also clarified the uniformity in the local mechanical properties in both the melt pool and across the layers [197,198]. It has to be noted that the material softens upon annealing treatment [169] and that the nano-hardness drops to ~1.5 GPa. This is because the benefits reaped by the fine microstructure created during the SLM process disappear as the microstructure becomes coarse, thereby losing its mechanical properties, especially the strength and hardness.

The Vickers micro-hardness was also measured for AlSi10Mg alloys, where SLM as-prepared samples exhibited a hardness of 125–130 HV and the micro-hardness drops nearly to 100 HV after solutionizing treatment [160,166,169,197,198,199]. The micro-hardness drops between 30 and 80 HV depending on the annealing conditions and it never reaches the original micro-hardness of the as-prepared SLM condition even after precipitation hardening treatment [157,198,200]. The hardness results for the AlSi10Mg alloy at different length scales reveal that the microstructure is locally uniform in the as-prepared SLM samples. However, the material softens with annealing/solutionizing treatment and it is not possible to regain the hardness of the original as-prepared SLM level after annealing, simply because the advantages in the microstructure after SLM (fine microstructure with uniform distribution of phases) are lost and cannot be mimicked [199,201].

The compressive stress of the AlSi10Mg samples in both the as-prepared SLM and annealed conditions was studied by Aboulkhair et al. (Figure 63) [169]. Both the as-built as well as the annealed samples do not fracture until an applied load of 230 kN is reached and the samples show a typical buckling/barrelling effect (Figure 63 (inset)), as it is common for Al-based alloys, where the presence of heterogeneous deformation is observed. The as-prepared AlSi10Mg SLM samples have a yield strength of 371 ± 2 MPa and an ultimate strength (at 25% strain) of 714 ± 1 MPa under compression, whereas the annealed sample shows a drop in yield- and ultimate strength (at 25% strain) of ~170 MPa and ~350 MPa, respectively, as shown in Figure 63 [169]. Nevertheless, the compressive strength of both the as-prepared SLM and annealed samples was higher than that of the cast counterpart [202]. Table 4 shows the tensile properties of AlSi10Mg samples fabricated with different process conditions (in both as-built and annealed conditions), as reported by different groups.

The tensile properties of the as-prepared SLM AlSi10Mg samples show a variation in yield strength between 170 and 330 MPa, the ultimate tensile strength between 290 and 435 MPa and the fracture strain between 1.0 and 5.5%, depending on the processing parameters and building direction. The results are scattered, i.e., some studies reveal uniform (minimal changes) in the tensile properties irrespective of the building direction [166,199], while others show a significant influence of the build direction on the tensile properties [203,204]. Hence, at this point, no definite conclusions concerning the dependency of the tensile properties on the building direction can be drawn. Nevertheless, the fracture strain increases with annealing treatment and the strength of the samples decreases considerably due to microstructure coarsening. However, the tensile properties of the as-prepared AlSi10Mg SLM samples are superior to those of conventional A360 die-cast material. The conventional behavior of SLM samples, where the strength of the material decreases with annealing treatment and fails to regain the same strength even after precipitation hardening, is also observed in this case. In a similar way, this also holds for the hardness results. The fracture surface of the as-prepared AlSi10Mg SLM samples exhibits both dimples and cleavage planes as shown in Figure 64.

Similar to the Al-12Si alloy [67], the fracture surface of the as-built SLM AlSi10Mg alloy showed dimples of size approximately 1 µm (marked by red arrows in Figure 64a, which were indicative of a ductile fracture. At the same time, the fracture surface also showed “river pattern-like” stepped cleavage planes (marked by the yellow arrow in Figure 64a suggesting a simultaneous local brittle failure of the sample. Thus, the fracture features of the as-built SLM AlSi10Mg indicate a ductile as well as a brittle failure. The fracture surface of the AlSi10Mg sample, solution heat-treated at 725 K for 2 h, showed equiaxed dimples with an average size of around 2 µm with no apparent cleavage planes, indicative of a highly ductile fracture. Moreover, the fractured Si particles were often observed at the end of the dimples and no decohesion of the Si particles from the matrix was observed, indicating a good bonding between the Al matrix and Si particles. As the solution treatment temperature was further increased, the size of the equiaxed dimples increased and reached 5 µm at 823 K (Figure 64b). Careful observation showed that the edges of the dimples passed through both Al matrix and eutectic Si particles, which further verified that the eutectic Si was firmly embedded within the Al matrix. After artificial aging at 723 K for 12 h, there was no significant change in the fracture morphologies, but a “dimples accumulation” phenomenon (dimples stacked together layer-by-layer) occurs on the fracture surface. Most likely, a fracture initiates and propagates through the “dimple accumulation” regions where the plastic deformation capacity is relatively low.

### 3.5. Al-Zn-Mg-Cu Based Alloys

The mechanical properties of SLM Al-Zn-Mg-Cu alloys have been very rarely studied compared to that of cast alloys. Wang et al. investigated the hardness of SLM Al-Zn-Mg-Cu alloy with Zn (9.1 ± 0.03), Mg (2.33 ± 0.06) and Cu (1.48 ± 0.02) content, which increased from 133 ± 6 HV_0.05_ for the as-prepared alloy to 219 ± 4 HV_0.05_ after T6 heat treatment [183]. Qi et al. reported a nano-hardness of 1.55 ± 0.26 GPa for as-prepared SLM Al7075 for the keyhole mode. The nano-hardness was not uniform in different regions of the melt pool under keyhole mode conditions with the highest measured nano-hardness at the bottom [184]. Compressive tests for SLM Al7075 and SLM Al7050 alloy were conducted by Sistiaga et al. [95] and Singh et al. [188], respectively. The yield strength increases from 279 ± 10 MPa for the as-prepared state to 338 ± 13 MPa for the heat-treated Al7050 + 4 wt.%Si alloy. The strength of the SLM alloy is much lower than that of conventional cast Al7075-T6 in both as-prepared and heat-treated conditions.

Reschetnik et al. [204] reported a highly anisotropic behavior for an SLM EN AW7075 alloy during tensile testing, as shown in Table 5. In addition, fatigue crack growth was also studied (Figure 65). The threshold value, ΔK_th,I_ for as-prepared SLM AW7075 samples is 1.77 ± 0.08 MPa m^1/2^, while for heat-treated samples it is 1.58 ± 0.03 MPa m^1/2^, indicating that the heat treatment does not have much influence on the crack growth behavior. The fatigue crack growth curves show a typical double S shape as it is well known for aluminium [204]. An exceptional tensile strength as high as ~400 MPa of 7XXX alloy was found by Martin et al. [185], which was within the expected bounds for its wrought counterpart (Figure 66). According to the authors, the incorporation of Al_3_Zr nucleant particles induced grain refinement and, hence, reduced solidification cracking and hot tearing, which are common issues for the additive manufacturing of 7XXX alloys [185].

## 4. Other Properties

Other properties of the SLM-fabricated metals and alloys such as tribological properties [205,206,207,208,209,210,211,212,213,214,215,216,217,218,219], fatigue properties [122,209,210,211,212,213,214,215], thermal conductivity [112,136,214,215], weldability [112,216,217,218,219,220], and corrosion [221,222,223,224,225,226] have also been studied recently.

### 4.1. Tribological Properties

Additively manufactured alloys are found to have better tribological properties than their counterparts produced by conventional techniques such as powder metallurgy and casting, which can be explained by their refined structure and high hardness [208,227,228]. Al-Si alloys are widely applied as engineering components, especially as moving parts, in automotive and aerospace industries. Besides low density, good castability, and good weldability, excellent wear and corrosion resistance are paramount for these alloys used in dynamic applications involving high friction. The sliding wear and fretting wear properties of SLM Al-12Si alloy were studied by Prashanth et al. [205]. Compared to cast parts, the SLM Al-12Si alloys exhibit superior wear resistance owing to their higher strength as well as hardness, resulting from an extremely refined grain size and Si particle distribution.

To avoid brittle fracture of the SLM Al-12Si alloys, heat treatment such as annealing at different temperatures was carried out, where the Si particle size, hardness, and strength decreased while the plasticity increased. The wear rate and the wear volume of the alloy increase with increasing annealing temperature. Figure 67 shows the influence of annealing temperature on the size of Si particles, the wear rate and the Vickers hardness of SLM Al-12Si samples, along with the corresponding values for as-cast Al-12Si. The as-prepared SLM samples exhibit the highest hardness and minimum wear rate. The fretting wear results for the SLM Al-12Si alloy show a similar behavior as for sliding wear, as shown in Figure 68. This behavior is significantly different from that of the cast material, which does not follow the trend shown by the SLM Al-12Si alloy. Figure 67 compares the wear rates of Al-Si alloys produced by different techniques, as a function of the Si content under identical testing conditions.

Despite implementing lower sliding speeds, the wear rates of samples fabricated by techniques other than SLM are much higher regardless of a high Si content. Prasad et al. [229] evaluated that the wear rate of Al-23.5Si samples produced by gravity casting (indicated by an arrow in Figure 69) was 50% higher than the as-prepared SLM sample, whereas the sample produced by pressure die-casting showed a similar wear rate even though the Si content was about two times higher. The wear resistance of SLM Al-18Si produced with different laser beam powers was studied by Kang et al. [153]. The microhardness of the SLM samples increased continually from 80 HV at 120 W to 105 HV at 210 W. This is probably because, at higher laser powers, the SLM samples tend to mitigate porosity. Regardless of the laser power, all the SLM Al-18Si samples showed the same average friction coefficient of about 0.46. However, the wear rate decreased as the laser power increased, owing to the improvement in hardness, where the samples obtained at 210 W exhibited the lowest wear rate of about 7.0 × 10^−4^ mm^3^·N^−1^·m^−1^. As reported by Torabian et al. [230], only chill casting is able to produce Al-12Si samples with superior wear resistance, comparable with that of SLM specimens, owing to its high cooling rate. As expected, the wear rate of the chill-cast materials decreases with increasing amount of hard Si phase (Figure 69). Except for the alloys produced by chill casting, all the Al-Si alloys produced by other processes [231,232] exhibit higher wear rates than the as-prepared SLM material. This indicates that the microstructural refinement achievable by SLM processing leads to a significant strengthening of the Al-12Si alloy, thereby inducing remarkable tribological properties [205].

### 4.2. Fatigue Properties

Fatigue strength is an essential property for engineering components exposed to cyclic loads encountered often in automotive and aerospace applications. SLM Al-Si alloys show poor fatigue strength compared with their cast counterparts. The remnant porosity, high surface roughness, conspicuous laser tracks, and extremely fine grains trigger early crack initiation and decrease the fatigue strength of SLM parts under cyclic loading. According to a few studies, the two important factors responsible for the poor fatigue performance of SLM parts are (a) high surface roughness, where surface irregularities behave like surface notches favoring premature crack initiation, decreasing the fatigue strength of the material [233], and (b) remnant porosity leading to fatigue scatter of SLM parts due to internal crack initiation. Though the overall relative density is higher than 99.5%, the existing pores about 100 µm show a deleterious effect on fatigue reliability [122,234]. The effect of porosity on the fatigue strength of the SLM Al-12Si alloy was studied systematically by Siddique et al. [122]. Figure 70 shows the porosity distribution in SLM Al-12Si samples, where the amount of porosity for these samples is 0.25% for batch B (Figure 70a) and 0.12% for batch D (Figure 70b). The distribution of defects is summarized in Figure 70c in the form of stacked histograms. Apparently, the fatigue strength improves by about 45% for SLM samples built with base plate heating, compared to those built without. For instance, the fatigue strength at 2 × 10^6^ cycles was 92.3 MPa and 67.4 MPa for the SLM Al-12Si samples with and without base plate heating, respectively, which has to be compared to about 55 MPa for conventionally manufactured material [122].

The large size and the scattering of defects are responsible for strongly varying fatigue properties. Figure 68 shows the facture surfaces of SLM Al-12Si samples indicating that large defects cause lower fatigue life. Large pores or un-melted powder particles can decrease the effective area of the specimen and initiate early cracks, even resulting in multi-crack initiation when several material defects are present (Figure 71c). Some extreme cases are that fatigue crack initiate at a smaller pore resulting in higher fatigue life (Figure 71d). The threshold value of the stress intensity factor, K is 3.2 MPam^0.5^ for batch B, which increases to 3.5 MPa m^0.5^ for batch D. The variation of the critical stress intensity factor for the base plate-heated batch is also higher than that without. Therefore, it can be deduced that the resistance to crack growth can be improved by building samples with base plate heating, which is attributed to lower cooling rates and, consequently, a coarsened microstructure. Coarse grains are favorable for better crack growth resistance since they have higher values in the critical region as well as in the threshold region [122,235,236].

### 4.3. Thermal Conductivity

SLM pure aluminum exhibits higher thermal conductivity than SLM Al alloys owing to the absence of solute elements in the former. For example, the thermal conductivity is ~200 W·m^−1^·K^−1^ and ~130 W·m^−1^·K^−1^ for SLM pure aluminum and AlSi10Mg alloy, respectively [84,104]. However, the thermal conductivity of the SLM pure aluminum is 20–30 W·m^−1^·K^−1^ lower than that of the A1060-O wrought material, which may be due to the presence of residual strain resulting from high thermal gradients during SLM processing. Nevertheless, heat treatments such as annealing may be performed to reduce the residual strains in the as-prepared SLM parts to improve the thermal conductivity. For example, the thermal conductivity increases to ~240 W·m^−1^·K^−1^ for the SLM sample annealed at 723–773 K for 10 min, which is even higher than the thermal conductivity of A1060-O wrought material (~130 W·m^−1^·K^−1^). The thermal conductivity of Al-Si SLM alloy decreases with increasing Si content. The thermal conductivity of about 160 W·m^−1^·K^−1^ for the Al-4Si SLM sample drops to about 105 W·m^−1^·K^−1^ for Al-20Si SLM sample. Besides, the thermal conductivity of Al-Si SLM alloys also depends on the building direction. For example, the conductivity of Al-1Si SLM sample built at 0° and 90° are 180 W/m·K and 130 W/m K respectively. Interestingly, no anisotropy in thermal conductivity between the 0° and 90° specimens was observed for other Al-xSi SLM samples, except for Al-1Si.

### 4.4. Weldability

A major limitation for the widespread application of SLM as a commercial processing route is the limited size of the products, which is a direct consequence of the limited dimensions of the SLM building chambers. The present technology allows for the production of samples with volumes of about 0.02 m^3^. A possible way to overcome this problem and create larger components with no dimensional limitations is the use of welding processes to join smaller SLM parts [112,216]. The yield strength of the Al-12Si alloy processed by SLM is four times higher than that of a conventionally cast material. Such high strength and internal strains in the SLM alloys favor crack formation during welding. A possible exception is solid-state welding that prevents issues related to solidification cracking, liquation cracking, segregation and formation of brittle eutectics/intermetallics. In addition, solid-state welding results in fine-grained microstructures, a narrow heat-affected zone and low residual stresses in the weldment. Friction stir processing of SLM-fabricated AlSi10Mg alloy was studied by Wang et al. indicating that larger parts can be obtained by this process [217]. Among the common solid-state joining processes, friction welding (FW) has attracted considerable attention due to economic considerations and high productivity. In this process, fusion is facilitated by the heat that is generated by the conversion of mechanical energy into thermal energy at the interfaces of the parts, rotated under pressure. Compared with other welding techniques, FW offers advantages such as high material saving, short joining time and the possibility of creating dissimilar joints.

Solid-state welding of SLMed parts was first tried by Prashanth et al. [216], using FW to join Al-12Si parts produced by selective laser melting. The microstructure, hardness and tensile properties of the welded samples are pronouncedly changed compared to the initial SLMed specimens. The welded Al-12Si parts are shown in Figure 72, indicating a smooth flash at the joint, which shows adequate heat generation and plastic deformation. The weld zone contains the same phases as the SLM-based material but with a reduced amount of Si precipitates. The hardness of the weld zone decreases quickly from ~95 HV_0.01_ for the SLM-based metal to ~81 HV_0.01_. Figure 73 shows the tensile properties of the FW samples together with the properties of as-SLM and cast specimens. For the casting samples, the strength increases while the ductility decreases after FW. On the contrary, the strength decreases while the plastic deformability significantly increases after FW for the SLM samples, where the plastic strain increases from ~3% to ~10% after FW. This demonstrates that the materials produced by SLM can be successfully joined by friction welding.

Friction stir welding (FSW) was successfully used to weld SLM AlSi10Mg plates by Scherillo et al. [219]. They showed that the layer-by-layer morphology created by SLM is broken down during FSW, leading to a more homogenous microstructure and finer grains. The refined microstructure causes a higher hardness of the welded zone than for the SLM-base metal, which is different from the FW samples discussed above. These findings are in contrast with another work, which shows that the micro-hardness decreases significantly in the stir zone for FSW AlSi10Mg SLM samples [220]. The decreased hardness in the stir zone was explained by the dissolution of hardening precipitates in the aluminum matrix. Hence, the strength and ductility are significantly reduced for the FSW specimens compared to the SLM samples before welding [220]. However, these are just preliminary and findings and there is plenty of room for further optimization of the solid-state welding of SLM parts to improve their mechanical properties such as, for example, their tensile properties.

### 4.5. Corrosion

Corrosion resistance is another important property for Al and its alloys. For example, Al-Si alloys have been extensively used as pistons or cylinder heads in engineering components in the automotive and aerospace industries, which experience high temperatures and complex environmental conditions. Prashanth et al. reported on the corrosion resistance of an Al-12Si alloy produced by select laser melting in acidic conditions [205]. Figure 74a shows the weight-loss curves for the as-prepared SLM specimens as a function of the HNO_3_ concentration, revealing that the weight-loss of the as-prepared SLM samples increases quickly with increasing nitric acid concentration. The weight-loss curves for the cast, as-prepared SLM and SLM heat-treated samples in 1 M HNO_3_ solution are shown in Figure 74b. The corrosion behavior of the as-prepared SLM samples is similar to that of the cast material. However, the weight-loss gradually increases with increasing annealing temperature when the SLM sample is annealed.

The Pourbaix diagrams for Al and Si are shown in Figure 75, suggesting the dissolution of Al in the form of Al^3+^ ions [205]. It was found that the Al-rich phase in the samples corrodes preferentially, Si-rich phase remains. It is known that SiO_2_ is the most favored state for Si; therefore, further oxidation of Si can be blocked due to the formation of SiO_2_ passive films (Figure 75b). Finally, selective corrosion with Al dissolution occurs in the Al–Si system in contact with HNO_3_ along with the formation of a SiO_2_ passivation layer. It shows that both the morphology and size of the Si phase plays a significant role in determining the corrosion resistance of Al-xSi based alloys.

## 5. Summary and Outlook

Aluminum alloys are one of the most important class of non-ferrous materials in terms of applications and problems that challenge science and technology developments. AM processes like selective laser melting (SLM) are recently developed production technologies and are regarded as one of the manufacturing processes of the future. This review presents the state-of-the-art advances in the processing of several important Al alloys by the most promising modern SLM process. The advantages and limitations of processing Al-based alloys by SLM have been elucidated and the challenges associated with the laser-aluminum interactions have been addressed. The effect of laser parameters such as the power, energy density, wavelength, hatch style, hatch distance, and powder characteristics, such as the flowability, composition, size, shape, and size distribution, on the probable defects and densification level of the materials have been discussed in detail for every alloy type. It may be observed from the processing of Al-based alloys that the process parameters vary largely depending on the alloying addition and the factors to be considered in developing the processing parameters change, depending on the alloy constituent. For instance, in the case of Al-Cu, there are no restrictions with the alloying addition, when it comes to process parameter selection. On the other hand, with AlSiMg or with 7XXX series, vaporization of the low volatile elements like Mg or Zn has to be given due care and hence the energy density cannot be increased beyond certain values. In addition, the laser scan speed has to be kept as high as possible to avoid vaporization of these volatile elements.

Although pure aluminum has extremely limited applications, its fabrication via laser processing is fundamental to understanding and overcoming issues such as high reflectivity, electrical conductivity and so on. Hence, SLM of pure Al, followed by the processing of industrially relevant Al-Cu-Mg, Al-Cu (2XXX), Al-Si (4XXX), Al-Si-Mg (6XXX), and Al-Zn-Mg-Cu (7XXX) alloys have been discussed. Common defects, such as porosity, balling, residual stresses, arising in the SLM-processed Al alloys, have been addressed separately for every alloy class. An extensive analysis of the available literature on the densification, microstructural and compositional features of the alloys produced by SLM, in constant comparison with that of their cast counterparts, has been undertaken for relating the process parameters to the microstructure and properties of the alloys. The mechanical and tribological properties, as well as thermal conductivity and weldability, of SLM-fabricated Al alloys have shown to possess different characteristics compared to their cast counterparts.

It is interesting to observe that the mechanical properties (tensile properties) of the Al-based alloys fabricated by SLM do not exceed the following values: yield strength: 350 MPa (max.), ultimate tensile strength: 500–525 MPa (max.) and ductility: 20% (max.) in the as-produced condition in the wide range of alloys considered from 1XXX to 7XXX alloys. Even precipitation-hardening treatments do not significantly improve the tensile properties of these alloys in terms of strength. However, the ductility of the alloys may be improved with thermal treatments. The SLM samples in any condition (as prepared or solutionized or precipitation hardened) do not match the tensile properties of the precipitation-hardened conventional Al-based alloys like 2XXX and/or 7XXX series. This is due to premature failure in these Al-based alloys processed by SLM and the presence of defects like porosity and un-melted particles. Hence, we need a strategy to design and develop novel Al-based alloys in accordance with the process conditions (high cooling rates, peculiar solidification conditions, etc.), which can be processed without the presence of defects like porosity and un-melted particles. This can avoid premature failure, may help in reaping the benefits of the SLM process, and at the same time exhibit unprecedented properties in tension (the combination of strength, ductility, and toughness).

## Figures and Tables

**Figure 1 materials-13-04564-f001:**
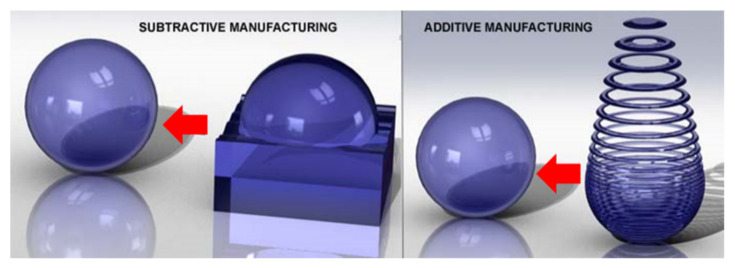
Schematic illustration showing the differences between subtractive (**left**) and additive (**right**) manufacturing [12].

**Figure 2 materials-13-04564-f002:**
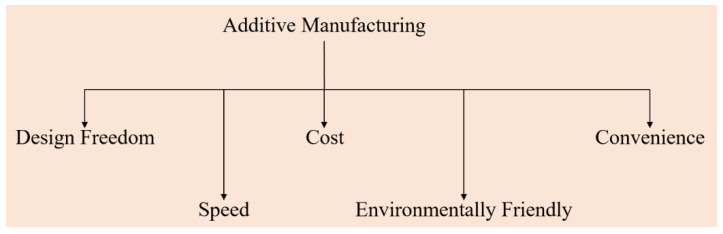
Schematic illustration showing the advantages of additive manufacturing in fabricating parts.

**Figure 3 materials-13-04564-f003:**
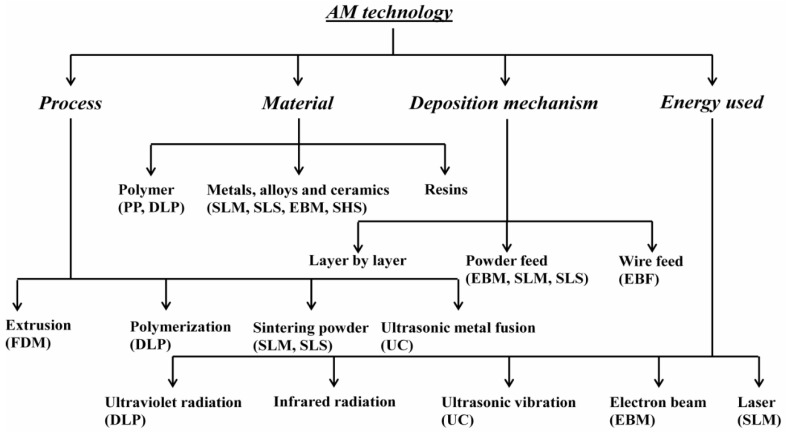
Detailed flowchart showing the classification of the additive manufacturing process based of four major groups.

**Figure 4 materials-13-04564-f004:**
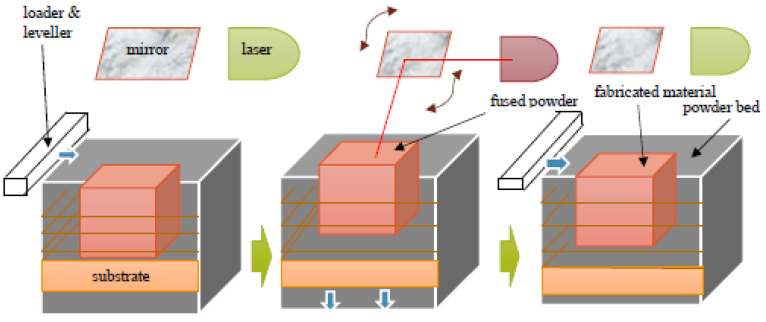
Schematics of Selective Laser Melting process—flow diagram.

**Figure 5 materials-13-04564-f005:**
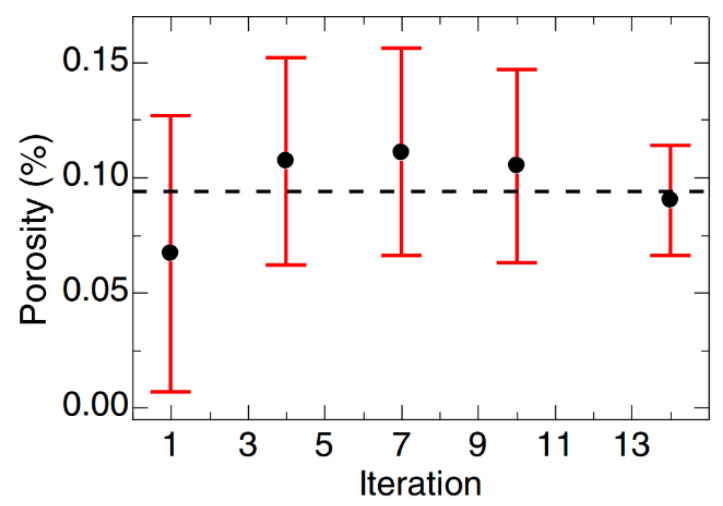
Results from porosity measurements performed on 3D printed samples. Each point represents the average porosity (in percentage) found in six samples fabricated in five particular iterations. Red bars indicate standard deviation from this value. The dashed line represents the average porosity obtained from these five values [14]. Copyright 2014. Adapted with permission from Elsevier Science Ltd. under the license number 4803690436753 (Figure 5 [14]), dated 7 April 2020.

**Figure 6 materials-13-04564-f006:**
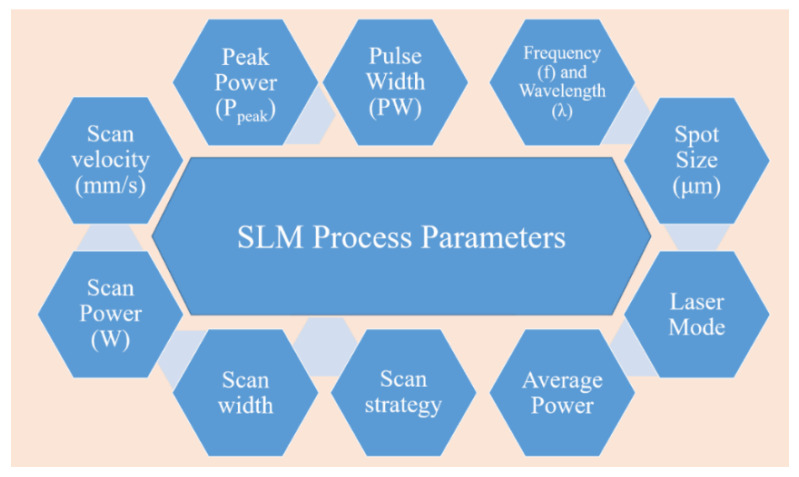
Schematic illustration showing the most influential SLM process parameters.

**Figure 7 materials-13-04564-f007:**
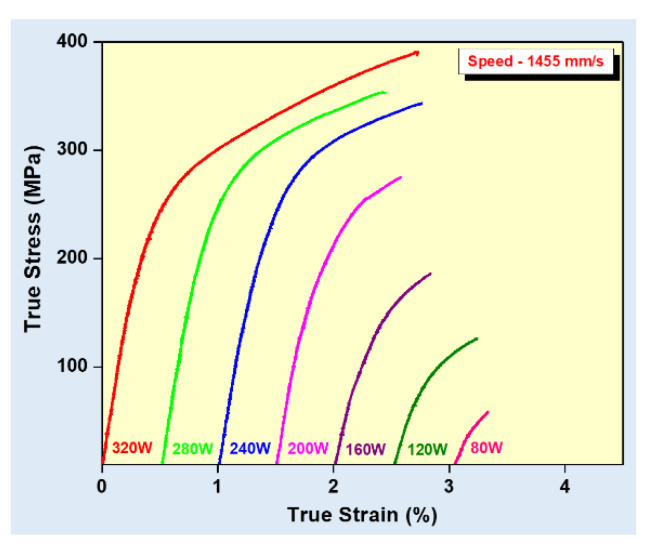
Room temperature tensile true stress-true strain curves of Al-12Si samples fabricated with decreasing laser power (320 W–80 W) at a constant laser scan speed (1455 mm/s) in an SLM solutions SLM 250 device [68]. Copyright 2017. Adapted as per the open access policy of Taylor and Francis group (Figure 1b [68]).

**Figure 8 materials-13-04564-f008:**
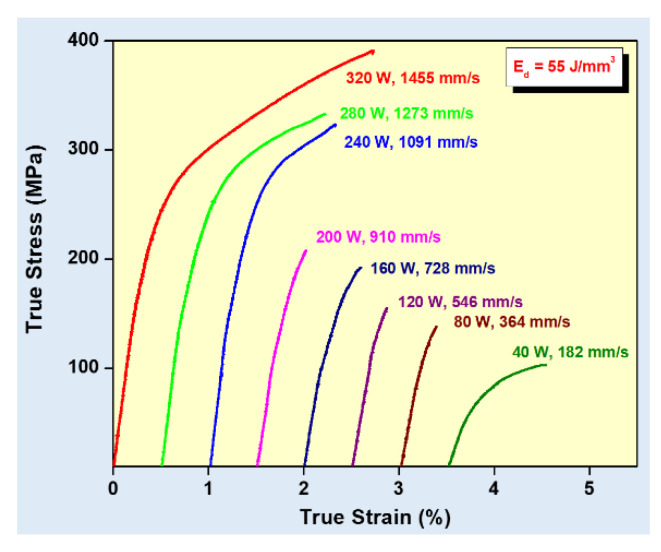
Room temperature tensile true stress-true strain curves of Al-12Si samples fabricated at a constant energy density of 55 J/mm^3^ with varying laser power and laser scan speed in an SLM Solutions SLM 250 device [68]. Copyright 2017. Adapted as per the open access policy of Taylor and Francis group (Figure 1a [68]).

**Figure 9 materials-13-04564-f009:**
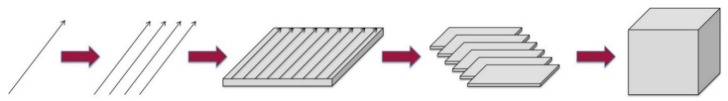
Flowchart describing SLM processing from a single laser track to the 3D object [12].

**Figure 10 materials-13-04564-f010:**
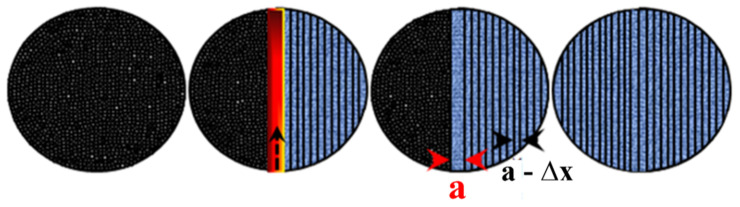
Schematics showing the hatch distance and hatch overlaps observed during the selective laser melting process [12].

**Figure 11 materials-13-04564-f011:**
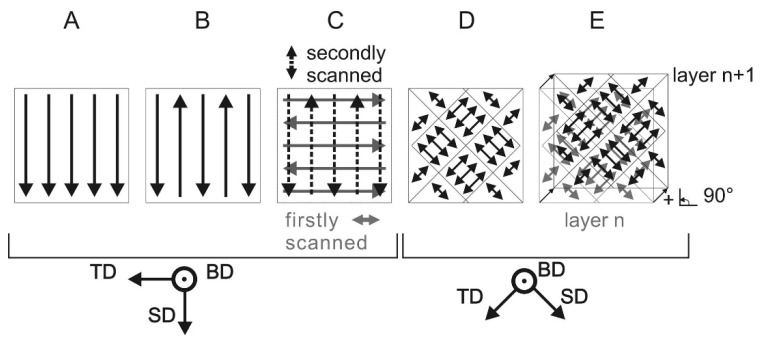
Schematics showing the different types of hatch styles that are employed during the SLM process. The building (BD), scanning (SD), and transverse directions (TD) are indicated. Sample A—scan with unidirectional vectors; sample B—bidirectional vectors that reverse between each hatch; sample C double scan strategy, firstly with bidirectional vectors in TD and secondly scanned with bidirectional vectors in SD; sample D—scan with island strategy with 90° rotation but without shift and sample E—scan with island strategy with 90° rotation and 1 mm shift between the layers [71]. Copyright 2012. Adapted with permission from Elsevier Science Ltd. under the license number 4803690935278 (Figure 1 [71]), dated 7 April 2020.

**Figure 12 materials-13-04564-f012:**
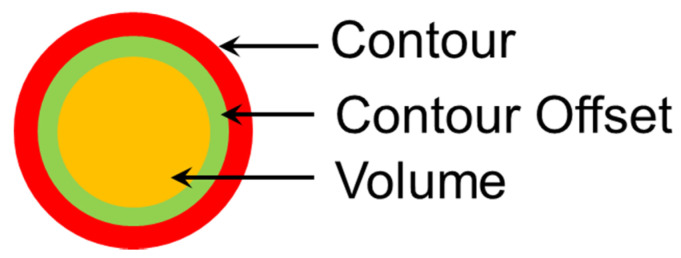
Schematic illustrating the different zones identified in an SLM part: contour, contour offset and actual volume of the part.

**Figure 13 materials-13-04564-f013:**
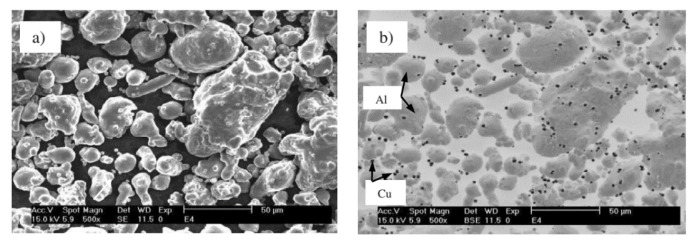
SEM micrographs of custom-developed Al-Cu powders (**a**) powder observed by secondary electron imaging in SEM; (**b**) powder sample observed in backscattering (BSE)-SEM mode [114]. Copyright 2011. Adapted with permission from Elsevier Science Ltd. under the license number 4803700253132 (Figure 5 [114]), dated 7 April 2020.

**Figure 14 materials-13-04564-f014:**
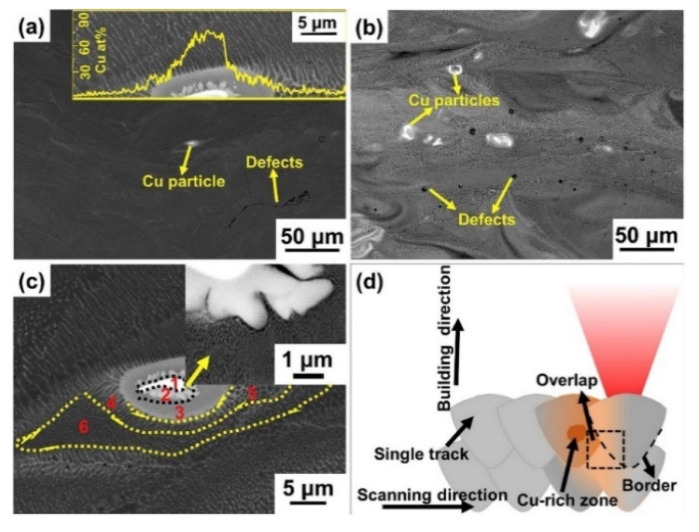
BSE-SEM micrographs of SLM Al-xCu alloys: (**a**) 6Cu alloy (inset: Cu distribution across the Cu-rich zone); (**b**) 40Cu alloy; (**c**) Cu-rich zone in the 6Cu alloy (inset: high magnification). (**d**) Schematic illustrating the diffusion of Cu during SLM processing [115]. Copyright 2017. Adapted with permission from Elsevier Science Ltd. under the license number 4803700463746 (Figure 2 [115]), dated 7 April 2020.

**Figure 15 materials-13-04564-f015:**
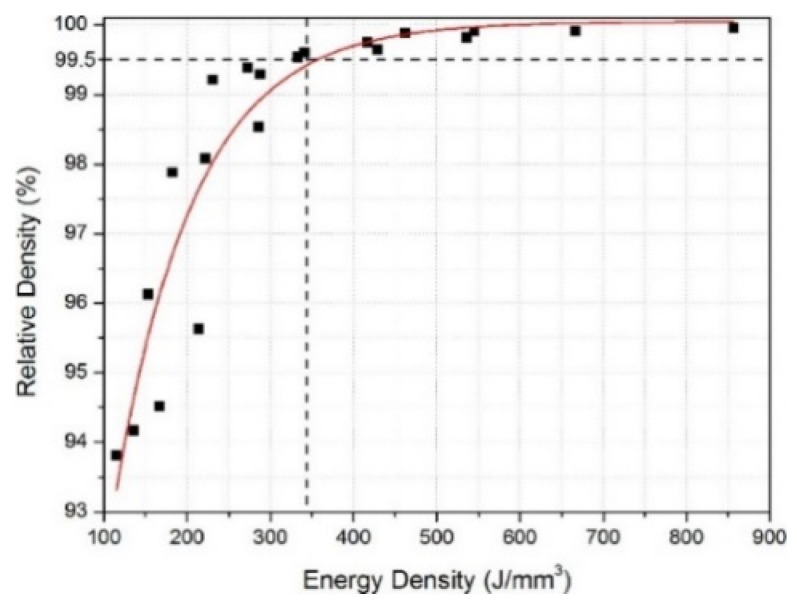
Relative density of SLM Al–Cu–Mg samples as a function of laser energy density [118]. Copyright 2016. Adapted with permission from Elsevier Science Ltd. under the license number—4803710573901 (Figure 5 [118]), dated 7 April 2020.

**Figure 16 materials-13-04564-f016:**
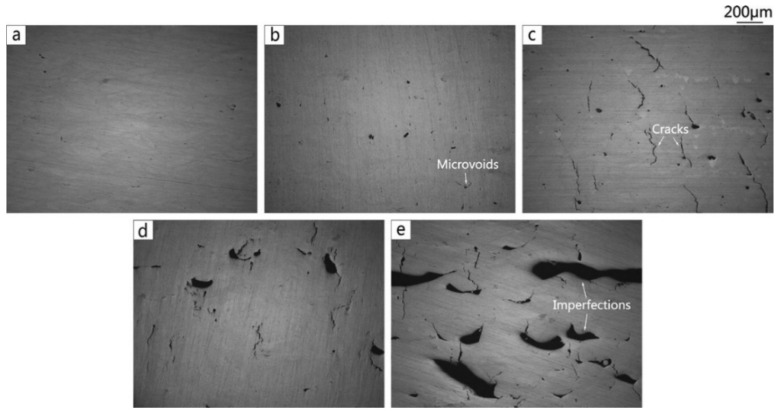
Cross-sections of Al-Cu-Mg samples produced with different scanning speeds: (**a**) 5 m/min, (**b**) 8 m/min, (**c**) 10 m/min, (**d**) 15 m/min, and (**e**) 20 m/min [118]. Copyright 2016. Adapted with permission from Elsevier Science Ltd. under the license number 4803710859427 (Figure 4 [118]), dated 7 April 2020.

**Figure 17 materials-13-04564-f017:**
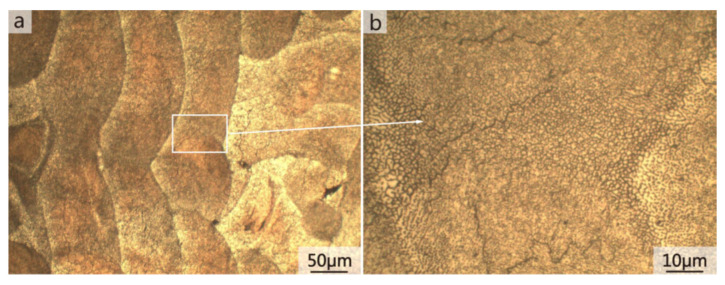
(**a**) Optical micrographs of SLM-processed Al-Cu-Mg samples, (**b**) a higher magnification micrograph detailing the laser tracks shown in (**a**) [117]. Copyright 2016. Adapted with the permission from SPIE publishers (Figure 4 [117]).

**Figure 18 materials-13-04564-f018:**
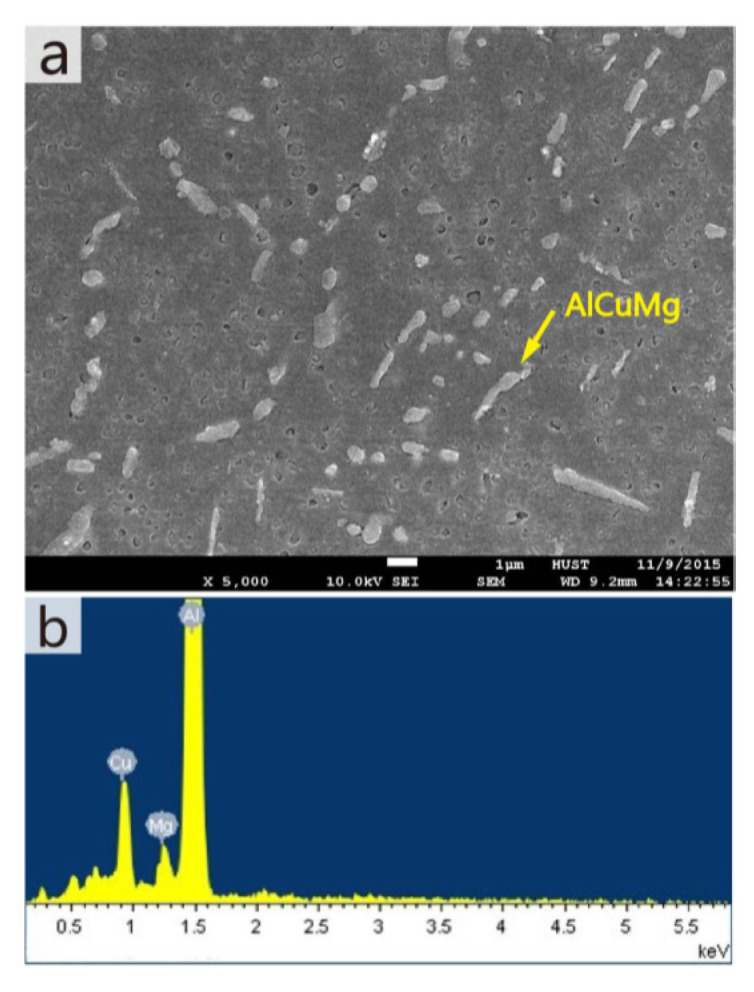
SEM image of an SLM-processed Al-Cu-Mg alloy sample under H4 treatment, (**b**) EDS spectrum corresponding to the AlCuMg phase shown in (**a**) [117]. Copyright 2016. Adapted with the permission from SPIE publishers (Figure 6 [117]).

**Figure 19 materials-13-04564-f019:**
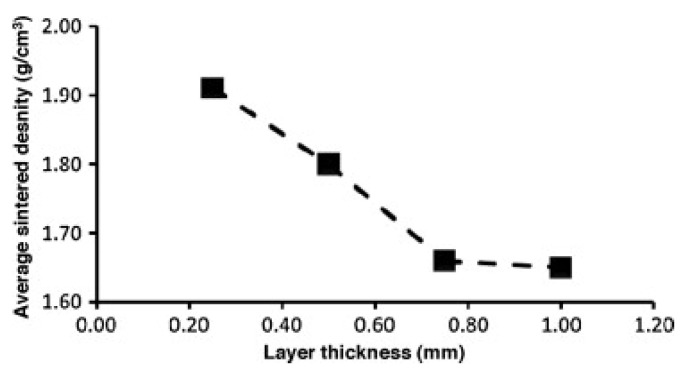
Variation of the density of SLS processed Al-12Si powder (45–75 µm) with layer thickness at fixed laser power (240 W), scanning rate (120 mm/s) and scan spacing (0.1 mm) [133]. Copyright 2010. Adapted with permission from Elsevier Science Ltd. under the license number—4803711190793 (Figure 5 [133]), dated 7 April 2020.

**Figure 20 materials-13-04564-f020:**
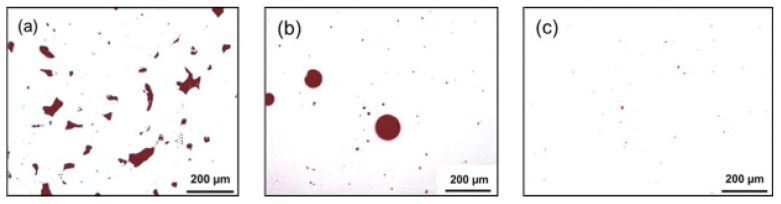
Polished micro-sections for Al-12Si alloy processed by selective laser melting [135]. Samples prepared with (**a**) low energy density (20 J/mm^3^), no base plate heating and no post-build stress relief, (**b**) high energy density (39.6 J/mm^3^), 200 °C base-plate heating and no post-build stress relief, and (**c**) high energy density (39.6 J/mm^3^), 200 °C base-plate heating and 240 °C post-build stress relief. Copyright 2015. Adapted with permission from Elsevier Science Ltd. under the license number 4803720074625 (Figure 3 [135]), dated 7 April 2020.

**Figure 21 materials-13-04564-f021:**
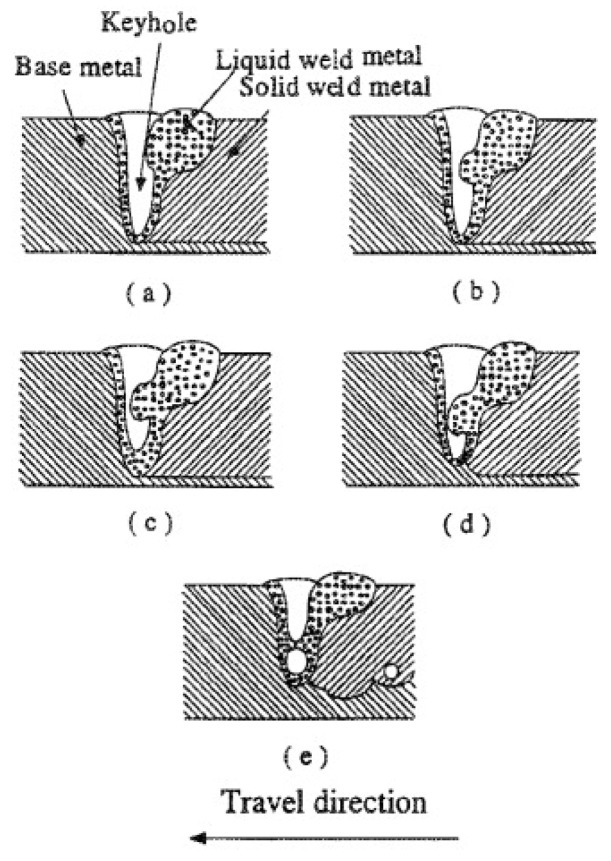
Schematic illustration of keyhole pore formation in aluminum alloys during SLM [92,132]: (**a**) formation of key hole due to excessive energy, (**b**) flow of liquid weld metal, (**c**,**d**) partial flow of the liquid metal to the keyhole pore, and (e) formation of a power within the keyhole pore. Copyright 2017. Adapted with permission from Elsevier Science Ltd. under the license number—4803720213102 (Figure 1 [132]), dated 7 April 2020.

**Figure 22 materials-13-04564-f022:**
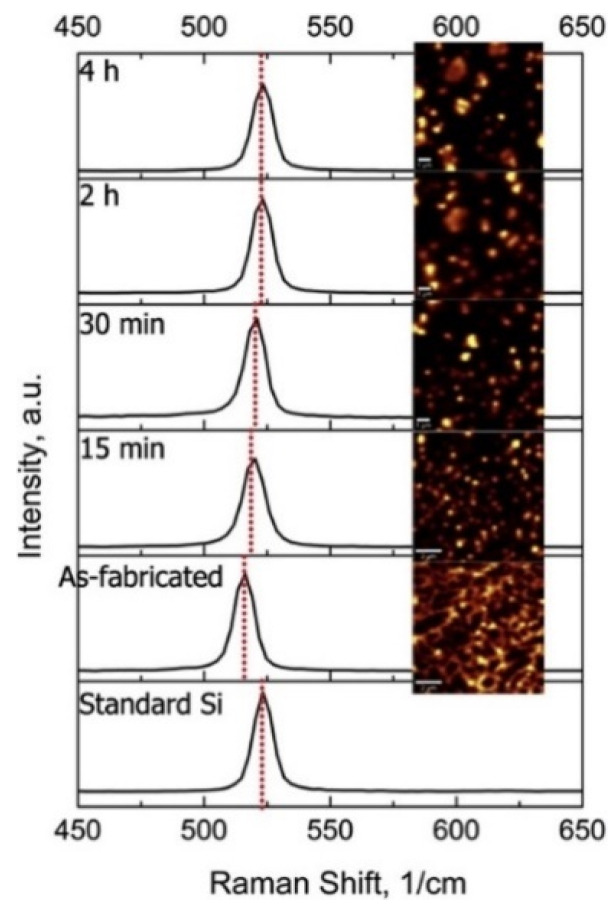
Raman spectra for Si particles in Al-12Si alloys solution treated for different durations. The inset shows the Raman intensity mapping of the corresponding Al-12Si alloys measured in the region of 500–550 cm^−1^. The Raman spectrum for standard Si is also shown. A gradual decrease in the Raman shift during heat treatment can be observed [130]. Copyright 2015. Adapted with permission from Elsevier Science Ltd. under the license number 4803720364624 (Figure 8 [130]), dated 7 April 2020.

**Figure 23 materials-13-04564-f023:**
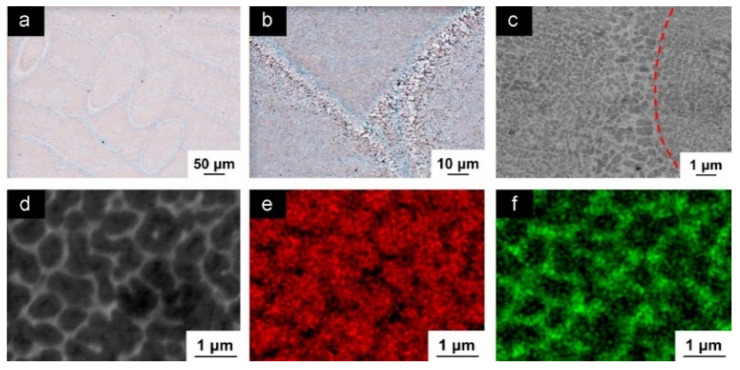
Microstructure of Al-12Si samples prepared by SLM with angle of inclination γ = 90° (i.e., perpendicular to the substrate plate): (**a**,**b**) OM and (**c**,**d**) SEM micrographs, and (**e**,**f**) EDX composition maps for Al (red) and Si (green), respectively [67]. Copyright 2013. Adapted with permission from Elsevier Science Ltd. under the license number 4803720747532 (Figure 1 [67]), dated 7 April 2020.

**Figure 24 materials-13-04564-f024:**
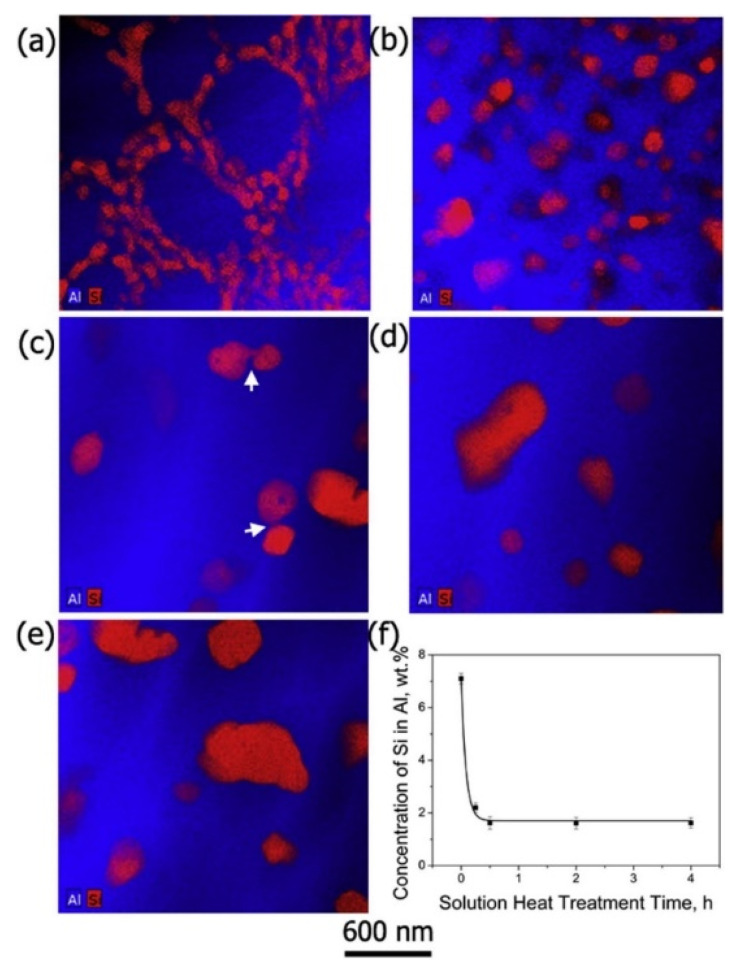
STEM-EDX maps of the Al and Si distribution in the Al-12Si alloy in the (**a**) as-fabricated condition and after solution treatment at 773.15 K for (**b**) 15 min; (**c**) 30 min; (**d**) 2 h; and (**e**) 4 h; (**f**) Concentration of Si in Al for different solution heat treatment durations. The white arrows in (**c**) show the position of the joining neck between adjacent Si particles [130]. Copyright 2015. Adapted with permission from Elsevier Science Ltd. under the license number 4803721064179 (Figure 4 [130]), dated 7 April 2020.

**Figure 25 materials-13-04564-f025:**
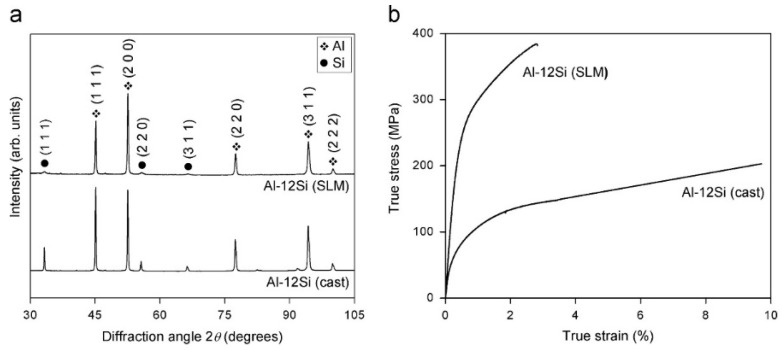
(**a**) XRD patterns (λ = 0.17889 nm) and (**b**) room temperature tensile tests of cast and as-prepared SLM Al-12Si samples [67]. Copyright 2013. Adapted with permission from Elsevier Science Ltd. under the license number 4803720747532 (Figure 3 [67]), dated 7 April 2020.

**Figure 26 materials-13-04564-f026:**
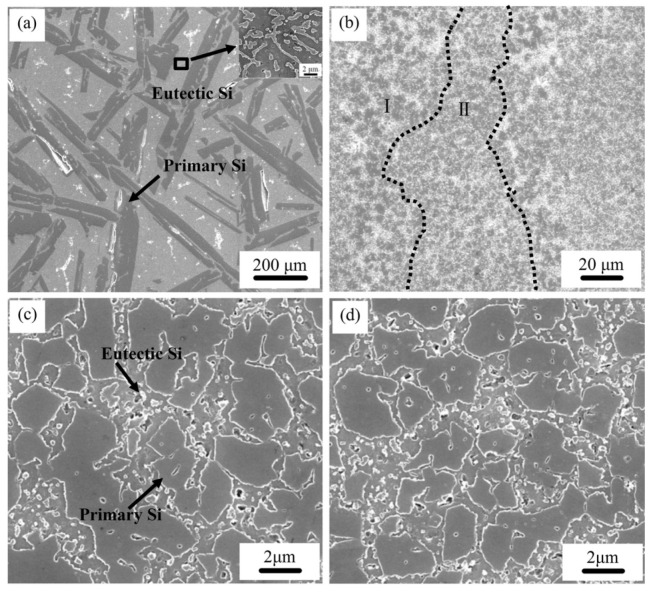
Microstructures of as-cast and SLM processed Al-50Si alloys. (**a**) As-cast alloy; (**b**) SLM alloy. (**c**) Part I in (**b**); (**d**) Part II in (**b**) [90]. Copyright 2017. Adapted with permission from Elsevier Science Ltd. under the license number 4803721234381 (Figure 1 [90]), dated 7 April 2020.

**Figure 27 materials-13-04564-f027:**
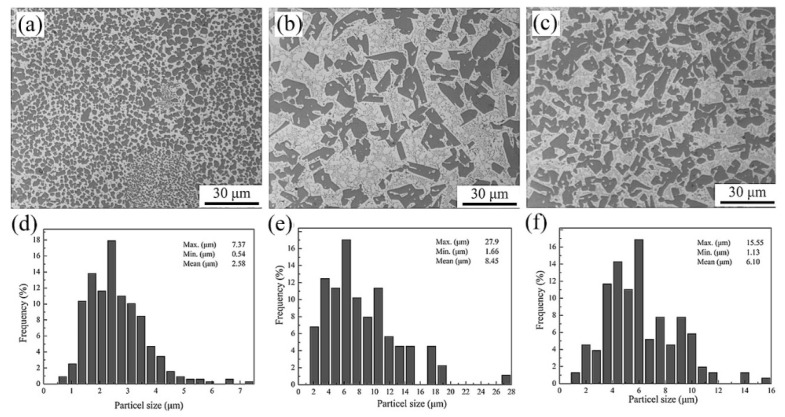
Optical microscopy images of the microstructure of SLM-processed parts obtained at a laser power of 320 W, (**a**) contour, (**b**) middle (contour offset) and (**c**) central regions and the corresponding dimension analysis of the primary silicon phase (**d**) contour, (**e**) middle and (**f**) central regions [152]. Copyright 2015. Adapted with permission from Elsevier Science Ltd. under the license number 4803730069705 (Figure 3 [152]), dated 7 April 2020.

**Figure 28 materials-13-04564-f028:**
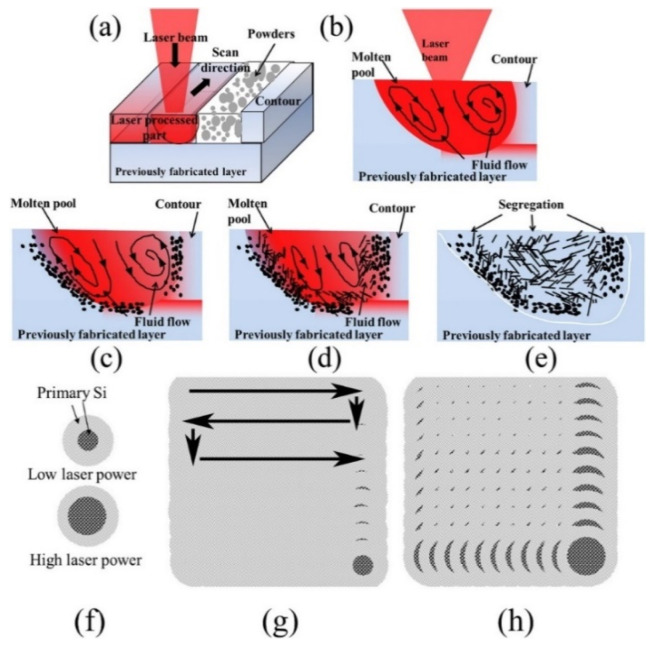
Schematic illustration of segregation mechanism of an SLM-processed hypereutectic Al-50Si alloy during remelting and solidification: (**a**) scanning mode, (**b**–**e**) solidification process, (**f**) melt pool, scanning results of single layer of samples obtained at (**g**) low and (**h**) high laser powers [152]. Copyright 2015. Adapted with permission from Elsevier Science Ltd. under the license number 4803730069705 (Figure 4 [152]), dated 7 April 2020.

**Figure 29 materials-13-04564-f029:**
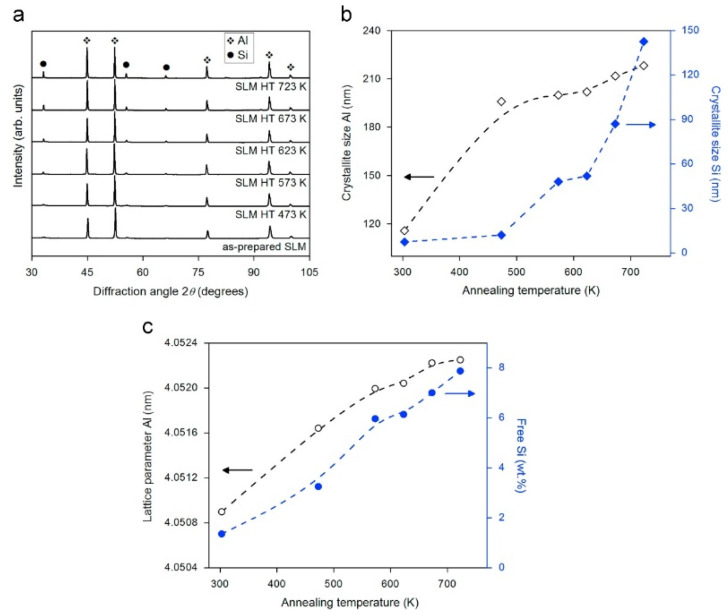
(**a**) XRD patterns (λ = 0.17889 nm) of the Al-12Si SLM specimens (γ = 90°) isothermally annealed for 6 h at temperatures between 473 and 723 K; (**b**) crystallite sizes of Al and Si, and (**c**) lattice parameters of Al and the amount of free Si versus the annealing temperature [67]. Copyright 2013. Adapted with permission from Elsevier Science Ltd. under the license number 4803720747532 (Figure 5 [67]), dated 7 April 2020.

**Figure 30 materials-13-04564-f030:**
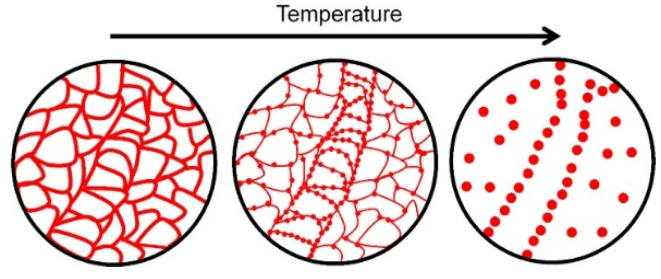
Schematic description of the microstructure evolution of the SLM samples during annealing. Red features represent Si-rich areas [67]. Copyright 2013. Adapted with permission from Elsevier Science Ltd. under the license number 4803720747532 (Figure 9 [67]), dated 7 April 2020.

**Figure 31 materials-13-04564-f031:**
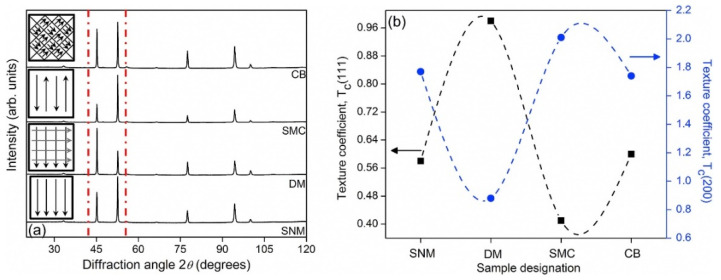
(**a**) XRD patterns (λ = 0.17889 nm) of Al-12Si SLM specimens produced using different hatch styles and (**b**) corresponding texture coefficient of the Al (111) and (200) planes [157]. Copyright 2016. Adapted with permission from Elsevier Science Ltd. under the license number 4803730407471 (Figure 3 [157]), dated 7 April 2020.

**Figure 32 materials-13-04564-f032:**
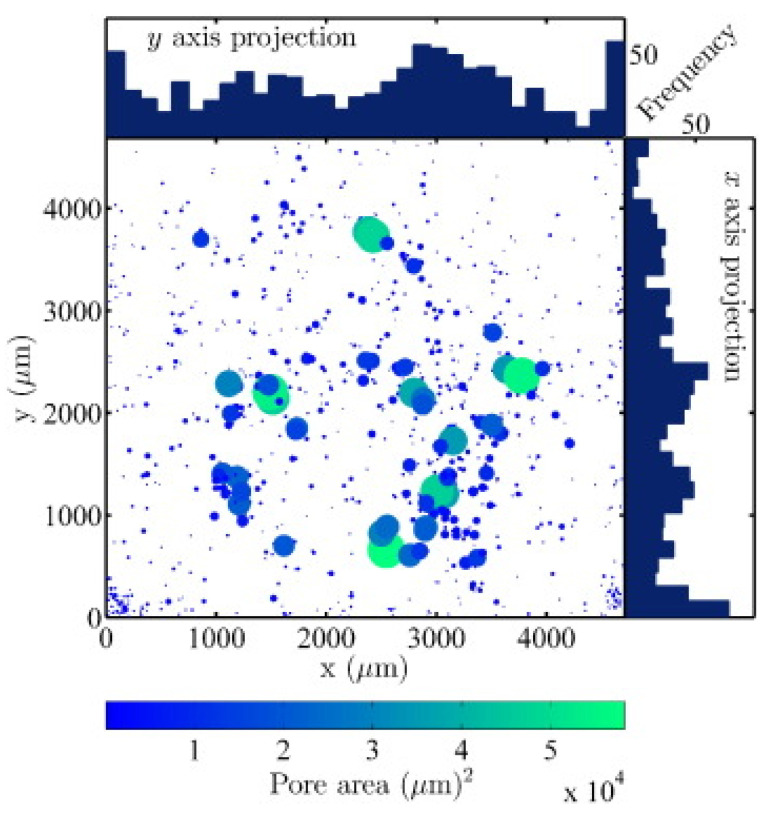
Pores in an AlSi10Mg SLM sample characterized by X-ray tomography [162]. Copyright 2015. Adapted with permission from Elsevier Science Ltd. under the license number 4803730656907 (Figure 7 [162]), 7 April 2020.

**Figure 33 materials-13-04564-f033:**
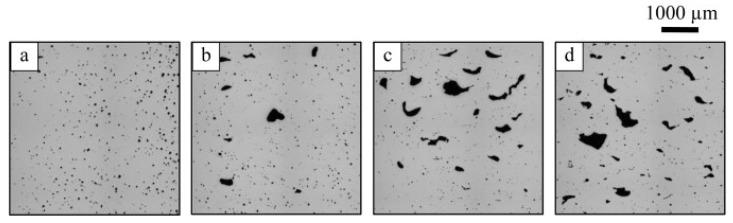
Evolution of pores in SLM-processed AlSi10Mg as a function of laser scan speed: (**a**) 250 mm/s, (**b**) 500 mm/s, (**c**) 750 mm/s and (**d**) 1000 mm/s [76]. Copyright 2014. Adapted from Elsevier Science Ltd. (Figure 5 [76]).

**Figure 34 materials-13-04564-f034:**
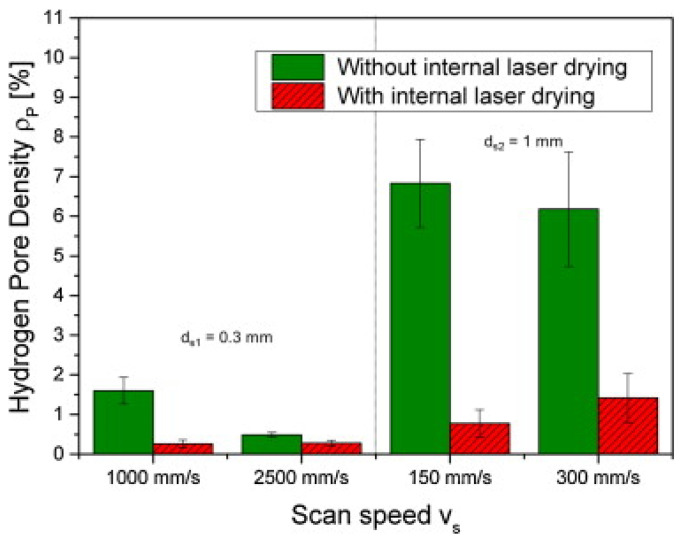
Laser scan rate dependence of the hydrogen pore density observed in AlSi10Mg samples processed by SLM [88]. Copyright 2015. Adapted with permission from Elsevier Science Ltd. under the license number 4803730859506 (Figure 7 [88]), dated 7 April 2020.

**Figure 35 materials-13-04564-f035:**
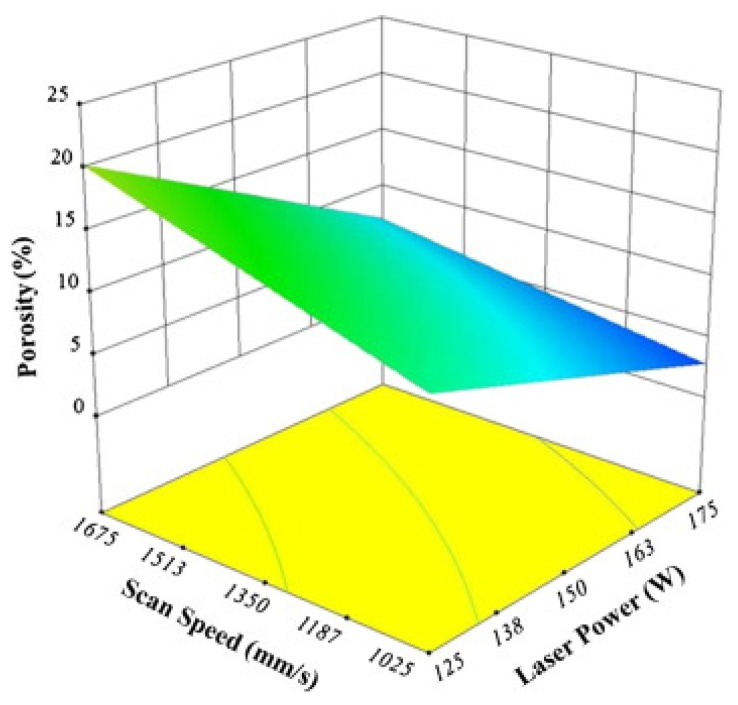
Response surface plot showing the effect of the laser power and the laser scan speed on the porosity level at a constant hatch spacing (0.5 µm) and island size (5 mm) [166]. Copyright 2014. Adapted with permission from Elsevier Science Ltd. under the license number 4803731023945 (Figure 4 [166]), dated 7 April 2020.

**Figure 36 materials-13-04564-f036:**
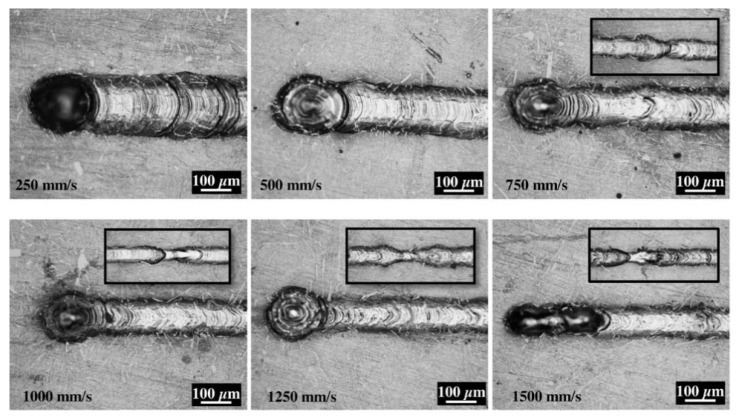
Variation in the shape of the fusion line of AlSi10Mg SLM single tracks as a function of increasing laser scan speed [167]. Copyright 2015. Adapted with permission from Elsevier Science Ltd. under the license number 4803731171579 (Figure 2 [167]), dated 7 April 2020.

**Figure 37 materials-13-04564-f037:**
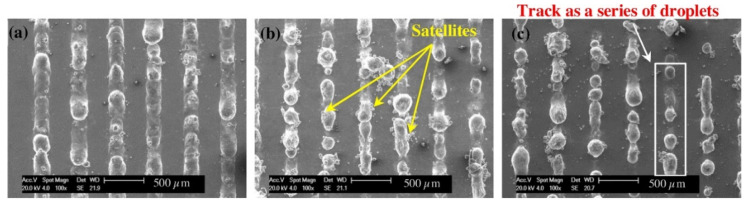
AlSi10Mg single tracks showing the variations in the shape of the fusion line and the defect formation as a function of laser scan speed (**a**) 250 mm/s, (**b**) 500 mm/s and (**c**) 750 mm/s [167]. Copyright 2015. Adapted with permission from Elsevier Science Ltd. under the license number 4803731331814 (Figure 9 [167]), dated 7 April 2020.

**Figure 38 materials-13-04564-f038:**
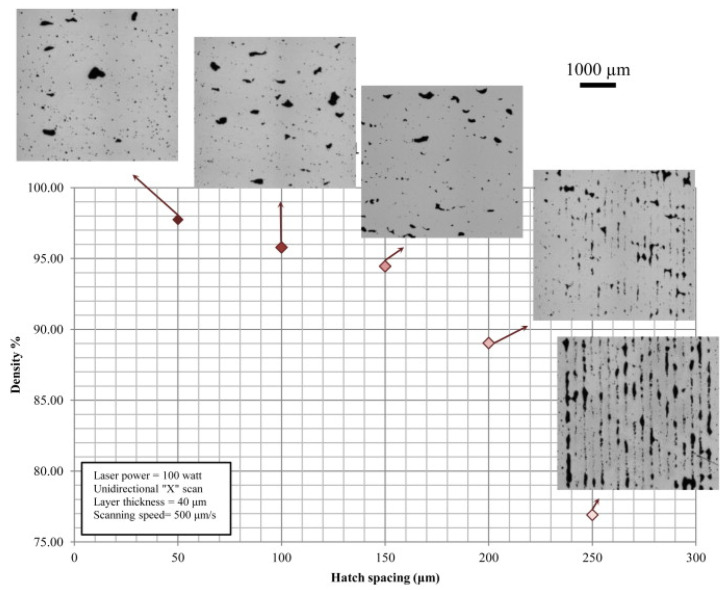
Evolution of pores in AlSi10Mg processed by SLM as a function of varying hatch spacing [76]. Copyright 2014. Adapted from Elsevier Science Ltd. (Figure 4 [76]).

**Figure 39 materials-13-04564-f039:**
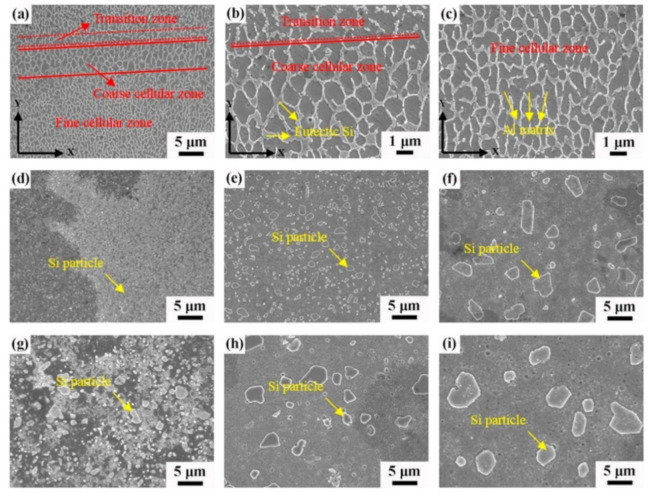
SEM images of AlSi10Mg SLM specimens after etching with Keller’s reagent (**a**) Single track melt (**b**) hatch boundary and (**c**) core of the hatch with fine cellular microstructure. SEM images of the AlSi10Mg microstructure after heat treatment under different conditions (**d**) 723 K for 2 h (**e**) 773 K for 2 h (**f**) 823 K for 2 h (**g**) 723 K for 2 h + 453 K for 12 h (**h**) 773 K for 2 h + 453 K for 12 h and (**i**) 823 K for 2 h + 453 K for 122 h [160]. Copyright 2016. Adapted with permission from Elsevier Science Ltd. under the license number 4803740322935 (Figure 3 [160]), dated 7 April 2020.

**Figure 40 materials-13-04564-f040:**
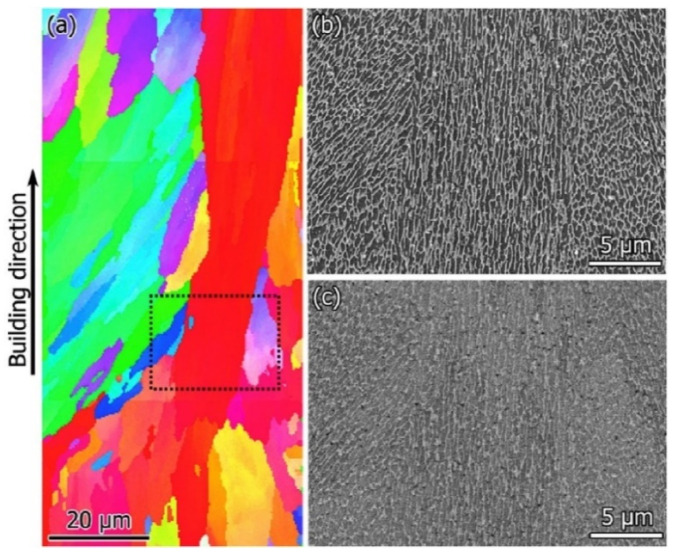
SEM images of AlSi10Mg SLM samples along the YZ plane: (**a**) electron back scattered diffraction, (**b**) secondary electron image and (**c**) back-scattered image. The inset in (**a**) shows the area in the electron back-scattered diffraction image, where the secondary and back-scattered images are taken [172]. Copyright 2016. Adapted with permission from Elsevier Science Ltd. under the license number 4803740596529 (Figure 1 [172]), dated 7 April 2020.

**Figure 41 materials-13-04564-f041:**
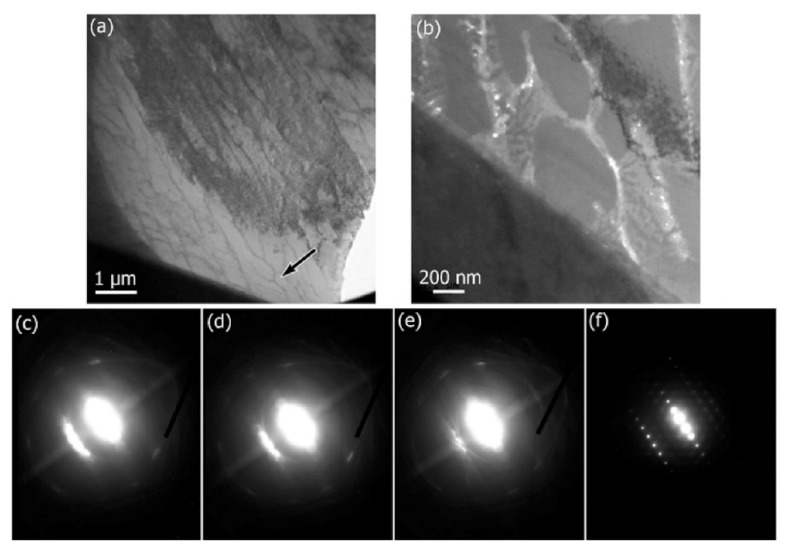
TEM images of AlSi10Mg SLM samples along YZ plane taken using Al-reflection: (**a**) bright-field image and (**b**) dark-field image. The arrow in (**a**) indicates the region from which the dark field image (**b**) was obtained. The corresponding micro-diffraction patterns are shown in (**c**–**f**). The images in (**c**,**d**) correspond to the area of the cells on either side of the cell boundary and (**e**) is directly on the cell boundary. (**f**) Shows a diffraction pattern covering the two cells with the cell boundary [172]. Adapted with permission from Elsevier Science Ltd. under the license number 4803740596529 (Figure 2 [172]), dated 7 April 2020.

**Figure 42 materials-13-04564-f042:**
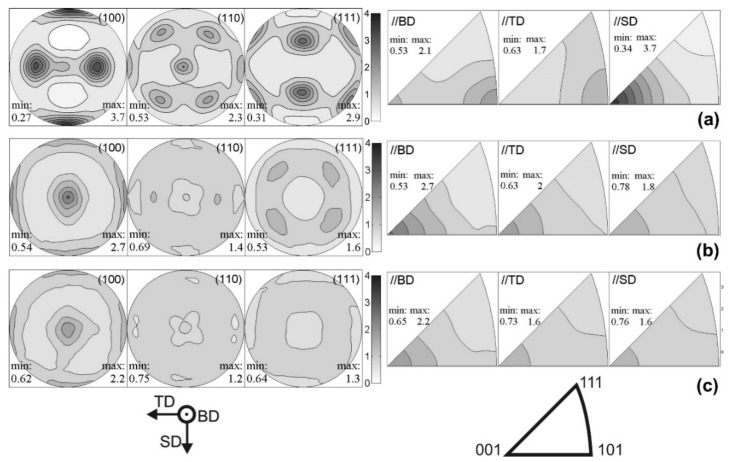
AlSi10Mg SLM samples showing pole and inverse pole figures as a function of different scanning strategy. The building direction (BD) is correlated to the (1 0 0) pole figures and the scanning direction (SD) and the transverse direction (TD) are correlated to the (1 1 0) and (1 1 1) pole figure data. The figures correspond to: (**a**) unidirectional long scans, (**b**) bidirectional long scans with the vectors rotated to 90° and (**c**) an island scanning strategy without shift. The orientation of the coordinate system is displayed by the relative intensity of the diffraction peaks compared to the reference sample indicated in grey scale [71]. Copyright 2012. Adapted with permission from Elsevier Science Ltd. under the license number 4803740839232 (Figure 7 [71]), dated 7 April 2020.

**Figure 43 materials-13-04564-f043:**
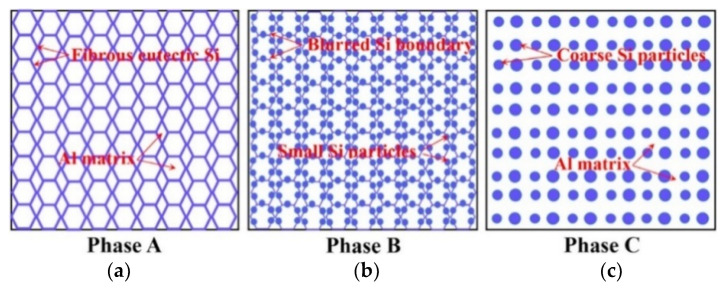
Schematic illustration showing the microstructure evolution of SLM AlSi10Mg samples in the (**a**) as-prepared state, (**b**) after solutionizing treatment and (**c**) after artificial ageing heat treatment. Si-rich areas are marked with blue color, whereas white matrix represents the Al-rich areas [160]. Adapted with permission from Elsevier Science Ltd. under the license number 4803740322935 (Figure 5 [160]), dated 7 April 2020.

**Figure 44 materials-13-04564-f044:**
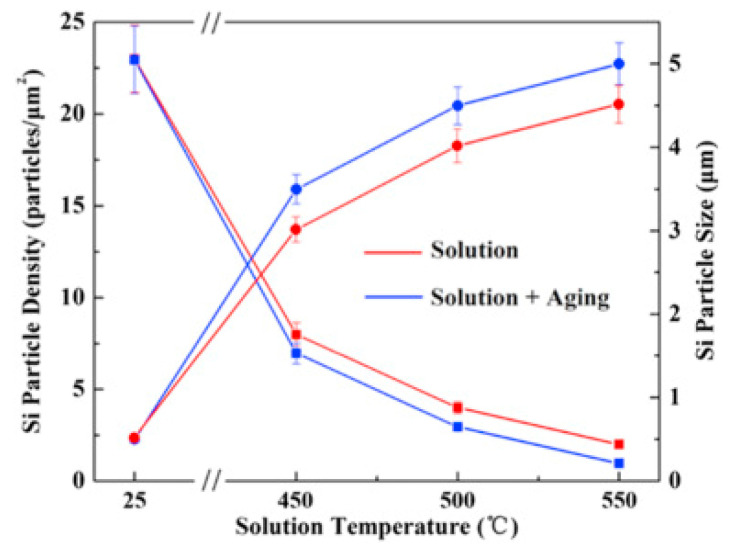
Plot showing the Si particle density and size as a function of annealing treatment for AlSi10Mg SLM samples, calculated from scanning electron microscopy images [160]. Adapted with permission from Elsevier Science Ltd. under the license number 4803740322935 (Figure 5 [76]), dated 7 April 2020.

**Figure 45 materials-13-04564-f045:**
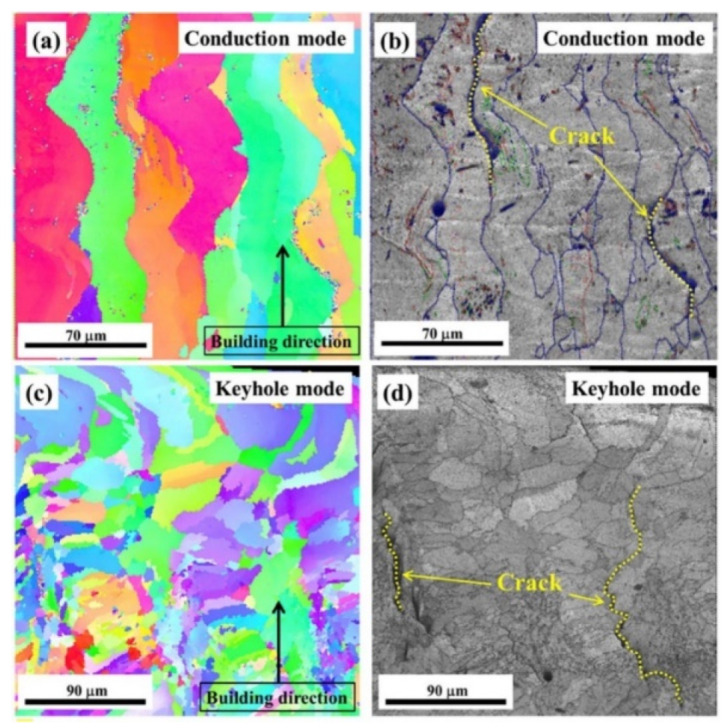
EBSD maps for two melting modes. (**a**) orientation image map and (**b**) grain boundary misorientation angle map in conduction mode, (**c**) orientation image map and (**d**) grain boundary misorientation angle map in keyhole mode [184]. Copyright 2017. Adapted with permission from Elsevier Science Ltd. under the license number 4803741018430 (Figure 10 [184]), dated 7 April 2020.

**Figure 46 materials-13-04564-f046:**
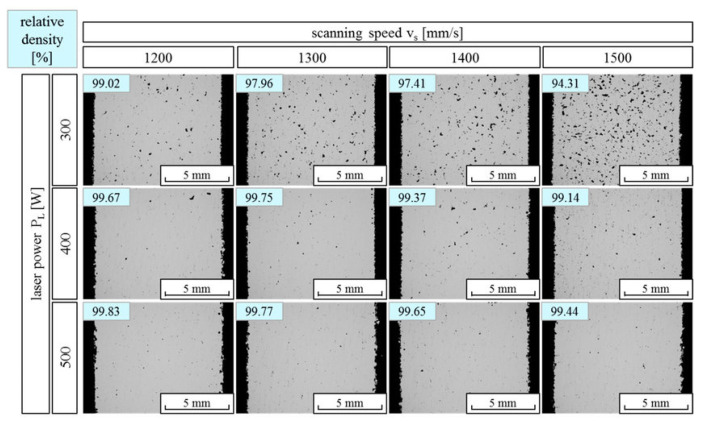
Comparison of SEM images of polished SLM 7075 alloy specimens, presented along with relative densities (top left of the images) obtained for different combinations of laser power and scanning speed [182]. Copyright 2016. Adapted with permission from Elsevier Science Ltd. under the license number 4803741261901 (Figure 5 [182]), dated 7 April 2020.

**Figure 47 materials-13-04564-f047:**
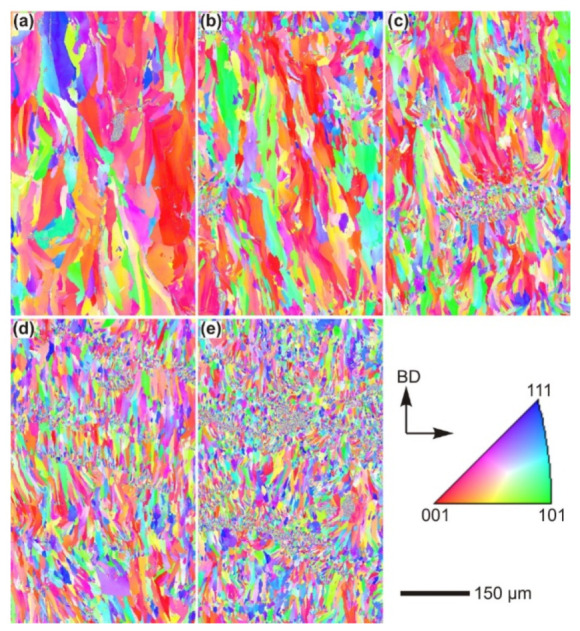
EBSD orientation maps of samples with different concentrations of Si: 0 wt.% (**a**), 1 wt.% (**b**), 2 wt.% (**c**), 3 wt.% (**d**) and 4 wt.% (**e**). The observed plane is parallel to the building direction. The crystallographic orientation is represented by the inverse pole figure for aluminium [95]. Copyright 2016. Adapted with permission from Elsevier Science Ltd. under the license number 4803741466657 (Figure 5 [95]), dated 7 April 2020.

**Figure 48 materials-13-04564-f048:**
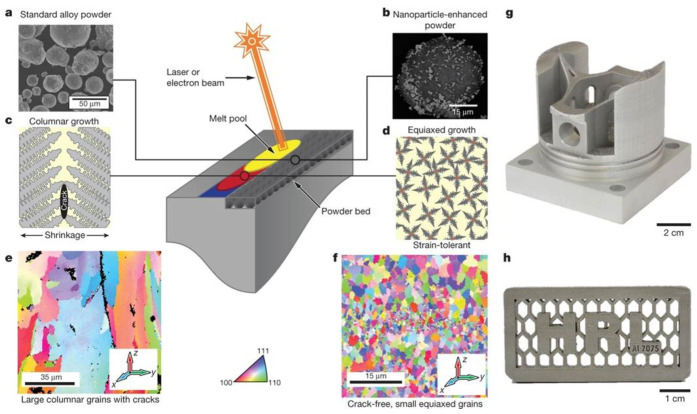
The central schematic presents an overview of the additive manufacturing process, whereby a directed energy source (laser or electron beam) melts a layer of metal powder (yellow), which solidifies (red to blue), fusing it to the previous (underlying) layer of metal (grey). (**a**) Conventional Al7075 powder feedstock. (**b**) Al7075 powder functionalized with nanoparticles. (**c**) Many alloys including Al7075 tend to solidify by columnar growth of dendrites, resulting in cracks due to solidification shrinkage. (**d**) Suitable nanoparticles can induce heterogeneous nucleation and facilitate equiaxed grain growth, thereby reducing the effect of solidification strain. (**e**) Many alloys exhibit intolerable microstructures with large grains and periodic cracks when 3D-printed using conventional approaches, as illustrated by the inverse pole figure. (**f**) Functionalizing the powder feedstock with nanoparticles produces fine equiaxed grain growth and eliminates hot cracking. (**g**) A 3D-printed, topologically optimized Al6061 piston on the build plate. (**h**) 3D-printed Al7075 HRL logo [185]. Copyright 2017. Adapted with permission from Springer Nature under the license number 4605280085008 (Figure 1 [185]), dated 7 April 2020.

**Figure 49 materials-13-04564-f049:**
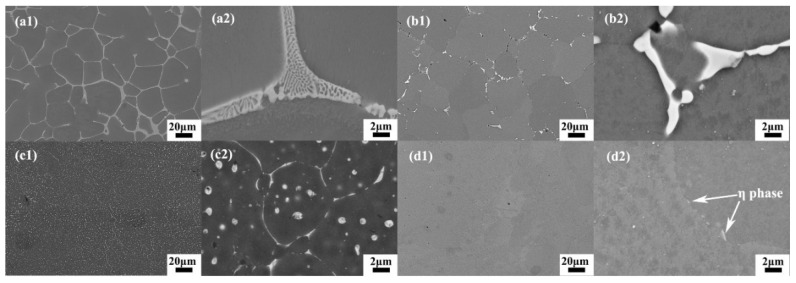
SEM images of: (**a**) Cast, (**b**) T6 heat-treated cast sample, (**c**) as-prepared SLM and (**d**) T6 heat treated SLM samples [183]. Copyright 2016. Adapted with permission from Elsevier Science Ltd. under the license number 4803750252863 (Figure 2 [183]), dated 7 April 2020.

**Figure 50 materials-13-04564-f050:**
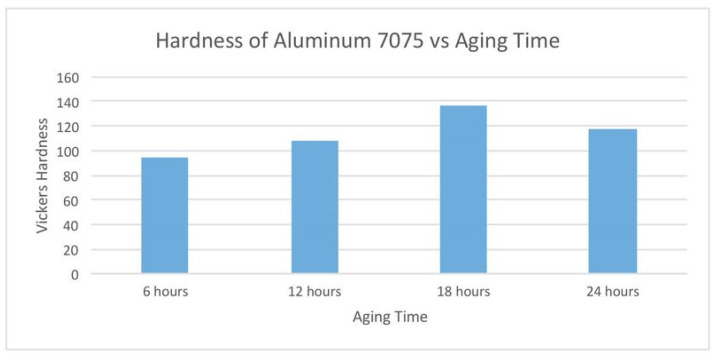
Variation of hardness of the SLM Al7075 alloy with aging duration [185]. Copyright 2017. Adapted with permission from Springer Nature. under the license number 4605280085008, dated 10 June 2019.

**Figure 51 materials-13-04564-f051:**
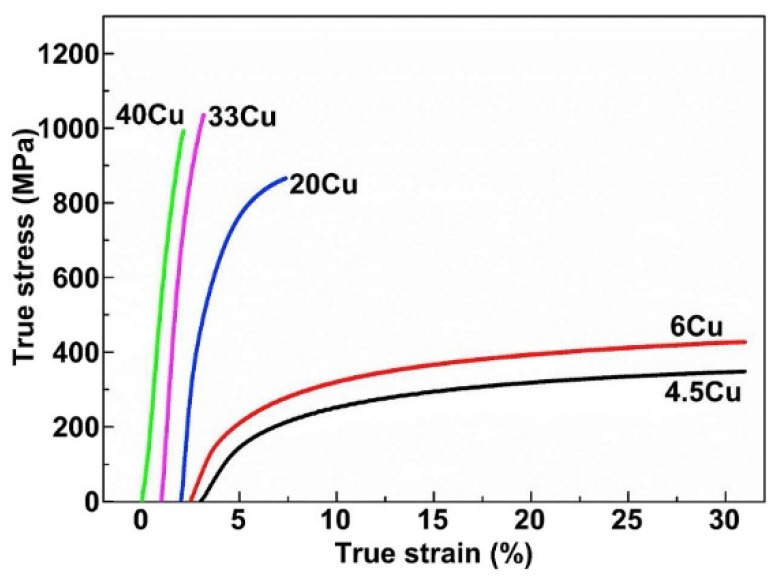
Compressive stress-strain curves of SLM Al-xCu alloys [115]. Copyright 2017. Adapted with permission from Elsevier Science Ltd. under the license number 4803750369772 (Figure 4 [115]), dated 7 April 2020.

**Figure 52 materials-13-04564-f052:**
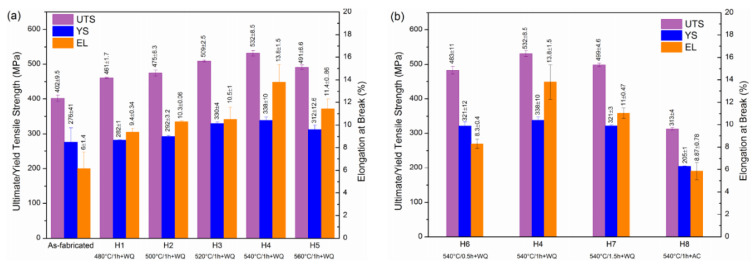
Mechanical properties of the SLM Al-Cu-Mg alloy in as-fabricated condition and after different heat treatments. WQ and AC refer to water quenching and air-cooling, respectively. The mean values and standard deviations are listed above the columns in the figure [117]. (**a**) as-fabricated and H1-H5 heat treated conditions and (**b**) H4, H6, H7 and H8 heat treated conditions. Copyright 2016. Adapted permission from SPIE publishers (Figure 3 [117]).

**Figure 53 materials-13-04564-f053:**
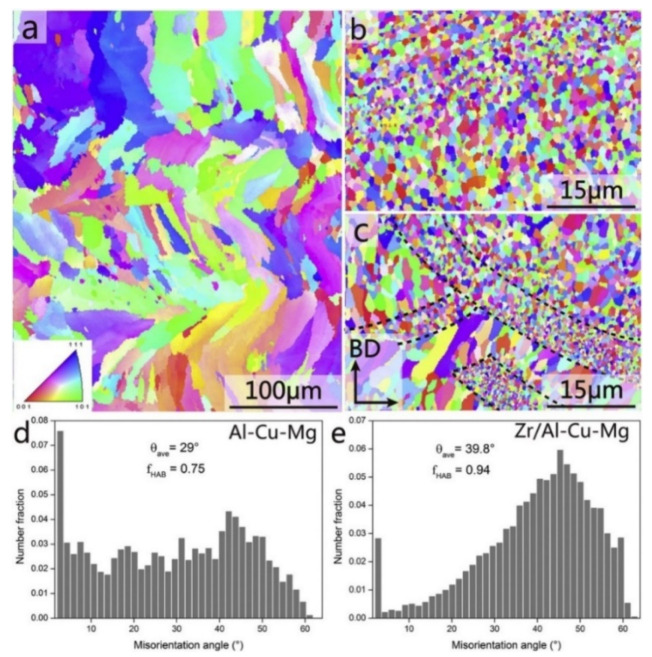
EBSD inverse pole figure (IPF) maps of Al-Cu-Mg fabricated at a scanning speed of 5 m/min (**a**) and Zr/Al-Cu-Mg sample fabricated at a scanning speed of 5 m/min (**b**) and 15 m/min (**c**), respectively. (**d**,**e**) misorientation angle distribution [190]. Copyright 2017. Adapted with permission from Elsevier Science Ltd. under the license number 4803750527871 (Figure 4 [190]), dated 7 April 2020.

**Figure 54 materials-13-04564-f054:**
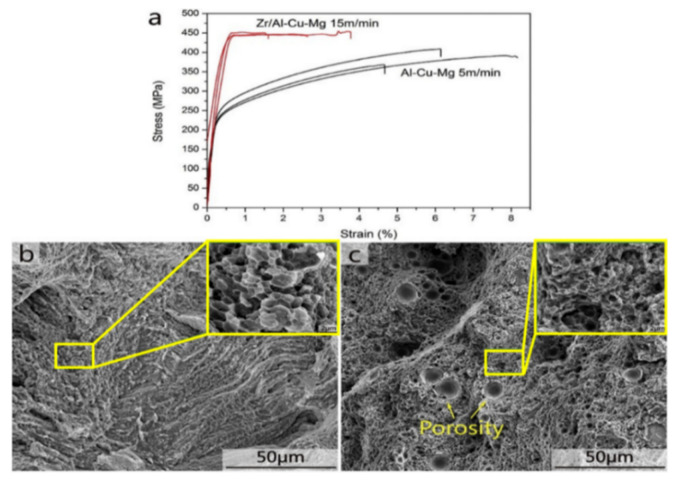
(**a**) Stress-Strain curves of the Al-Cu-Mg and Zr/Al-Cu-Mg samples fabricated at different scanning speeds. SEM images showing typical morphologies of the fracture surfaces of the Al-Cu-Mg sample fabricated at a scanning speed of 5 m/min (**b**) and for a Zr/Al-Cu-Mg sample fabricated at a scanning speed of 15 m/min (**c**), respectively. The insets show magnified images of the areas enclosed in yellow squares [190]. Copyright 2017. Adapted with permission from Elsevier Science Ltd. under the license number 4595281038046 (Figure 4 [190]), dated 7 April 2020.

**Figure 55 materials-13-04564-f055:**
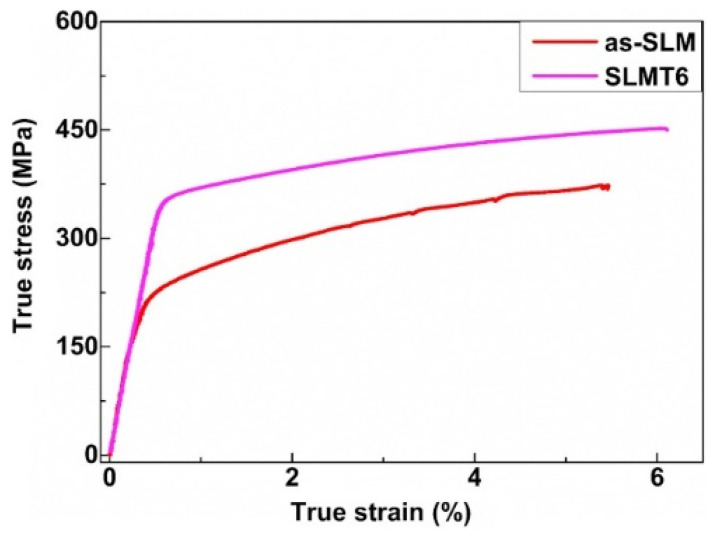
Tensile stress-strain curves of as-SLM and SLM T6 Al-3.5Cu-1.5Mg-1Si specimens [191]. Copyright 2017. Adapted with permission from Elsevier Science Ltd. under the license number 4803750779574 (Figure 8 [191]), dated 7 April 2020.

**Figure 56 materials-13-04564-f056:**
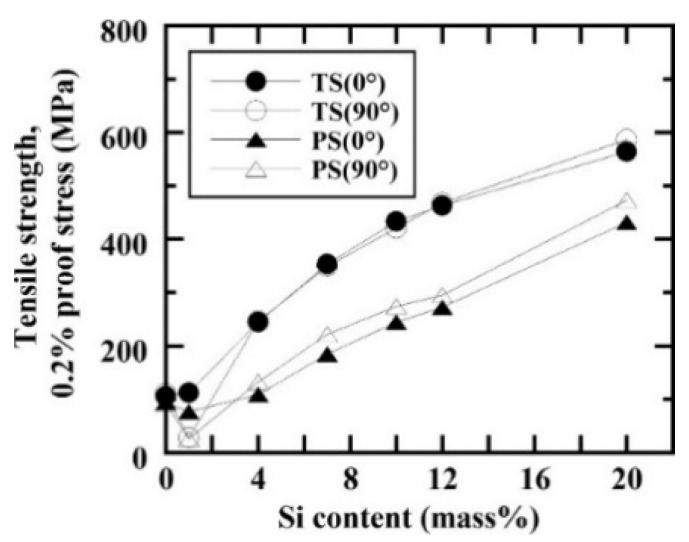
Tensile strength (TS) and proof stress (PS) of Al-xSi SLM samples fabricated using the optimal laser scanning parameters [136]. Copyright 2016. Adapted with permission from Elsevier Science Ltd. under the license number 4803760249108 (Figure 11 [136]), dated 7 April 2020.

**Figure 57 materials-13-04564-f057:**
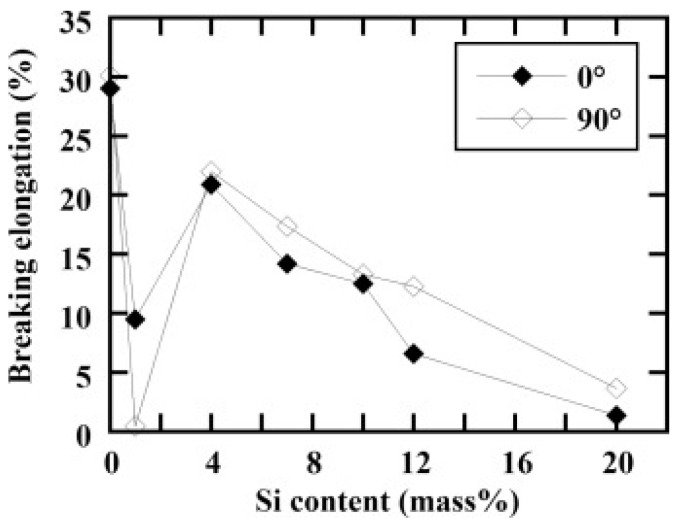
Fracture elongation of Al-xSi SLM samples fabricated using optimal laser processing parameters [136]. Copyright 2016. Adapted with permission from Elsevier Science Ltd. under the license number 4803760249108 (Figure 12 [136]), dated 7 April 2020.

**Figure 58 materials-13-04564-f058:**
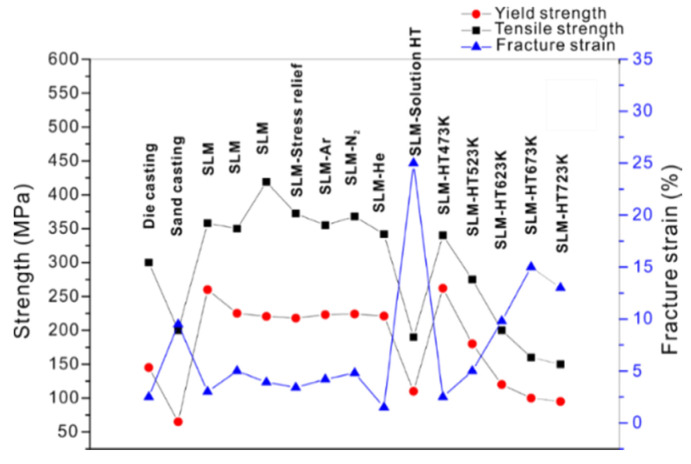
Comparison of the tensile properties of Al-12Si alloy produced by die-casting, sand casting and various SLM processes with/without heat treatments.

**Figure 59 materials-13-04564-f059:**
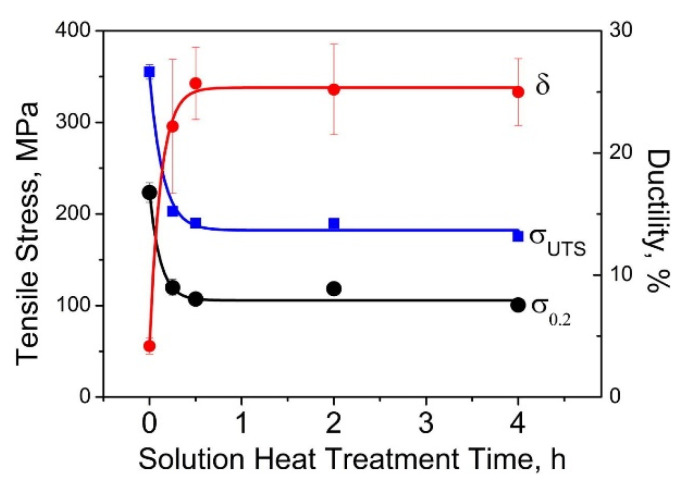
Variation in the mechanical properties of the Al-12Si alloy upon solution heat treatment for different durations. *δ*, *σ*_0.2_ and *σ_UTS_* represent the ductility, yield strength and ultimate tensile strength, respectively [130]. Copyright 2015. Adapted with permission from Elsevier Science Ltd. under the license number 4803760532619 (Figure 9 [130]), dated 7 April 2020.

**Figure 60 materials-13-04564-f060:**
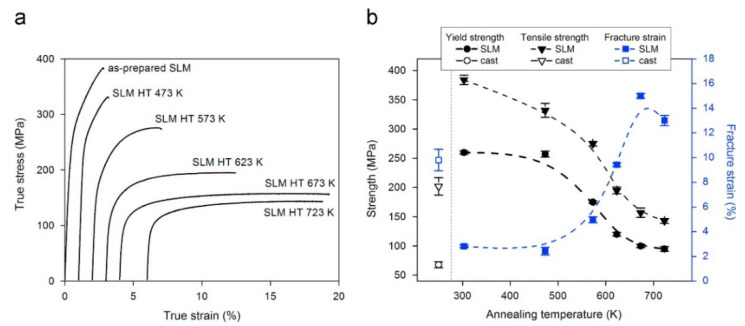
(**a**) Room temperature tensile test curves of SLM Al-12Si samples (γ = 90°) annealed at different temperatures and (**b**) corresponding mechanical data [67]. Copyright 2013. Adapted with permission from Elsevier Science Ltd. under the license number 4803760682810 (Figure 6 [67]), dated 7 April 2020.

**Figure 61 materials-13-04564-f061:**
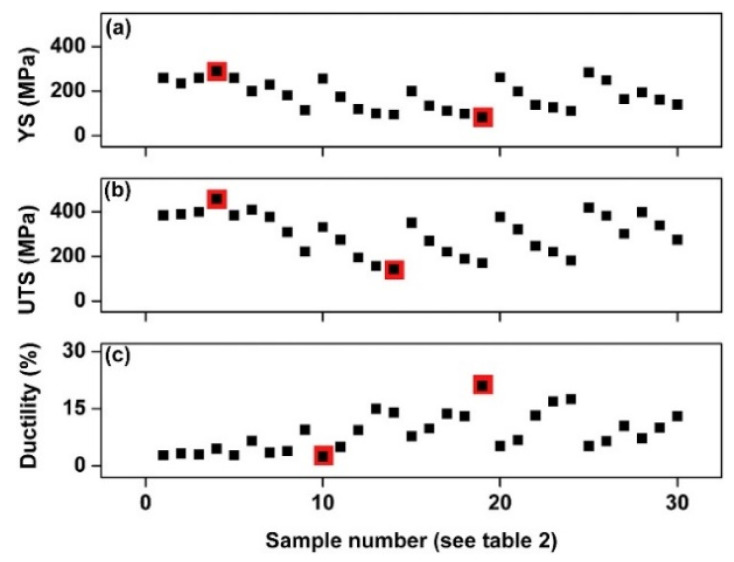
Tensile properties of Al-12Si SLM samples produced under different conditions with or without external heat treatment: (**a**) yield stress (YS), (**b**) ultimate tensile stress (UTS) and (**c**) ductility. Each square represents a data point and both maximum and minimum values were highlighted with red color [157]. Copyright 2016. Adapted with permission from Elsevier Science Ltd. under the license number 4803760806634 (Figure 13 [157]), dated 7 April 2020.

**Figure 62 materials-13-04564-f062:**
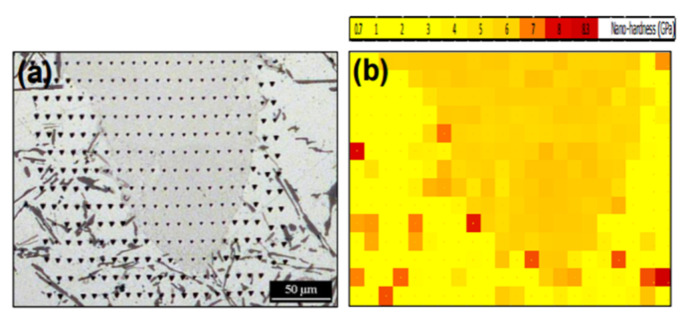
(**a**) SEM image of AlSi10Mg SLM sample showing an array of nanoindents across the melt pool and (**b**) corresponding nanohardness map [196]. Copyright 2016. Adapted as per the open access policy of VBRI press (Figure 3 [196]).

**Figure 63 materials-13-04564-f063:**
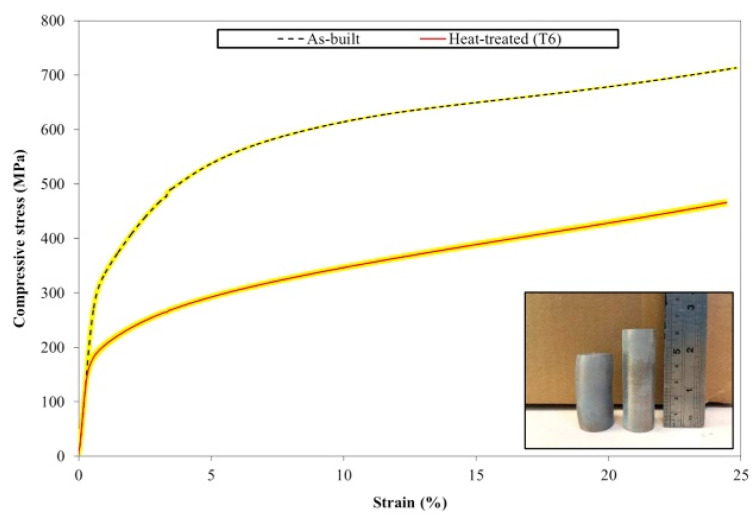
Compressive behavior of as-prepared AlSi10Mg SLM samples with and without annealing treatment. The curves are average of three experimental data sets and the standard deviation is highlighted in yellow. The inset in the figure shows AlSi10Mg sample before and after compression test [169]. Copyright 2016. Adapted with permission from Elsevier Science Ltd. under the license number 4803761103240 (Figure 6 [169]), dated 7 April 2020.

**Figure 64 materials-13-04564-f064:**
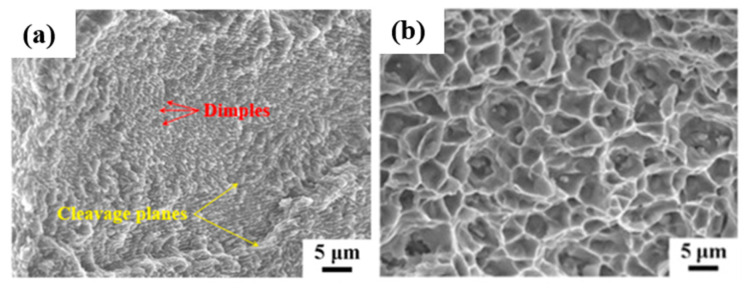
Fracture surfaces of AlSi10Mg SLM samples after tensile testing in the (**a**) as-prepared and (**b**) annealed (823 K, 2 h) condition [160]. Copyright 2016. Adapted with permission from Elsevier Science Ltd. under the license number 4803740322935 (Figure 8 [160]), dated 7 April 2020.

**Figure 65 materials-13-04564-f065:**
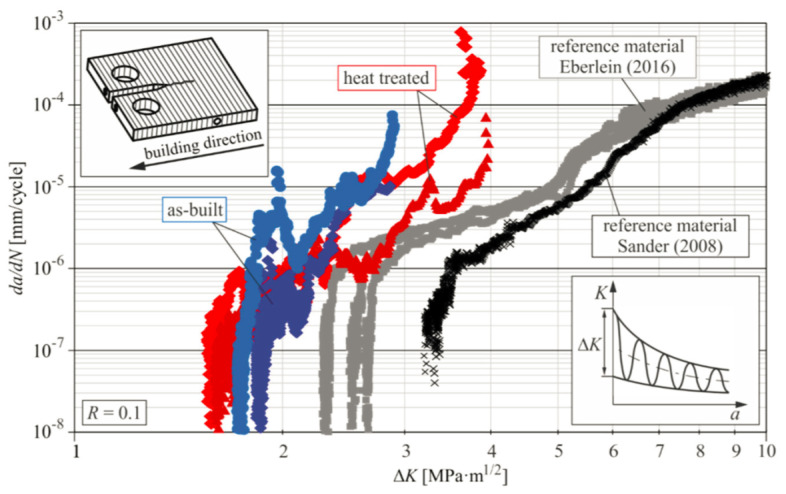
Fatigue crack growth curves for SLM-processed EN AW-7075 in different conditions. The crack plane and crack growth are parallel to the building direction. Data for conventionally processed reference material by Sander (2008) and Eberlein (2016) are displayed in black and grey color [204]. Copyright 2016. Adapted with permission from Elsevier Science Ltd. under the license number 4803761334383 (Figure 4 [204]), dated 7 April 2020.

**Figure 66 materials-13-04564-f066:**
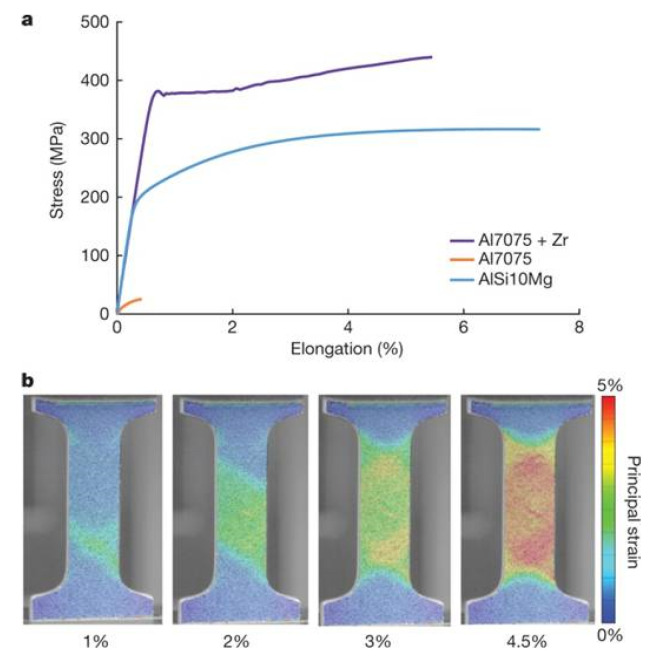
Mechanical testing of 3D-printed aluminum alloys: (**a**) Representative tensile curves of the 3D-printed materials. (**b**) Representative deformation behavior of Al7075+Zr, indicating Lüders band propagation due to the refined grain size. The color scale shows the local principal strain, with the total elongation listed under each panel [185]. Copyright 2017. Adapted with permission from Springer Nature, under the license number 4605280085008 (Figure 4 [185]), dated 7 April 2020.

**Figure 67 materials-13-04564-f067:**
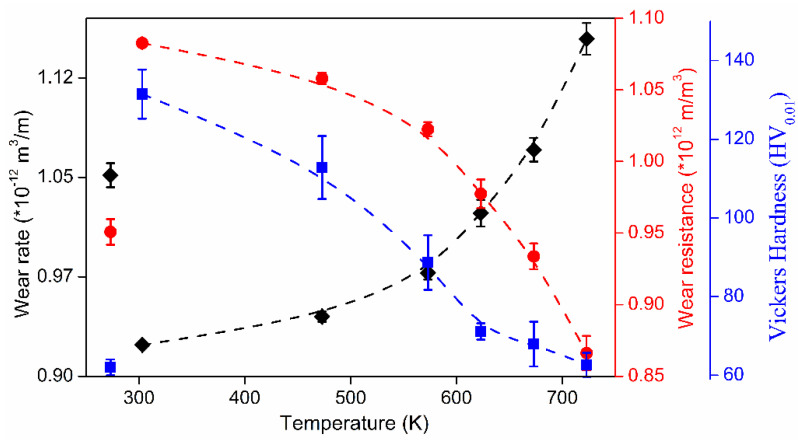
Sliding wear rate, wear volume and the Vickers hardness (HV) for Al-12Si cast, as-prepared SLM (300 K) and SLM samples annealed at different temperatures [205]. Copyright 2014. Adapted with permission from Cambridge University Press under the license number 4605401109358 (Figure 2 [205]), dated 24 May 2019.

**Figure 68 materials-13-04564-f068:**
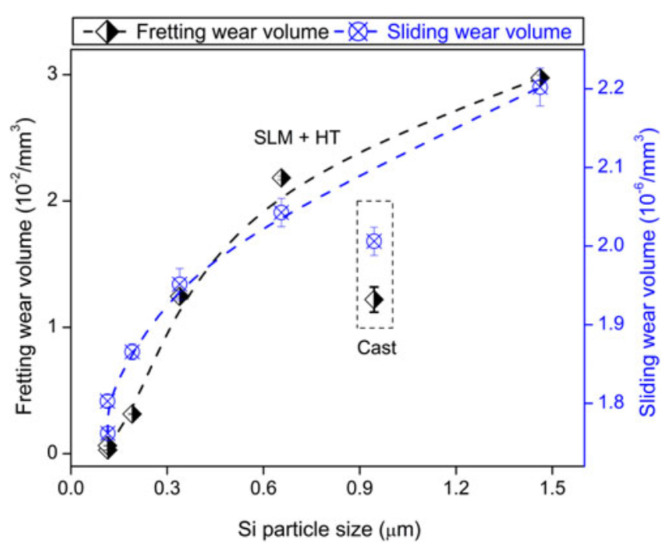
Fretting and sliding wear volumes for the as-cast and SLM Al-12Si specimens as a function of the average Si particle size [205]. Copyright 2014. Adapted with permission from Cambridge University Press under the license number 4605401109358 (Figure 5 [205]), dated 24 May 2019.

**Figure 69 materials-13-04564-f069:**
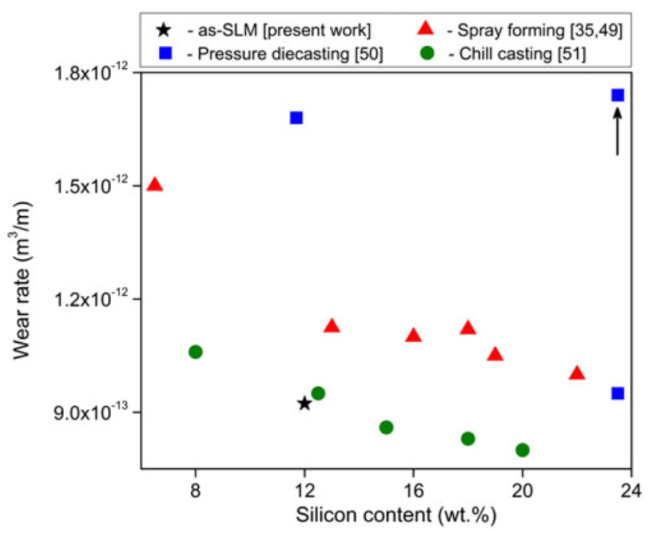
Wear rates of Al-Si alloys produced by different techniques as a function of the Si content [205]. Copyright 2014. Adapted with permission from Cambridge University Press under the license number 4605401109358 (Figure 11 [205]), dated 24 May 2019.

**Figure 70 materials-13-04564-f070:**
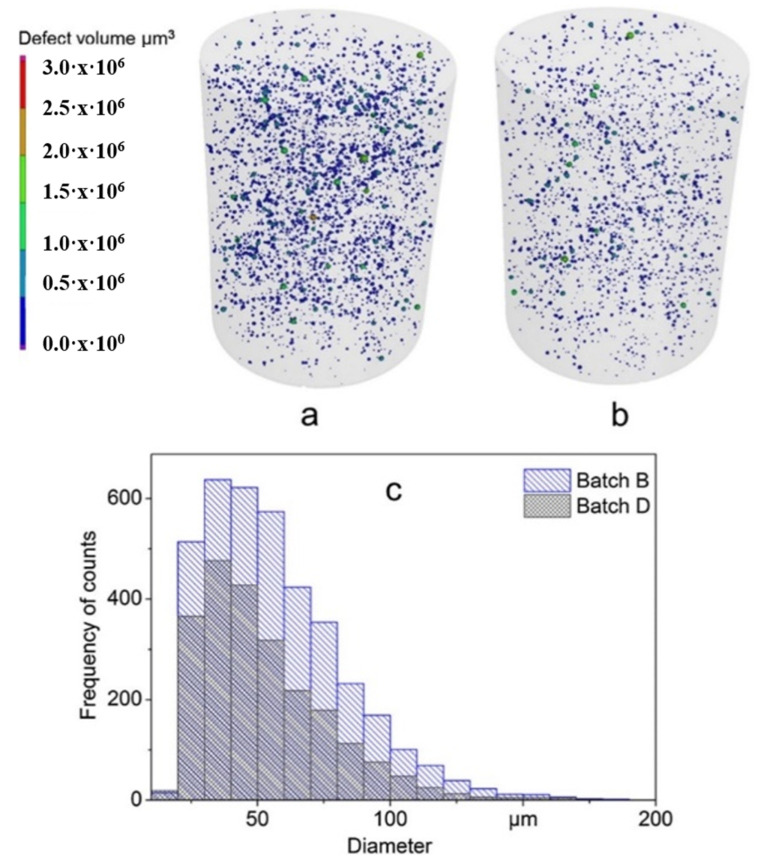
Micro-computed tomography images of Al-12Si alloy with two different batches. (**a**) batch B: 0.25% porosity (**b**) batch D: 0.12% porosity and (**c**) histograms showing the distribution of defects in both batches [122]. Copyright 2016. Adapted with permission from Elsevier Science Ltd. under the license number 4803770892666 (Figure 8 [122]), dated 7 April 2020.

**Figure 71 materials-13-04564-f071:**
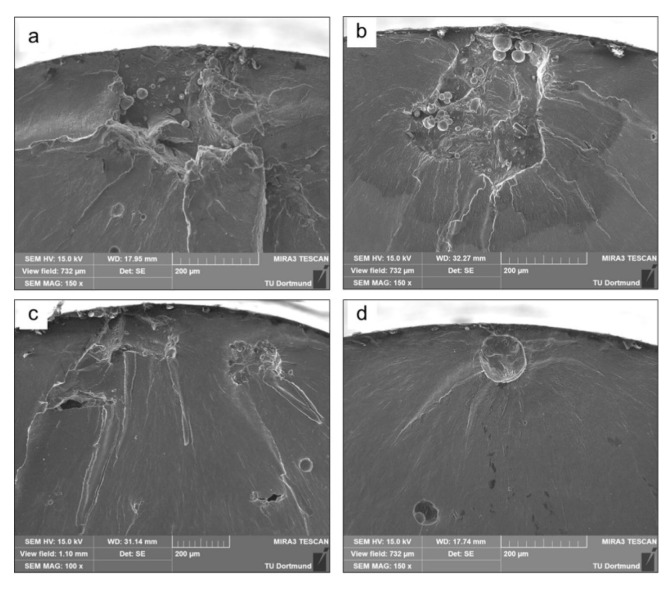
Fatigue crack initiation at 70 MPa for batch B (Al-12Si) with resulting fatigue life 1.6 × 10^6^ cycles (**a**); 2.6 × 10^6^ cycles (**b**); 3.6 × 10^6^ cycles (**c**); and 3.2 × 10^8^ cycles (**d**) [122]. Copyright 2016. Adapted with permission from Elsevier Science Ltd. under the license number 4803770892666 (Figure 12 [122]), dated 7 April 2020.

**Figure 72 materials-13-04564-f072:**
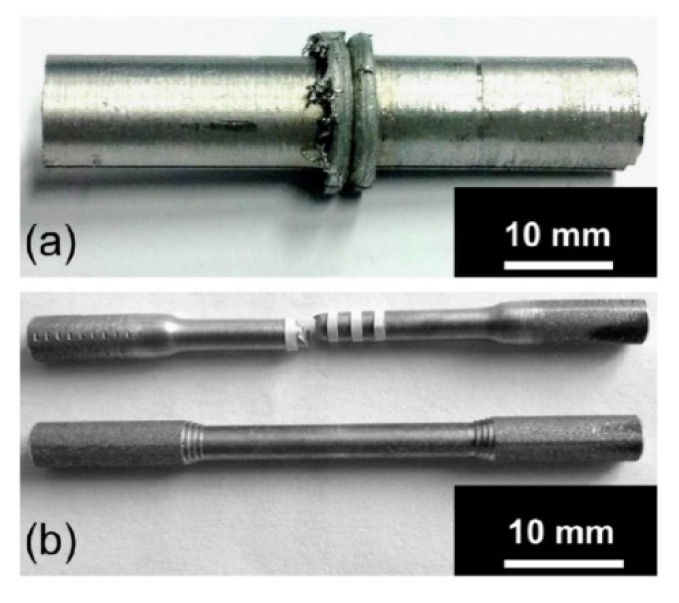
(**a**) Typical image of a friction welded Al-12Si joint with symmetrical and smooth flash at the joint. (**b**) Examples of tensile specimens machined from the welded samples [216]. Copyright 2014. Adapted with permission from Elsevier Science Ltd. under the license number 4803771284201 (Figure 1 [216]), dated 7April 2020.

**Figure 73 materials-13-04564-f073:**
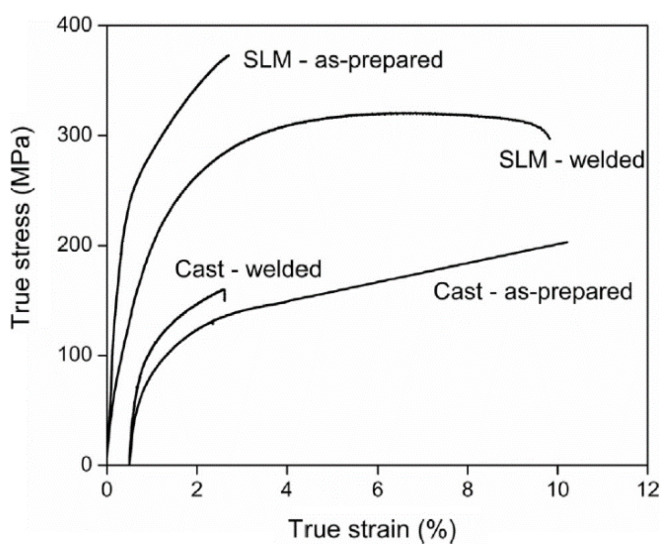
Room temperature tensile curves of the cast and SLM specimens tested in the as-prepared and welded conditions [216]. Copyright 2014. Adapted with permission from Elsevier Science Ltd. under the license number 4803771284201 (Figure 6 [216]), dated 7 April 2020.

**Figure 74 materials-13-04564-f074:**
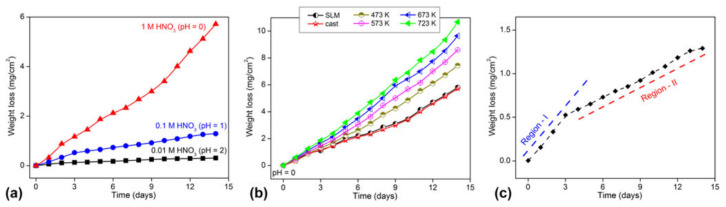
(**a**) Weight-loss curves for the as-prepared SLM Al-12Si samples as a function of the immersion time for three different HNO_3_ concentrations (0.01, 0.1, and 1 M). (**b**) Weight-loss plots for the as-prepared SLM, cast, and SLM heat-treated samples as a function of time for the 1 M HNO_3_ solution. (**c**) Weight-loss curve (black line) for the as-prepared Al-12Si SLM samples for 0.1 M HNO3, showing the different rates (rate 1—blue line and rate 2—red line) of corrosion as a function of the immersion time [205]. Copyright 2014. Adapted with permission from Cambridge University Press under the license number 4605401109358 (Figure 6 [205]), dated 24 May 2019.

**Figure 75 materials-13-04564-f075:**
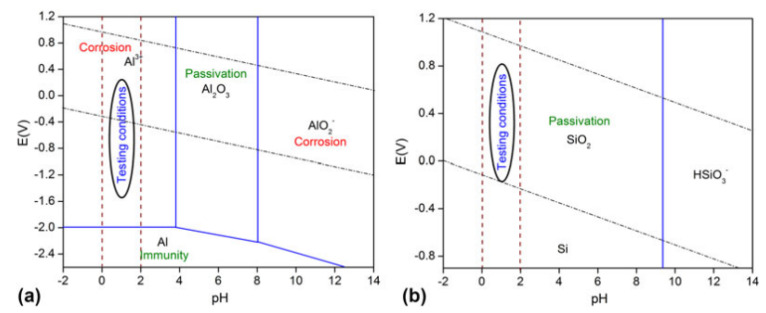
Pourbaix diagrams for (**a**) aluminum and (**b**) silicon, showing the regions of corrosion, immunity, and passivation [205]. Copyright 2014. Adapted with permission from Cambridge University Press under the license number 4605401109358 (Figure 12 [205]), dated 24 May 2019.

**Table 1 materials-13-04564-t001:** The families of Al-based alloys fabricated by SLM processing (with or without heat treatment).

Alloy Systems	Compositions	Yield Strength (MPa)	Tensile Strength (MPa)	Elongation/Plastic Strain (%)
1xxx	Pure Al	90	110	30
2xxx	Al-xCu; Al-Cu-Mg	205–250	313–532	5–14
4xxx	Al-xSi	83–263	150–420	1–27
6xxx	AlMgxSi	90–275	140–460	1.5–19
7xxx	Al-Zn-Mg-Cu(Zr)	225–373	255–417	0.5–5.4

**Table 2 materials-13-04564-t002:** Summary of the tensile properties of Al-xSi alloys. Results of the Al-12Si alloy will be reported in Table 3 separately.

Composition	Process	Post-Treatment	Orientation	Yield Strength (MPa)	Ultimate Strength (MPa)	Strain at Fracture (%)	Reference
Al-0Si	SLM	-	Vertical	100	110	30	[112]
Al-0Si	SLM	-	Horizontal	95	115	30	[112]
Al-0Si	SLM	723 K (10 min)	Vertical	80	100	27	[112]
Al-0Si	SLM	723 K (10 min)	Horizontal	85	115	25	[112]
Al-5Si	SLM	-	-	146	226	14	[66]
Al-5Si	SLM	T4	-	67	133	27	[66]
Al-7Si-Mg	SLM	-	-	193	320	5	[66]
Al-7Si-Mg	SLM	T4	-	109	204	17	[66]
Al-7Si-Mg	SLM	T6	-	227	273	10	[66]

**Table 3 materials-13-04564-t003:** Summary of the tensile properties of Al-12Si alloy.

Composition	Process	Condition	Yield Strength (MPa)	Ultimate Strength (MPa)	Strain at Facture (%)	Reference
Al-12Si	Casting	As-prepared	145	300	2.5	[196]
Al-12Si	Cast	As-prepared	65	200	9.5	[67]
Al-12Si	SLM	N_2_ atmosphere	224 ± 7	368 ± 11	4.8 ± 0.6	[197]
Al-12Si	SLM	Ar atmosphere	223 ± 11	355 ± 8	4.2 ± 0.6	[197]
Al-12Si	SLM	He atmosphere	221 ± 11	342 ± 43	1.5 ± 0.6	[197]
Al-12Si	SLM	As-prepared	260	385	3	[67]
Al-12Si	SLM	As-prepared	225	350	5	[130]
Al-12Si	SLM	As-prepared	184 ± 9	231 ± 5	1 ± 01	[135]
Al-12Si	SLM	As-prepared	221 ± 10	420 ± 10	4 ± 0.3	[135]
Al-12Si	SLM	As-prepared	154 ± 5	190 ± 3	1 ± 0.1	[135]
Al-12Si	SLM	As-prepared	221 ± 10	419 ± 10	4 ± 0.3	[135]
Al-12Si	SLM	Post-build stress relief—513 K	180 ± 7	220 ± 6	1 ± 0.1	[135]
Al-12Si	SLM	Post-build stress relief—513 K	187 ± 3	230 ± 4	1 ± 0.1	[135]
Al-12Si	SLM	Post-build stress relief—513 K	218 ± 7	372 ± 7	3 ± 0.3	[135]
Al-12Si	SLM	Heat treated—473 K	262	340	2.5	[67]
Al-12Si	SLM	Heat treated—523 K	180	275	5	[67]
Al-12Si	SLM	Heat treated—623 K	120	200	10	[67]
Al-12Si	SLM	Heat treated—673 K	100	160	15	[67]
Al-12Si	SLM	Heat treated—723 K	95	150	13	[67]
Al-12Si	SLM	Solution heat treated (30 min)	110	190	25	[130]

**Table 4 materials-13-04564-t004:** Summary of the tensile properties of AlSi10Mg samples.

Process	Condition	Yield Strength (MPa)	Ultimate Strength (MPa)	Strain at Fracture (%)	Reference
Wrought	As-prepared (longitudinal)	293	310	15	[201]
Cast	Aged	-	300–371	2.5–3.5	[197]
HDPC *	As-prepared	-	300–350	2.5–3.5	[197]
HDPC *	T6	-	330–365	3–5	[199]
SLM	As-prepared	208	368	4	[66]
SLM	As-prepared hotizontal	227	358	4	[201]
SLM	As-prepared vertical	172	289	3	[203]
SLM	Building angle—0°	-	420	-	[202]
SLM	Building angle—45°	-	405	-	[202]
SLM	Building angle—90°	-	360	-	[204]
SLM	Horizontal	250	330	1	[163]
SLM	Vertical	240	320	1	[163]
SLM	XY	-	391 ± 8	5.5 ± 0.4	[163]
SLM	Z	-	396 ± 8	3 ± 1	[163]
SLM	As-prepared	240	360	3	[163]
SLM	As-prepared	268 ± 2	333 ± 2	1.4 ± 0.3	[166]
SLM	As-prepared	332 ± 8	434 ± 11	5.3 ± 0.2	[160]
SLM	T4	119	212	12	[163]
SLM	T6	210	269	10	[66]
SLM	Solutionized	196 ± 4	282 ± 4	13.5 ± 0.5	[156]
SLM	T6	187 ± 3	197 ± 4	20 ± 1	[163]
SLM	T6	239 ± 2	292 ± 4	4 ± 1	[169]

* High pressure die casting.

**Table 5 materials-13-04564-t005:** Mechanical properties of SLM 7075 alloys under different loading conditions.

Composition	Condition	Load Direction	Yield Strength (MPa)	Ultimate Strength (MPa)	Elongation (%)	Reference
EN AW7075	As-built	Parallel to building direction	NA	203 ± 12	0.5 ± 0.2	[204]
EN AW7075	As-built	Perpendicular to building direction	NA	42 ± 8	0.5 ± 0.3	[204]
EN AW7075	Heat treated	Parallel to building direction	NA	206 ± 26	0.5 ± 0.1	[204]
EN AW7075	Heat treated	Perpendicular to building direction	NA	45 ± 1	0.2 ± 0.1	[204]
7075	T6	-	NA	26	0.4	[185]
7075+Zr	T6	-	325–373	383–417	4–6	[185]

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
