# Peer review of "Selective Laser Melting of Aluminum and Its Alloys"

_materials, 2020, doi:10.3390/ma13204564_

Round 1

Reviewer 1 Report

The paper presents a comprehensive overview of the current state in the field of Selective Laser Melting of Aluminum and its Alloys. A wide team of renowned authors is a guarantee of the quality of the contribution.  

Some of my remarks, comments, recommendations and suggestions for improving the article can be found directly in the attached file.

I strongly recommend publishing the paper after correction of typing errors.

Author Response

We thank the reviewer for pointing out the errors. Accordingly, we have modified/corrected all the errors in the revised version of the manuscript.

Reviewer 2 Report

In this submission selective laser melting (SLM) of Aluminum and its alloys is observed. This technology is becoming more and more widespread, and review about its features is relevant. However, some minor remarks should be made:

  1. The measurement unit “mm/sec” should be corrected as “mm/s” overall review;
  2. In all figure captures the copywriting information should be specified (for example, figs. 1, 2, 6, 9 and so on);
  3. The Authors should corrected overall review text basing on ISO/ASTM 52900:2015

Author Response

We thank the reviewer for the comments. Accordingly,

(1) The measurement unit “mm/sec” should be corrected as “mm/s” overall review;

Mm/sec was changed to mm/s in the manuscript.

(2) In all figure captures the copywriting information should be specified (for example, figs. 1, 2, 6, 9 and so on);

The figure numbers are specified in figure captions as recommended by the reviewer.

(3) The Authors should corrected overall review text basing on ISO/ASTM 52900:2015

Where necessary the terms according to ISO/ASTM 52900:2015 are introduced.

Reviewer 3 Report

It can be a better paper if the authors make a standard format for this paper. This version needs a major revision and my comments are in below;

+I recommend you to add a table of content at the beginning of your paper. Then it would be easier to follow the content of your paper.

+Page 2-line 68: Is not it better to make a graphic picture and address the important parameters there?

+Introduction: Your introduction about AM and the methods is very week. At least, you can add a table of various AM techniques, after that focus on SLM. You can also address the future of this process or application in industry 4.0. Check the following publications:

-Frazier, William E. "Metal additive manufacturing: a review." Journal of Materials Engineering and performance 23.6 (2014): 1917-1928.

-Gisario, Annamaria, et al. "Metal additive manufacturing in the commercial aviation industry: A review." Journal of Manufacturing Systems 53 (2019): 124-149.

-Mehrpouya, Mehrshad, et al. "The potential of additive manufacturing in the smart factory industrial 4.0: A review." Applied Sciences 9.18 (2019): 3865.

+Figure 1- This figure is not very necessary! You just need to improve your introduction as I mentioned in the previous comment.

+Figure 2: I do not believe that fig 2 describes the SLM process well.

+1.1.2. Drawbacks: What about the machine and material costs?!!!

+1.1.4. Process Parameters: isn't it better to show all important parameters in a figure/table and discuss them a bit?

+Figure 6: I cannot understand why figure 6 is defined in the energy density section? It is mostly about the layers and the application of AM to produce an object layer by layer! 

+It is very difficult to follow your sections and subsections in this paper! Why do not you use any graphs, diagrams,... to make it easier for the reader to understand your process, materials,...

+ I strongly recommend you to add a table for Al alloys for SLM process including the process parameters and cite the published papers for each items. This table is missing!

Author Response

It can be a better paper if the authors make a standard format for this paper. This version needs a major revision and my comments are in below;

We thank the reviewer for the comments and accordingly, we have undertaken major revision so as to satisfy all the comments from the reviewer.

+I recommend you to add a table of content at the beginning of your paper. Then it would be easier to follow the content of your paper.

Table of content is introduced as recommended by the reviewer.

+Page 2-line 68: Is not it better to make a graphic picture and address the important parameters there?

We agree with the reviewer and we have introduced a picture (Fig. 2) as recommended.

+Introduction: Your introduction about AM and the methods is very week. At least, you can add a table of various AM techniques, after that focus on SLM. You can also address the future of this process or application in industry 4.0. Check the following publications:

-Frazier, William E. "Metal additive manufacturing: a review." Journal of Materials Engineering and performance 23.6 (2014): 1917-1928.

-Gisario, Annamaria, et al. "Metal additive manufacturing in the commercial aviation industry: A review." Journal of Manufacturing Systems 53 (2019): 124-149.

-Mehrpouya, Mehrshad, et al. "The potential of additive manufacturing in the smart factory industrial 4.0: A review." Applied Sciences 9.18 (2019): 3865.

The introduction has been modified and different AM processes were introduced in the introduction section as suggested by the reviewer.

+Figure 1- This figure is not very necessary! You just need to improve your introduction as I mentioned in the previous comment.

We can understand the concern of the reviewer. However, to have a basic comparison between AM and SM, we have retained Fig. 1. However, we have modified the introduction, where more insights about different AM processes were introduced.

+Figure 2: I do not believe that fig 2 describes the SLM process well.

All the necessary obligations of the SLM process were depicted in Fig. 2. However, we understand that this is not complete, since additional figures were introduced later.

+1.1.2. Drawbacks: What about the machine and material costs?!!!

We have also introduced statements about machine and materials costs in the revised version.

+1.1.4. Process Parameters: isn't it better to show all important parameters in a figure/table and discuss them a bit?

As recommended by the reviewer a figure is introduced showing the difference process parameters.

+Figure 6: I cannot understand why figure 6 is defined in the energy density section? It is mostly about the layers and the application of AM to produce an object layer by layer! 

Since Fig. 6 depicts the formation of a 3D sample from hatches and hatch styles it is categorized under section 1.1.6 – Hatches and hatch styles.

+It is very difficult to follow your sections and subsections in this paper! Why do not you use any graphs, diagrams,... to make it easier for the reader to understand your process, materials,...

In order to make the reading process easy, contents were introduced so that an overview is already presented.

+ I strongly recommend you to add a table for Al alloys for SLM process including the process parameters and cite the published papers for each item. This table is missing!

I can understand the concern of the reviewer. However, SLM is a complex process, where the process parameters even used with the same machine may vary depending on the properties. However, there are several machine manufacturers. Tens and hundreds of groups work on Al-based alloys and marking the process parameters may not be useful and hence we refrain from making this recommendation.

Round 2

Reviewer 3 Report

My comments are mostly addressed. However, there are still some points that should be discussed better in this paper;

  • This paper is a review paper for SLM printing of Al alloy. The introduction is improved, however, I cannot find any explanation about the future of AM in specifically industry 4.0. I added some references to help you to improve this section of your paper (related to my previous comment)!
  • That would be better if you also discuss the future of the 3D printing market of totally metal 3D printing and particularly Al alloy if there is a published work on that (cost trend).
  • Any reference for Fig 3? Would not be better to apply a better tpr of diagram for that? 
  • You have investigated various type of Al alloys in both microstructure and mechanical properties sections. Why do not you define a section about the Al alloys in the printing process. Then you can compare all alloys together and compare their advantages, challenges, and applications together. That would be a very good source for readers and your paper will get very good citation as well.
  • The aim of a review paper is gather good and enough information for the readers. I see that you improve the introduction section related to AM processes, however there is missing a table or chart which can categorize the "3D printing of Al alloys" based on materials or technique. You should show something new in your review although you reporting the work of the others. How about the series of alloys that you have mentioned in the conclusion "Al-Cu-Mg, Al-Cu (2XXX), Al-Si (4XXX), Al-Si-Mg (6XXX) and Al-Zn-Mg-Cu (7XXX)". Maybe it can be helpful for your category!
  • I do not see any discussion about the applications and the market of this process! Please add it to your review paper. That would be really helpful.
  • I still believe the summary is weak! You need to discuss you conclusion as well as your TOC.

Author Response

Reviewer 3

My comments are mostly addressed. However, there are still some points that should be discussed better in this paper;

We thank the reviewer for the acknowledgment stating that most of the comments are addressed in the revised version of the paper.

This paper is a review paper for SLM printing of Al alloy. The introduction is improved, however, I cannot find any explanation about the future of AM in specifically industry 4.0. I added some references to help you to improve this section of your paper (related to my previous comment)!

We thank the reviewer for his comments. The aim of this review is to give a critical analysis of the different Al-based alloys that are processed using the selective laser melting process (including the processing difficulties, defect formation, etc.), followed by a detailed microstructural investigation and testing of properties. Hence, the introduction part of this review focuses on the SLM process (in detail) followed by the process parameters. The two publications suggested by the reviewer: Gisario et al. Metal additive manufacturing in the commercial aviation industry: A review J Manuf. Sys. 53 (2019) 124-149 and Mehrpouya et al. The potential of additive manufacturing in the smart factory industrial 4.0: A review details more on the use of metal additive manufacturing in the particular sector along with the commercialization of the process and the implementation of industry 4.0 concepts in the present metal additive manufacturing process. However the scope of these two mentioned review articles are different from the present article, we refrain from including any information regarding the future of AM, especially in the context of industry 4.0, and did not cite the two manuscripts. For readers who are interested in these field, they already have reviews (like the two mentioned by the reviewer) and hence we are not including these aspects in the present manuscript.

That would be better if you also discuss the future of the 3D printing market of totally metal 3D printing and particularly Al alloy if there is a published work on that (cost trend).

As mentioned above the present manuscript deal with the scientific information about the processing of Al-based alloys by the SLM process. In this context, the future of the 3D printing market and cost factors are out of the scope of the manuscript and we cannot include these factors, since this will alter the core of the manuscript.

Any reference for Fig 3? Would not be better to apply a better tpr of diagram for that? 

Figure 3 is self-made and so there is no reference for the same. If there exists any issues with Figure 3, we would ask the reviewer to point out the same so that we may modify them accordingly.

You have investigated various types of Al alloys in both microstructure and mechanical properties sections. Why do not you define a section about the Al alloys in the printing process? Then you can compare all alloys together and compare their advantages, challenges, and applications together. That would be a very good source for readers and your paper will get very good citation as well.

There are some reviews already published in the same context as described by the reviewer. We do not want our review article to be aligned with the published reviews and we want it to be unique and hence we have organized in a different way. So we feel it may not be revise, since revising as per the suggestion from the reviewer will align our manuscript towards the published reviews.

The aim of a review paper is gather good and enough information for the readers. I see that you improve the introduction section related to AM processes, however there is missing a table or chart which can categorize the "3D printing of Al alloys" based on materials or technique. You should show something new in your review although you reporting the work of the others. How about the series of alloys that you have mentioned in the conclusion "Al-Cu-Mg, Al-Cu (2XXX), Al-Si (4XXX), Al-Si-Mg (6XXX) and Al-Zn-Mg-Cu (7XXX)". Maybe it can be helpful for your category!

Thanks for the comment. We would like to highlight that the present review article is very focussed: For instance, only one 3D printing process has been focussed, which is selective laser melting and only different Al-based alloys are covered. Hence, we do not see a necessity to include a table that deals with the processing of Al-based alloy by different 3D printing processes, since that will be out of the scope of the present review article.

I do not see any discussion about the applications and the market of this process! Please add it to your review paper. That would be really helpful.

As discussed: The aim of this review is to give a critical analysis of the different Al-based alloys that are processed using the selective laser melting process (including the processing difficulties, defect formation, etc.), followed by a detailed microstructural investigation and testing of properties. Hence, we do not indulge in any applications/market/cost factors in this particular review.